# PAC Learning Linear Thresholds
# from Label Proportions

**Anand Brahmbhatt**[*]
Google Research India
anandpareshb@google.com

**Rishi Saket**[*]
Google Research India
rishisaket@google.com

**Aravindan Raghuveer**
Google Research India
araghuveer@google.com

## Abstract

Learning from label proportions (LLP) is a generalization of supervised learning in which the training data is available as sets or *bags* of feature-vectors (instances) along with the average instance-label of each bag. The goal is to train a good instance classifier. While most previous works on LLP have focused on training models on such training data, computational learnability of LLP was only recently explored by [25, 26] who showed worst case intractability of properly learning *linear threshold functions* (LTFs) from label proportions. However, their work did not rule out efficient algorithms for this problem on natural distributions.

In this work we show that it is indeed possible to efficiently learn LTFs using LTFs when given access to random bags of some label proportion in which feature-vectors are, conditioned on their labels, independently sampled from a Gaussian distribution $N(\boldsymbol{\mu}, \boldsymbol{\Sigma})$. Our work shows that a certain matrix – formed using co-variances of the differences of feature-vectors sampled from the bags with and without replacement – necessarily has its principal component, after a transformation, in the direction of the normal vector of the LTF. Our algorithm estimates the means and covariance matrices using subgaussian concentration bounds which we show can be applied to efficiently sample bags for approximating the normal direction. Using this in conjunction with novel generalization error bounds in the bag setting, we show that a low error hypothesis LTF can be identified. For some special cases of the $N(\mathbf{0}, \mathbf{I})$ distribution we provide a simpler mean estimation based algorithm. We include an experimental evaluation of our learning algorithms along with a comparison with those of [25, 26] and random LTFs, demonstrating the effectiveness of our techniques.

## 1 Introduction

In *learning from label proportions* (LLP), the training data is aggregated into sets or *bags* of feature-vectors (instances). For each bag we are given its constituent feature-vectors along with only the sum or average of their labels The goal is a to obtain a *good* instance-level classifier – one that minimizes the classification error on a test set of instances or bags. In this work we study the LLP learnability over Gaussian distributions of linear threshold functions (LTFs), also called *linear classifiers* or *halfspaces*, given by $f(\mathbf{x}) = \mathsf{pos}\left(\mathbf{r}^{\mathsf{T}}\mathbf{x} + c\right)$ where $\mathsf{pos}(a) := 1$ if $a > 0$ and $0$ otherwise.

The *probably approximately correct* (PAC) model of [29] states that a *concept* class $\mathcal{C}$ of $\{0, 1\}$-valued functions can be learnt by a *hypothesis* class $\mathcal{H}$ if there is an algorithm to efficiently obtain, using iid samples from a distribution on $(\mathbf{x}, f(\mathbf{x}))$, a hypothesis $h \in \mathcal{H}$ of arbitrarily high accuracy on that distribution, for any unknown $f \in \mathcal{C}$. If $\mathcal{H} = \mathcal{C}$ we say that $\mathcal{C}$ is *properly* learnable, for

---

[*] – equal contribution

37th Conference on Neural Information Processing Systems (NeurIPS 2023).

e.g. LTFs are known to be properly learnable using linear programming ([3]). This notion can be extended to the LLP setting – which for brevity we call PAC-LLP – as follows: distribution $D$ is over bags and their label proportions $(B, \sigma(B, f))$ where $B = \{\mathbf{x}_1, \dots, \mathbf{x}_q\}$ is a bag of feature vectors and $\sigma(B, f) = \text{Avg}\{f(\mathbf{x}) \mid \mathbf{x} \in B\}$. A bag $(B, \sigma(B, f))$ is said to be *satisfied* by $h$ iff $\sigma(B, h) = \sigma(B, f)$, and the accuracy of $h$ is the fraction of bags satisfied by it.

With the above notion of PAC-LLP, [25] studied the learnability of LTFs and rather disturbingly showed that for any constant $\varepsilon > 0$ it is NP-hard to PAC-LLP learn an LTF using an LTF which satisfies $(1/2 + \varepsilon)$-fraction of the bags when all bags are of size at most 2, which was subsequently strengthened to a $(4/9 + \varepsilon)$-factor hardness by [26] who also proved $(1/q + \varepsilon)$-factor hardness when the bag size is at most $q$, for any $q \geq 3$. This is in contrast to the supervised learning (i.e, with unit-sized bags) in which an LTF can be efficiently learnt by an LTF using linear programming. On the algorithmic side, [25] gave a *semi-definite programming* (SDP) based algorithm to find an LTF satisfying $(2/5)$-fraction of bags of size $\leq 2$, which was extended by [26] to a fairly involved SDP yielding $(1/12)$-approximation for bags of size $\leq 3$, while no non-trivial algorithms for bags of size $> 3$ are known. These results show that PAC-LLP learning LTFs using LTFs is intractable on *hard* bag distributions, and even the non-trivial algorithms for bag sizes $\leq 3$ are via complicated convex programming techniques. A natural question therefore, is whether the problem is tractable on natural distributions that may arise out of real world scenarios.

We answer the above question in the affirmative when the feature-vectors are distributed according to some (unknown) Gaussian distribution $\mathcal{D} = N(\boldsymbol{\mu}, \boldsymbol{\Sigma})$ in $d$-dimensions. Gaussian distributions are ubiquitous in machine learning and in many applications the input data distribution is modeled as multivariate Gaussians, and several previous works [8, 30] have studied learnability in Gaussian distributions. An unkown target LTF is given by $f(\mathbf{x}) := \text{pos}\left(\mathbf{r}_*^{\mathsf{T}} \mathbf{x} + c_*\right)$ where $\|\mathbf{r}_*\|_2 = 1$. Let $\mathcal{D}_a$ be the distribution of $\mathbf{x} \leftarrow \mathcal{D}$ conditioned on $f(\mathbf{x}) = a$, for $a \in \{0, 1\}$. Using this we formalize the notion of a distribution $\mathcal{O}$ on bags of size $q$ and average label $k/q$: a random bag $B$ sampled from $\mathcal{O}$ consists of $k$ iid samples from $\mathcal{D}_1$ and $(q - k)$ iid samples from $\mathcal{D}_0$. The case of $k \in \{0, q\}$ is uninteresting as all instances in such bags are either labeled 0 or 1 and traditional PAC-learning for LTFs can be employed directly. Unlike [25, 26] our objective is to directly maximize the instance-level level accuracy on $\mathcal{D}$. With this setup we informally describe our main result.

**Our PAC-LLP LTF Learner** (Informal): Assuming mild conditions on $\boldsymbol{\Sigma}, \boldsymbol{\mu}$ and $c_*$, for any $q, k \in \mathbb{Z}^+$ s.t. $1 \leq k \leq q - 1$ and $\varepsilon, \delta > 0$, there is an algorithm that samples at most $m$ bags from $\mathcal{O}$ and runs in time $O(t + m)$ and with probability $1 - \delta$ produces an LTF $h$ s.t.
$\Pr_{\mathcal{D}}[f(\mathbf{x}) \neq h(\mathbf{x})] \leq \varepsilon$ if $k \neq q/2$, and
$\Pr_{\mathcal{D}}[f(\mathbf{x}) \neq h(\mathbf{x})] \leq \varepsilon$ or $\Pr[f(\mathbf{x}) \neq (1 - h(\mathbf{x}))] \leq \varepsilon$ if $k = q/2$,
where $t, m$ are fixed polynomials in $d, q, (1/\varepsilon), \log(1/\delta)$. We also obtain a more efficient algorithm when $k \neq q/2$, $\boldsymbol{\mu} = \mathbf{0}$, $c_* = 0$ and $\boldsymbol{\Sigma} = \mathbf{I}$. The ambiguity in the case of $k = q/2$ is inherent since bags of label proportion $1/2$ consistent with an LTF $f(\mathbf{x})$ are also consistent with $(1 - f(\mathbf{x}))$.

**Remark 1.1** (Mixtures of $(q, k)$). *The training data could consist of bags of different sizes and label proportions, however typically the the maximum size of bags is bounded by (say) $Q$, and in a large enough sample we would have at least $(1/Q^2)$-fraction of bags of a particular size and label proportion and we can apply our* PAC-LLP LTF Learner *above to that subsample.*

## 1.1 Related Work

The LLP problem is motivated by many real applications where labels are available not for each feature-vector but only as the average labels of bags of feature-vectors. This may occur because of privacy and legal ([24, 33])) reasons, supervision cost ([5]) or lack of labeling instrumentation ([10]). Previous works ([9, 15, 20, 24]) on LLP have applied techniques such as such as clustering, and linear classifiers and MCMC. Specifically for LLP, assuming class conditional independence of bags, [23] gave an algorithm to learn an exponential generative model, which was further generalized by [22]. On the other hand, the work of [34] proposed a novel *proportional* SVM based algorithms which optimized the SVM loss over instance-labels which were constrained by bag-level loss w.r.t the given label-proportions. Subsequently, approaches based on deep neural nets for large-scale and multi-class data ([18, 11, 19, 21]), as well as bag pre-processing techniques ([28, 27]) have been developed. Recently, [4, 6] have proposed model training methods for either random or curated bags.

The LLP framework (as an analogue of PAC learning) was first formalized in the work of [35]. They bounded the generalization error of a trained classifier when taking the (bag, label-proportion)-pairs

as instances sampled iid from some distribution. Their loss function was different – a weaker notion than the strict bag satisfaction predicate of [25, 26]. A single-bag variant – *class ratio estimation* – of LLP was studied by [13] in which learning LTFs has a simple algorithm (see Appendix G). Nevertheless, the study of computational learning in the LLP framework has been fairly limited, apart from the works of [25, 26] whose results of learning LTFs in the LLP setting have been described earlier in this section.

In the fully supervised setting [3] showed that LTFs can be learnt using LTFs via linear programming without any distributional assumptions. Adversarial label noise makes the problem NP-hard to approximate beyond the trivial $(1/2)$-factor even using constant degree polynomial thresholds as hypothesis ([12, 14, 2]). However, under distributional assumptions a series of results ([16, 17, 1, 7]) have given efficient algorithms to learn adversarially noisy LTFs.

Next, Sec. 1.2 mathematically defines our problem statement. Sec. 1.3 states the main results of this paper. Sec. 1.4 provides an overview of our techniques. Sec. 2 mentions some preliminary results which are used in our proofs. Sec. 3 defines and analyses a subroutine which we use in all our algorithms. Sec. 4 provides a complete proof for one of our main results. Sec 5 gives brief proof sketches of our other results. Sec. 6 mentions some experiments which support of our results.

## 1.2 Problem Definition

**Definition 1.2** (Bag Oracle)**.** *Given distribution $\mathcal{D}$ over $\mathbb{R}^d$ and a target concept $f : \mathbb{R} \to \{0,1\}$, the bag oracle for size $q$ and label proportion $k/q$ $(1 \leq k \leq q-1)$, denoted by $\mathsf{Ex}(f, \mathcal{D}, q, k)$, generates a bag $\{\mathbf{x}^{(i)}\}_{i=1}^q$ such that $\mathbf{x}^{(i)}$ is independently sampled from* (i) *$\mathcal{D}_{f,1}$ for $i = \{1, \ldots, k\}$, and* (ii) *$\mathcal{D}_{f,0}$ for $i = \{k+1, \ldots, q\}$, where $\mathcal{D}_{f,a}$ is $\mathbf{x} \leftarrow \mathcal{D}$ conditioned on $f(\mathbf{x}) = a$, for $a \in \{0,1\}$.*

## 1.3 Our results

We first state our result (proved in Appendix A) for the case of standard $d$-dimensional Gaussian distribution $N(\mathbf{0}, \mathbf{I})$, homogeneous target LTF and unbalanced bags.

**Theorem 1.3.** *For $q > 2$ and $k \in \{1, \ldots, q-1\}$ s.t. $k \neq q/2$ and LTF $f(\mathbf{x}) := \mathsf{pos}(\mathbf{r}_*^\mathsf{T}\mathbf{x})$, there is an algorithm that samples $m$ iid bags from $\mathsf{Ex}(f, N(\mathbf{0},\mathbf{I}), q, k)$ and runs in $O(m)$ time to produce a hypothesis $h(\mathbf{x}) := \mathsf{pos}(\hat{\mathbf{r}}^\mathsf{T}\mathbf{x})$ s.t. w.p. at least $1 - \delta$ over the sampling, $\mathrm{Pr}_{\mathcal{D}}[f(\mathbf{x}) \neq h(\mathbf{x})] \leq \varepsilon$, for any $\varepsilon, \delta > 0$, when $m \geq O\left((d/\varepsilon^2)\log(d/\delta)\right)$.*

The above algorithm is based on estimating the mean of the bag vectors, which unfortunately does not work when $k = q/2$ or for a general covariance matrix $\mathbf{\Sigma}$. We instead use a covariance estimation based approach – albeit with a worse running time – for our next result which is proved in Sec. 4. $\lambda_{\min}$ and $\lambda_{\max}$ denote the minimum and maximum eigenvalues of the covariance matrix $\mathbf{\Sigma}$.

**Theorem 1.4.** *For $q > 2$, $k \in \{1, \ldots, q-1\}$, $f(\mathbf{x}) := \mathsf{pos}(\mathbf{r}_*^\mathsf{T}\mathbf{x})$, and positive definite $\mathbf{\Sigma}$ there is an algorithm that samples $m$ iid bags from $\mathsf{Ex}(f, N(\mathbf{0},\mathbf{\Sigma}), q, k)$ and runs in $\mathrm{poly}(m)$ time to produce a hypothesis $h(\mathbf{x}) := \mathsf{pos}(\hat{\mathbf{r}}^\mathsf{T}\mathbf{x})$ s.t. w.p. at least $1 - \delta$ over the sampling*

- *if $k \neq q/2$, $\mathrm{Pr}_{\mathcal{D}}[f(\mathbf{x}) \neq h(\mathbf{x})] \leq \varepsilon$, and*
- *if $k = q/2$, $\min\{\mathrm{Pr}_{\mathcal{D}}[f(\mathbf{x}) \neq h(\mathbf{x})], \mathrm{Pr}_{\mathcal{D}}[f(\mathbf{x}) \neq \tilde{h}(\mathbf{x})]\} \leq \varepsilon$, where $\tilde{h}(\mathbf{x}) := \mathsf{pos}(-\hat{\mathbf{r}}^\mathsf{T}\mathbf{x})$*

*for any $\varepsilon, \delta > 0$, when $m \geq O((d/\varepsilon^4)\log(d/\delta)(\lambda_{\max}/\lambda_{\min})^6 q^8)$.*

Our general result stated below (proved in Appendix C), extends our algorithmic methods to the case of non-centered Gaussian space and non-homogeneous LTFs.

**Theorem 1.5.** *For $q > 2$, $k \in \{1, \ldots, q-1\}$, $f(\mathbf{x}) := \mathsf{pos}(\mathbf{r}_*^\mathsf{T}\mathbf{x} + c_*)$, and positive definite $\mathbf{\Sigma}$ there is an algorithm that samples $m$ iid bags from $\mathsf{Ex}(f, N(\boldsymbol{\mu},\mathbf{\Sigma}), q, k)$ and runs in $\mathrm{poly}(m)$ time to produce a hypothesis $h(\mathbf{x}) := \mathsf{pos}(\hat{\mathbf{r}}^\mathsf{T}\mathbf{x} + \hat{c})$ s.t. w.p. at least $1 - \delta$ over the sampling*

- *if $k \neq q/2$, $\mathrm{Pr}_{\mathcal{D}}[f(\mathbf{x}) \neq h(\mathbf{x})] \leq \varepsilon$, and*
- *if $k = q/2$, $\min\left\{\mathrm{Pr}_{\mathcal{D}}[f(\mathbf{x}) \neq h(\mathbf{x})], \mathrm{Pr}_{\mathcal{D}}[f(\mathbf{x}) \neq \tilde{h}(\mathbf{x})]\right\} \leq \varepsilon$, $\tilde{h}(\mathbf{x}) := \mathsf{pos}(-\hat{\mathbf{r}}^\mathsf{T}\mathbf{x} - \hat{c})$*

*for any $\varepsilon, \delta > 0$, when $m \geq O\left((d/\varepsilon^4)\frac{O(\ell^2)}{(\Phi(\ell)(1-\Phi(\ell)))^2}\log(d/\delta)\left(\frac{\lambda_{\max}}{\lambda_{\min}}\right)^4\left(\frac{\sqrt{\lambda_{\max}}+\|\boldsymbol{\mu}\|_2}{\sqrt{\lambda_{\min}}}\right)^4 q^8\right)$ where $\Phi(.)$ is the standard Gaussian cdf and $\ell = -\frac{c_* + \mathbf{r}_*^\mathsf{T}\boldsymbol{\mu}}{\|\mathbf{\Sigma}^{1/2}\mathbf{r}_*\|_2}$*

The value of $\hat{\mathbf{r}}$ output by our algorithms is a close estimate of $\mathbf{r}_*$ (or possibly $-\mathbf{r}_*$ in the case of balanced bags). Note that our algorithms do not require knowledge of $\boldsymbol{\mu}$ or $\boldsymbol{\Sigma}$, and only the derived parameters in Thms. 1.4 and 1.5 are used for the sample complexity bounds. They are based on the certain properties of the empirical mean-vectors and covariance matrices formed by sampling vectors or pairs of vectors from random bags of the bag oracle. An empirical mean based approach has been previously developed by [23] in the LLP setting to estimate the parameters of an exponential generative model, when bag distributions satisfy the so-called class conditioned independencei.e., given its label, the feature-vector distribution is same for all the bags. These techniques were extended by [22] to linear classifiers with loss functions satisfying certain smoothness conditions. While the bag oracle in our setup satisfies such conditioned independence, we aim to minimize the instance classification error on which the techniques of [23, 22] are not applicable.

For the case when $q = 1$ (ordinary classification), the sample complexity is $O(d/\varepsilon \log(d/\delta))$ as one can solve a linear program to obtain an LTF and then use uniform convergence to bound the generalization error. The sample complexity expressions in Theorems 1.3, 1.4 and 1.5 have the same dependence on $d$ and $\delta$. However, they have higher powers of $1/\varepsilon$. They also include other parameters like the bag size ($q$), condition number of $\boldsymbol{\Sigma}$ ($\lambda_{\max}/\lambda_{\min}$) and the normalized distance of mean of the Gaussian to the LTF ($l$). The origins and significance of these discrepancies are discussed in Sec. 1.4.

## 1.4  Our Techniques

**Theorem 1.3: Case** $N(\mathbf{0}, \mathbf{I})$, $f(\mathbf{x}) = \mathsf{pos}(\mathbf{r}_*^\mathsf{T}\mathbf{x})$, $k \neq q/2$**.** Assume that $k > q/2$. A randomly sampled bag with label proportion $k/q$ has $k$ vectors iid sampled from the positive side of the separating hyperplane passing through origin, and $(q - k)$ iid sampled from its negative side. It is easy to see that the expected sum of the vectors vanishes in all directions orthogonal to the normal vector $\mathbf{r}_*$, and in the direction of $\mathbf{r}_*$ it has a constant magnitude. The case of $k < q/2$ is analogous with the direction of the expectation opposite to $\mathbf{r}_*$. Sampling a sufficient number of bags and a random vector from each of them, and taking their normalized expectation (negating if $k < q/2$) yields the a close estimate $\hat{\mathbf{r}}$ of $\mathbf{r}_*$, which in turn implies low classification error. The sample complexity is the same as that for mean estimation bag-vectors (see Section 3) and thus the power of $1/\varepsilon$ is 2.

This simple approach however does not work when $r = q/2$, in which case the expectation vanishes completely, or for general Gaussian distributions which (even if centered) could be skewed in arbitrary directions. We present our variance based method to handle these cases.

**Theorem 1.4: Case** $N(\mathbf{0}, \boldsymbol{\Sigma})$, $f(\mathbf{x}) = \mathsf{pos}(\mathbf{r}_*^\mathsf{T}\mathbf{x})$**.** To convey the main idea of our approach, consider two different ways of sampling two feature-vectors from a random bag of the oracle. The first way is to sample two feature-vectors $\mathbf{Z}_1, \mathbf{Z}_2$ independently and u.a.r from a random bag. In this case, the probability that they have different labels (given by $f$) is $2k(q - k)/q^2$. The second way is to sample a random *pair* $\tilde{\mathbf{Z}}_1, \tilde{\mathbf{Z}}_2$ of feature-vectors i.e., without replacement. In this case, the probability of different labels is $2k(q - k)/(q(q - 1))$ which is strictly greater than $2k(q - k)/q^2$. Since the labels are given by thresholding in the direction of $\mathbf{r}_*$, this suggests that the variance of $(\tilde{\mathbf{Z}}_1 - \tilde{\mathbf{Z}}_2)$ w.r.t. that of $(\mathbf{Z}_1 - \mathbf{Z}_2)$ is maximized in the direction of $\mathbf{r}_*$. Indeed, let $\boldsymbol{\Sigma}_D := \mathrm{Var}[\tilde{\mathbf{Z}}_1 - \tilde{\mathbf{Z}}_2]$ be the *pair* covariance matrix and let $\boldsymbol{\Sigma}_B := \mathrm{Var}[\mathbf{Z}_1] = (1/2)\mathrm{Var}[\mathbf{Z}_1 - \mathbf{Z}_2]$ be the *bag* covariance matrix. Then we show that $\pm\mathbf{r}_* = \mathrm{argmax}_{\mathbf{r}}\rho(\mathbf{r})$ where $\rho(\mathbf{r}) := \frac{\mathbf{r}^\mathsf{T}\boldsymbol{\Sigma}_D\mathbf{r}}{\mathbf{r}^\mathsf{T}\boldsymbol{\Sigma}_B\mathbf{r}}$. A simple transformation gives us that

$$\pm\mathbf{r}_* = \boldsymbol{\Sigma}_B^{-1/2}\mathsf{PrincipalEigenVector}(\boldsymbol{\Sigma}_B^{-1/2}\boldsymbol{\Sigma}_D\boldsymbol{\Sigma}_B^{-1/2}) \tag{1}$$

This suggests the following algorithm: sample enough bags to construct the corresponding empirical estimates $\hat{\boldsymbol{\Sigma}}_D$ and $\hat{\boldsymbol{\Sigma}}_B$ and then compute the empirical proxy of the RHS of (1). We show that using close enough empirical estimates w.h.p the algorithm computes a vector $\hat{\mathbf{r}}$ s.t. one of $\pm\hat{\mathbf{r}}$ is close to $\mathbf{r}_*$, and via a geometric stability argument this implies that one of $\pm\hat{\mathbf{r}}$ yields an LTF that has small instance-level classification error.

At this point, if $k = q/2$, there is no way to identify the correct solution from $\pm\hat{\mathbf{r}}$, since a balanced bag, if consistent with an LTF, is also consistent with its complement. On the other hand, if $k \neq q/2$ we can obtain the correct solution as follows. It is easy to show that since $f$ is homogeneous and the instance distribution is a centered Gaussian, the measure of $\{\mathbf{x} \mid f(\mathbf{x}) = a\}$ is $1/2$ for $a = \{0, 1\}$. Thus, one of $h(\mathbf{x}) := \mathsf{pos}(\hat{\mathbf{r}}^\mathsf{T}\mathbf{x})$, $\tilde{h}(\mathbf{x}) = \mathsf{pos}(-\hat{\mathbf{r}}^\mathsf{T}\mathbf{x})$ will have a high bag satisfaction accuracy. Thus, a large enough sample of bags can be used to identify one of $h, \tilde{h}$ having a high bag satisfaction

accuracy. Lastly, we use a novel generalization error bound (see below) to show that the identified LTF also has a high instance classification accuracy.

The algorithm incurs a sample complexity of $O\left((q\lambda_{\max}/(\varepsilon\lambda_{\min}))^4\right)$ to estimate $\mathbf{\Sigma}_D$ and $\mathbf{\Sigma}_B$ accurately so that the distance between $\hat{\mathbf{r}}$ (or $-\hat{\mathbf{r}}$) and $\mathbf{r}_*$ is sufficiently bounded. (see Lemmas 3.1 and 4.1). In fact, the distance is required to be $O\left((\varepsilon/q)\sqrt{\lambda_{\min}/\lambda_{\max}}\right)$ to translate the geometric bound into an $\varepsilon$-misclassification error bound, thereby incurring a further factor of $O\left(q^4(\lambda_{\max}/\lambda_{\min})^2\right)$ in the sample complexity (see Lemma 2.3). The higher powers in the dependencies are mainly due to the second moment estimates which degrade with larger values of $(\lambda_{\max}/\lambda_{\min})$. Note that the sample complexity explicitly depends on the bag size $q$ (and not just the label proportion $k/q$). This is because the probability of of sampling (without replacement from a bag) a pair of differently labeled feature-vectors is $2(k/q)(1-k/q)/(1-1/q)$. Keeping $k/q$ the same, this probability decreases with increasing bag size, thereby increasing the sample complexity for larger bags.

**Theorem 1.5: Case $N(\boldsymbol{\mu}, \mathbf{\Sigma})$, $f(\mathbf{x}) = \mathsf{pos}(\mathbf{r}_*^\mathsf{T}\mathbf{x} + c_*)$.** We show that (1) also holds in this case, and therefore we use a similar approach of empirically estimating the pair and bag covariance matrices solving (1) works in principle. However, there are complications, in particular the presence of $\boldsymbol{\mu}$ and $c_*$ degrades the error bounds in the analysis, thus increasing the sample complexity of the algorithm. This is because the measures of $\{\mathbf{x} \mid f(\mathbf{x}) = a\}$ for $a = \{0, 1\}$ could be highly skewed if $\|\mu\|_2$ and/or $|c_*|$ is large. Moreover, the spectral algorithm only gives a solution $\pm\hat{\mathbf{r}}$ for $\mathbf{r}_*$. An additional step is required to obtain an estimate of $c_*$. This we accomplish using the following procedure which, given a sample of $s$ bags and any $\mathbf{r}$ outputs a $\hat{c}$ which has the following property: if $s^* = \max_c\{$no. of bags satisfied by $\mathsf{pos}(\mathbf{r}^\mathsf{T}\mathbf{x} + c)\}$, then $\hat{c}$ will satisfy at least $s^* - 1$ bags. This is done by ordering the values $\mathbf{r}^\mathsf{T}\mathbf{x}$ of the vectors $\mathbf{x}$ within each bag in decreasing order, and then constructing set of the $k$th values of each bag. Out of these $s$ values, the one which taken as $c$ in $\mathsf{pos}(\mathbf{r}^\mathsf{T}\mathbf{x} + c)$ satisfies the most bags, is chosen to be $\hat{c}$.

Due to the non-homogeneity of the Gaussian distribution and the LTF $f$, the application of Lemma 2.3 instead incurs an $O\left(q^4((\sqrt{\lambda_{\max}} + \|\boldsymbol{\mu}\|_2)/\sqrt{\lambda_{\min}})^4\right)$ factor in the sample complexity, while a factor of $O(\ell^2)$ is towards the estimation of $\mathbf{\Sigma}_B$ and $\mathbf{\Sigma}_D$. Note that $\ell$ is the distance from $\boldsymbol{\mu}$ to the hyperplane of $f$ normalized by the stretch induced by $\mathbf{\Sigma}$, and thus a larger value of $\ell$ implies a lower density near the hyperplane leading to an increased sample complexity. Lastly, a further blowup by $1/(\Phi(\ell)(1 - \Phi(\ell)))^2$ comes from bounding the sample error from geometric bound between $\hat{\mathbf{r}}$ and $\mathbf{r}_*$, and is required for a sufficiently accurate approximation of $c_*$.

**Generalization Error Bounds.** We prove (Thm. 2.2) bounds on the generalization of the error of a hypothesis LTF $h$ in satisfying sampled bags to its distributional instance-level error. Using this, we are able to distinguish (for $k \neq q/2$) between the two possible solutions our principal component algorithm yields – the one which satisfies more of the sampled bags has w.h.p. low instance-level error. For proving these bounds, the first step is to use a bag-level generalization error bound shown by [26] using the techniques of [35]. Next, we show that low distributional bag satisfaction error by $h$ implies low instance level error. This involves a fairly combinatorial analysis of two independent binomial random variables formed from the incorrectly classified labels within a random bag. Essentially, unless $h$ closely aligns with $f$ at the instance level, with significant probability there will be an imbalance in these two random variables leading to $h$ not satisfying the bag.

**Subgaussian concentration bounds.** The standard estimation bounds for Gaussians are not directly applicable in our case, since the random vector sampled from a random bag is biased according to its label given by $f$, and is therefore not a Gaussian vector. To obtain sample complexity bounds linear in $\log(1/\delta)$ we use subgaussian concentration bounds for mean and covariance estimation ([32, 31]). For this, we show $O(\ell)$ bound on the expectation and subgaussian norm of the thresholded Gaussian given by $\{g \sim N(0, 1) \mid g > \ell\}$ for some $\ell > 0$. The random vectors of interest to us are (in a transformed space) distributed as a combination of thresholded Gaussians in one of the coordinates, and $N(0, 1)$ in the rest. We show that they satisfy the $O(\ell)$ bound on their subgaussian norm and admit the corresponding subgaussian Hoeffding (for empirical mean) and empirical covariance concentration bounds. Based on this, in Sec. 3 we abstract out the procedure used in our learning algorithms for obtaining the relevant mean and covariance estimates.

**Experiments.** We include in Sec. 6 an experimental evaluation of our learning algorithms along with a comparison of with those of [25, 26] and random LTFs, demonstrating the effectiveness of our techniques.

## 2 Preliminaries

We begin with some useful linear algebraic notions. Let $\lambda_{\max}(\mathbf{A})$ and $\lambda_{\min}(\mathbf{A})$ denote the maximum and minimum eigenvalue of a real symmetric matrix $\mathbf{A}$. The *operator norm* $\|\mathbf{A}\|_2 := \max_{\|\mathbf{x}\|_2=1} \|\mathbf{A}\mathbf{x}\|_2$ for such matrices is given by $\lambda_{\max}(\mathbf{A})$.

We shall restrict our attention to symmetric *positive definite* (p.d.) matrices $\mathbf{A}$ which satisfy $\mathbf{x}^\mathsf{T}\mathbf{A}\mathbf{x} > 0$ for all non-zero vectors $\mathbf{x}$, implying that $\lambda_{\min}(\mathbf{A}) > 0$ and $\mathbf{A}^{-1}$ exists and is symmetric p.d. as well. Further, for such matrices $\mathbf{A}$, $\mathbf{A}^{1/2}$ is well defined to be the unique symmetric p.d. matrix $\mathbf{B}$ satisfying $\mathbf{B}\mathbf{B} = \mathbf{A}$. The eigenvalues of $\mathbf{A}^{1/2}$ are the square-roots of those of $\mathbf{A}$. We have the following lemma which is proved in Appendix B.6.

**Lemma 2.1.** *Let $\mathbf{A}$ and $\mathbf{B}$ be symmetric p.d. matrices such that $\|\mathbf{A}-\mathbf{B}\| \leq \varepsilon_1\|\mathbf{A}\|_2$. Let $\mathbf{r}_1, \mathbf{r}_2 \in \mathbb{R}^d$ be two unit vectors such that $\|\mathbf{r}_1 - \mathbf{r}_2\|_2 \leq \varepsilon_2$. Then, $\left\|\frac{\mathbf{A}\mathbf{r}_1}{\|\mathbf{A}\mathbf{r}_1\|_2} - \frac{\mathbf{B}\mathbf{r}_2}{\|\mathbf{B}\mathbf{r}_2\|_2}\right\|_2 \leq 4\frac{\lambda_{\max}(\mathbf{A})}{\lambda_{\min}(\mathbf{A})}(\varepsilon_2 + \varepsilon_1)$ when $\frac{\lambda_{\max}(\mathbf{A})}{\lambda_{\min}(\mathbf{A})}(\varepsilon_2 + \varepsilon_1) \leq \frac{1}{2}$.*

**Bag Oracle and related statistics.** Let $\mathcal{O} := \mathsf{Ex}(f, \mathcal{D}, q, k)$ be any bag oracle with $k \in \{1, \ldots, q-1\}$ for an LTF $f(\mathbf{x}) := \mathbf{r}_*^\mathsf{T}\mathbf{x} + c_*$ in $d$-dimensions, and let $\mathcal{M}$ be a collection of $m$ bags sampled iid from the oracle. Define for any hypothesis LTF $h$,

$$\mathsf{BagErr}_{\mathrm{oracle}}(h, f, \mathcal{D}, q, k) := \Pr_{B \leftarrow \mathcal{O}} \left[\mathrm{Avg}\{h(\mathbf{x}) \mid \mathbf{x} \in B\} \neq k/q\right], \text{ and,} \tag{2}$$

$$\mathsf{BagErr}_{\mathrm{sample}}(h, \mathcal{M}) := |\{B \in \mathcal{M} \mid \mathrm{Avg}\{h(\mathbf{x}) \mid \mathbf{x} \in B\} \neq k/q\}| / m. \tag{3}$$

We define the following statistical quantities related to $\mathcal{O}$. Let $\mathbf{X}$ be a random feature-vector sampled uniformly from a random bag sampled from $\mathcal{O}$. Let,

$$\boldsymbol{\mu}_B := \mathbb{E}[\mathbf{X}] \qquad \text{and,} \qquad \boldsymbol{\Sigma}_B := \mathbb{E}\left[(\mathbf{X} - \boldsymbol{\mu}_B)(\mathbf{X} - \boldsymbol{\mu}_B)^\mathsf{T}\right] = \mathrm{Var}[\mathbf{X}]. \tag{4}$$

Now, let $\mathbf{Z} = \mathbf{X}_1 - \mathbf{X}_2$ where $(\mathbf{X}_1, \mathbf{X}_2)$ are a random pair of feature-vectors sampled (without replacement) from a random bag sampled from $\mathcal{O}$. Clearly $\mathbb{E}[\mathbf{Z}] = \mathbf{0}$. Define

$$\boldsymbol{\Sigma}_D := \mathbb{E}\left[\mathbf{Z}\mathbf{Z}^\mathsf{T}\right] = \mathrm{Var}[\mathbf{Z}]. \tag{5}$$

**Generalization and stability bounds.** We prove in Appendix D.1 the following generalization bound from bag classification error to instance classification error.

**Theorem 2.2.** *For any $\varepsilon < 1/4q$ if $\mathsf{BagErr}_{\mathrm{sample}}(h, \mathcal{M}) \leq \varepsilon$ then,*
*(i) if $k \neq q/2$, $\Pr_{\mathcal{D}}[f(\mathbf{x}) \neq h(\mathbf{x})] \leq 4\varepsilon$, and*
*(ii) if $k = q/2$, $\Pr_{\mathcal{D}}[f(\mathbf{x}) \neq h(\mathbf{x})] \leq 4\varepsilon$ or $\Pr[f(\mathbf{x}) \neq (1 - h(\mathbf{x}))] \leq 4\varepsilon$,*
*w.p. $1 - \delta$, when $m \geq C_0 d \left(\log q + \log(1/\delta)\right)/\varepsilon^2$, for any $\delta > 0$ and absolute constant $C_0 > 0$.*

In some cases we directly obtain geometric bounds on the hypothesis classifier and the following lemma (proved in Appendix D.2) allows us to straightaway bound the classification error.

**Lemma 2.3.** *Suppose $\|\mathbf{r} - \hat{\mathbf{r}}\|_2 \leq \varepsilon_1$ for unit vectors $\mathbf{r}, \hat{\mathbf{r}}$. Then, $\Pr\left[\mathsf{pos}\left(\mathbf{r}^T\mathbf{X} + c\right) \neq \mathsf{pos}\left(\hat{\mathbf{r}}^T\mathbf{X} + c\right)\right] \leq \varepsilon$ where $\varepsilon = \varepsilon_1(c_0\sqrt{\lambda_{\max}/\lambda_{\min}} + c_1\|\boldsymbol{\mu}\|_2/\sqrt{\lambda_{\min}})$ for some absolute constants $c_0, c_1 > 0$ and $\lambda_{\max}, \lambda_{\min}$ are the maximum and minimum eigenvalues of $\boldsymbol{\Sigma}$ respectively.*

## 3 Bag distribution statistics estimation

We provide the following estimator for $\boldsymbol{\mu}_B, \boldsymbol{\Sigma}_B$ and $\boldsymbol{\Sigma}_D$ defined in (4) and (5). We have the following lemma – which follows from the subgaussian distribution based mean and covariance concentration bounds shown for thresholded Gaussians (see Appendix E) – whose proof is given in Appendix E.3.

**Lemma 3.1.** *If $m \geq O\left((d/\varepsilon^2)O(\ell^2)\log(d/\delta)\right)$ where $\ell$ is as given in Lemma E.13 then Algorithm 1 returns $\hat{\boldsymbol{\mu}}_B, \hat{\boldsymbol{\Sigma}}_B, \hat{\boldsymbol{\Sigma}}_D$ such that $\|\hat{\boldsymbol{\mu}}_B - \boldsymbol{\mu}_B\|_2 \leq \varepsilon\sqrt{\lambda_{\max}}/2$, $\|\hat{\boldsymbol{\Sigma}}_B - \boldsymbol{\Sigma}_B\|_2 \leq \varepsilon\lambda_{\max}$, and $\|\hat{\boldsymbol{\Sigma}}_D - \boldsymbol{\Sigma}_D\|_2 \leq \varepsilon\lambda_{\max}$, w.p. at least $1 - \delta$, for any $\varepsilon, \delta > 0$. Here $\lambda_{\max}$ is the maximum eigenvalue of $\boldsymbol{\Sigma}$.*

---
**Algorithm 1** MeanCovsEstimator.
---
**Input:** $\mathsf{Ex}(f, \mathcal{D} = N(\boldsymbol{\mu}, \boldsymbol{\Sigma}), q, k), m$, where $f = \mathsf{pos}\left(\mathbf{r}^\mathsf{T}\mathbf{X} + c\right)$.
1. Sample $m$ bags from $\mathsf{Ex}(f, \mathcal{D}, q, k)$. Let $\{B_i\}_{i=1}^m$ be the sampled bags.
2. $V := \{\mathbf{x}_i \mid \mathbf{x}_i \text{ u.a.r. } \leftarrow B_i, i \in \{1, \dots, m\}\}$.
3. $\hat{\boldsymbol{\mu}}_B = \sum_{\mathbf{x} \in V} \mathbf{x}/m|$.
4. $\hat{\boldsymbol{\Sigma}}_B = \Sigma_{\mathbf{x} \in V}(\mathbf{x} - \boldsymbol{\mu}_B)(\mathbf{x} - \boldsymbol{\mu}_B)^\mathsf{T}/m$.
5. Sample $m$ bags from $\mathsf{Ex}(f, \mathcal{D}, q, k)$. Let $\{\tilde{B}_i\}_{i=1}^m$ be the sampled bags.
6. $\tilde{V} := \{\bar{\mathbf{x}}_i = \mathbf{x}_i - \tilde{\mathbf{x}}_i \mid (\mathbf{x}_i, \tilde{\mathbf{x}}_i) \text{ u.a.r. without replacement from } \tilde{B}_i, i \in \{1, \dots, m\}\}$.
7. $\hat{\boldsymbol{\Sigma}}_D = \Sigma_{\mathbf{z} \in \tilde{V}} \mathbf{z}\mathbf{z}^\mathsf{T}/m$.
8. **Return:** $\hat{\boldsymbol{\mu}}_B, \hat{\boldsymbol{\Sigma}}_B, \hat{\boldsymbol{\Sigma}}_D$.
---

## 4  Proof of Theorem 1.4

For the setting of Theorem 1.4, we provide Algorithm 2. It uses as a subroutine a polynomial time procedure PrincipalEigenVector for the principal eigen-vector of a symmetric matrix, and first computes two LTFs given by a normal vector and its negation, returning the one that has lower error on a sampled collection of bags.

---
**Algorithm 2** PAC Learner for no-offset LTFs over $N(\mathbf{0}, \boldsymbol{\Sigma})$
---
**Input:** $\mathcal{O} = \mathsf{Ex}(f, \mathcal{D} = N(\mathbf{0}, \boldsymbol{\Sigma}), q, k), m, s$, where $f(\mathbf{x}) = \mathsf{pos}\left(\mathbf{r}_*^\mathsf{T}\mathbf{x}\right), \|\mathbf{r}_*\|_2 = 1$.
1. Compute $\hat{\boldsymbol{\Sigma}}_B, \hat{\boldsymbol{\Sigma}}_D$ using MeanCovsEstimator with $m$ samples.
2. $\bar{\mathbf{r}} = \hat{\boldsymbol{\Sigma}}_B^{-1/2}\mathsf{PrincipalEigenVector}(\hat{\boldsymbol{\Sigma}}_B^{-1/2}\hat{\boldsymbol{\Sigma}}_D\hat{\boldsymbol{\Sigma}}_B^{-1/2})$ if $\hat{\boldsymbol{\Sigma}}_B^{-1/2}$ exists, else exit.
3. Let $\hat{\mathbf{r}} = \bar{\mathbf{r}}/\|\bar{\mathbf{r}}\|_2$. 4. If $k = q/2$ **return:** $h = \mathsf{pos}\left(\hat{\mathbf{r}}^\mathsf{T}\mathbf{X}\right)$, else
    a. Let $\tilde{h} = \mathsf{pos}\left(-\hat{\mathbf{r}}^\mathsf{T}\mathbf{X}\right)$.
    b. Sample a collection $\mathcal{M}$ of $s$ bags from $\mathcal{O}$.
    c. **Return** $h^* \in \{h, \tilde{h}\}$ which has lower $\mathsf{BagErr}_{\text{sample}}(h^*, \mathcal{M})$.
---

**Lemma 4.1.** *For any $\varepsilon, \delta \in (0, 1)$, if $m \geq O((d/\varepsilon^4)\log(d/\delta)(\lambda_{\max}/\lambda_{\min})^4 q^4)$, then $\hat{\mathbf{r}}$ computed in Step 3 of Alg. 2 satisfies $\min\{\|\hat{\mathbf{r}} - \mathbf{r}_*\|_2, \|\hat{\mathbf{r}} + \mathbf{r}_*\|_2\} \leq \varepsilon$, w.p. $1 - \delta/2$.*

The above, whose proof is deferred to Sec. 4.1, is used in conjunction with the following lemma.

**Lemma 4.2.** *Let $k \neq q/2$, $\varepsilon, \delta \in (0, 1)$ and suppose $\hat{\mathbf{r}}$ computed in Step 3 of Alg. 2 satisfies $\min\{\|\hat{\mathbf{r}} - \mathbf{r}_*\|_2, \|\hat{\mathbf{r}} + \mathbf{r}_*\|_2\} \leq \varepsilon$,. Then, with $s \geq O\left(d(\log q + \log(1/\delta))/\varepsilon^2\right)$, $h^*$ in Step. 3.c satisfies $\Pr_{\mathcal{D}}\left[h^*(\mathbf{x}) \neq f(\mathbf{x})\right] \leq 16c_0 q\varepsilon\sqrt{\frac{\lambda_{\max}}{\lambda_{\min}}}$ w.p. $1 - \delta/2$, where constant $c_0 > 0$ is from Lem. 2.3.*

With the above we complete the proof of Theorem 1.4 as follows.

*Proof.* (of Theorem 1.4) Let the parameters $\delta, \varepsilon$ be as given in the statement of the theorem.

For $k = q/2$, we use $O(\varepsilon\sqrt{\lambda_{\min}/\lambda_{\max}})$ for the error bound in Lemma 4.1 thereby taking $m = O((d/\varepsilon^4)\log(d/\delta)(\lambda_{\max}/\lambda_{\min})^6 q^4)$ in Alg. 2, so that Lemma 4.1 along with Lemma 2.3 yields the desired misclassification error bound of $\varepsilon$ for one of $h, \tilde{h}$.

For $k \neq q/2$, we use $O(\varepsilon\sqrt{\lambda_{\min}/\lambda_{\max}}/q)$ for the error bound in Lemma 4.1. Taking $m = O((d/\varepsilon^4)\log(d/\delta)(\lambda_{\max}/\lambda_{\min})^6 q^8)$ in Alg. 2 we obtain the following bound: $\min\{\|\hat{\mathbf{r}} - \mathbf{r}_*\|_2, \|\hat{\mathbf{r}} + \mathbf{r}_*\|_2\} \leq \varepsilon\sqrt{\lambda_{\min}/\lambda_{\max}}/(16c_0 q)$ with probability $1 - \delta/2$. Using $s \geq O\left(d(\log q + \log(1/\delta))q^2\frac{\lambda_{\max}}{\varepsilon^2\lambda_{\min}}\right)$, Lemma 4.2 yields the desired misclassification error bound of $\varepsilon$ on $h^*$ w.p. $1 - \delta$. $\qquad\square$

*Proof.* (of Lemma 4.2) Applying Lemma 2.3 we obtain that at least one of $h, \tilde{h}$ has an instance misclassification error of at most $O(\varepsilon\sqrt{\lambda_{\max}/\lambda_{\min}})$. WLOG assume that $h$ satisfies this error bound i.e., $\Pr_{\mathcal{D}}[f(\mathbf{x}) \neq h(\mathbf{x})] \leq c_0\varepsilon\sqrt{\lambda_{\max}/\lambda_{\min}} =: \varepsilon'$. Since the separating hyperplane of the LTF $f$

passes through the origin, and $\mathcal{D} = N(\mathbf{0}, \mathbf{\Sigma})$ is centered, $\mathrm{Pr}_{\mathcal{D}}[f(\mathbf{x}) = 1] = \mathrm{Pr}_{\mathcal{D}}[f(\mathbf{x}) = 0] = 1/2$. Thus,

$$\Pr_{\mathcal{D}}[h(x) \neq f(x) \mid f(\mathbf{x}) = 1], \Pr_{\mathcal{D}}[[h(x) \neq f(x) \mid f(\mathbf{x}) = 0] \leq 2\varepsilon'.$$

Therefore, the probability that a random bag from the oracle contains a feature vector on which $f$ and $h$ disagree is at most $2q\varepsilon'$. Applying the Chernoff bound (see Appendix B.1) we obtain that with probability at least $1 - \delta/6$, $\mathsf{BagErr}_{\mathrm{sample}}(h, \mathcal{M}) \leq 4q\varepsilon'$. Therefore, in Step 3.c. $h^*$ satisfies $\mathsf{BagErr}_{\mathrm{sample}}(h^*, \mathcal{M}) \leq 4q\varepsilon'$

On the other hand, applying Theorem 2.2, except with probability $\delta/3$, $\mathrm{Pr}_{\mathcal{D}}[f(\mathbf{x}) \neq h^*(\mathbf{x})] \leq 16q\varepsilon' = 16c_0 q\varepsilon\sqrt{\lambda_{\max}/\lambda_{\min}}$. Therefore, except with probability $\delta/2$, the bound in Lemma 4.2 holds. $\qquad\square$

## 4.1 Proof of Lemma 4.1

We define and bound a few useful quantities depending on $k, q, \lambda_{\min}$ and $\lambda_{\max}$ using $1 \leq k \leq q - 1$.

**Definition 4.3.** *Define, (i)* $\kappa_1 := \left(\frac{2k}{q} - 1\right)^2 \frac{2}{\pi}$ *so that* $0 \leq \kappa_1 \leq 2/\pi$, *(ii)* $\kappa_2 := \frac{1}{q-1}\frac{k}{q}\left(1 - \frac{k}{q}\right)\frac{16}{\pi}$ *so that* $\frac{16}{\pi q^2} \leq \kappa_2 \leq \frac{4}{\pi(q-1)}$, *(iii)* $\kappa_3 := \frac{\kappa_2}{1-\kappa_1}$ *so that* $\frac{16}{\pi q^2} \leq \kappa_3 \leq \frac{4}{(\pi-2)(q-1)}$, *and (iv)* $\theta := \frac{2\lambda_{\max}}{\lambda_{\min}}\left(\frac{1}{2-\max(0,2\kappa_1-\kappa_2)} + \frac{1}{1-\kappa_1}\right)$ *so that* $\frac{3\lambda_{\max}}{\lambda_{\min}} \leq \theta \leq \frac{3\lambda_{\max}}{(1-2/\pi)\lambda_{\min}}$.

For the analysis we begin by showing in the following lemma that $\hat{\mathbf{r}}$ in the algorithms is indeed $\pm\mathbf{r}_*$ if the covariance estimates were the actual covariances.

**Lemma 4.4.** *The ratio* $\rho(\mathbf{r}) := \mathbf{r}^{\mathsf{T}}\mathbf{\Sigma}_D\mathbf{r}/\mathbf{r}^{\mathsf{T}}\mathbf{\Sigma}_B\mathbf{r}$ *is maximized when* $\mathbf{r} = \pm\mathbf{r}_*$. *Moreover,*

$$\rho(\mathbf{r}) = 2 + \frac{\gamma(\mathbf{r})^2\kappa_2}{1 - \gamma(\mathbf{r})^2\kappa_1} \qquad where \qquad \gamma(\mathbf{r}) := \frac{\mathbf{r}^{\mathsf{T}}\mathbf{\Sigma}\mathbf{r}_*}{\sqrt{\mathbf{r}^{\mathsf{T}}\mathbf{\Sigma}\mathbf{r}}\sqrt{\mathbf{r}_*^{\mathsf{T}}\mathbf{\Sigma}\mathbf{r}_*}} \quad and$$

$$\mathbf{r}^{\mathsf{T}}\mathbf{\Sigma}_B\mathbf{r} = \mathbf{r}^{\mathsf{T}}\mathbf{\Sigma}\mathbf{r}(1 - \gamma(\mathbf{r})^2\kappa_1), \quad \mathbf{r}^{\mathsf{T}}\mathbf{\Sigma}_D\mathbf{r} = \mathbf{r}^{\mathsf{T}}\mathbf{\Sigma}\mathbf{r}(2 - \gamma(\mathbf{r})^2(2\kappa_1 - \kappa_2))$$

*Proof.* Let $\mathbf{\Gamma} := \mathbf{\Sigma}^{1/2}$, then $\mathbf{X} \sim N(\mathbf{0}, \mathbf{\Sigma}) \Leftrightarrow \mathbf{X} = \mathbf{\Gamma}\mathbf{Z}$ where $\mathbf{Z} \sim N(\mathbf{0}, \mathbf{I})$. Further, $\mathsf{pos}\left(\mathbf{r}^{\mathsf{T}}\mathbf{X}\right) = \mathsf{pos}\left(\mathbf{u}^{\mathsf{T}}\mathbf{Z}\right)$ where $\mathbf{u} = \mathbf{\Gamma}\mathbf{r}/\|\mathbf{\Gamma}\mathbf{r}\|_2$. Using this, we can let $\mathbf{X}_B = \mathbf{\Gamma}\mathbf{Z}_B$ as a random feature-vector sampled uniformly from a random bag sampled from $\mathcal{O}$. Also, let $\mathbf{X}_D = \mathbf{\Gamma}\mathbf{Z}_D$ be the difference of two random feature vectors sampled uniformly without replacement from a random bag sampled from $\mathcal{O}$. Observe that the ratio $\rho(\mathbf{r}) = \mathrm{Var}[\mathbf{r}^{\mathsf{T}}\mathbf{X}_D]/\mathrm{Var}[\mathbf{r}^{\mathsf{T}}\mathbf{X}_B] = \mathrm{Var}[\mathbf{u}^{\mathsf{T}}\mathbf{Z}_D]/\mathrm{Var}[\mathbf{u}^{\mathsf{T}}\mathbf{Z}_B]$.

Let $\mathbf{u}_* := \mathbf{\Gamma}\mathbf{r}_*/\|\mathbf{\Gamma}\mathbf{r}_*\|_2$, and $g^* := \mathbf{u}_*^{\mathsf{T}}\mathbf{Z}$ which is $N(0,1)$. For $a \in \{0, 1\}$, let $\mathbf{Z}_a$ be $\mathbf{Z}$ conditioned on $\mathsf{pos}\left(\mathbf{u}_*^{\mathsf{T}}\mathbf{Z}\right) = a$. Let $g_a^* := \mathbf{u}_*^{\mathsf{T}}\mathbf{Z}_a$, $a \in \{0, 1\}$, be the half normal distributions satisfying $\mathbb{E}[(g_a^*)^2] = 1$ and $\mathbb{E}[g_a^*] = (-1)^{1-a}\sqrt{2/\pi}$. With this setup, letting $g_B^* := \mathbf{u}_*^{\mathsf{T}}\mathbf{Z}_B$ and $g_D^* := \mathbf{u}_*^{\mathsf{T}}\mathbf{Z}_D$ we obtain (using Lemma B.2 in Appendix B.2)

$$\mathrm{Var}[g_B^*] = 1 - \kappa_1, \quad \mathrm{Var}[g_D^*] = 2(1 - \kappa_1) + \kappa_2$$

Now let $\tilde{\mathbf{u}}$ be a unit vector orthogonal to $\mathbf{u}_*$. Let $\tilde{g} = \tilde{\mathbf{u}}^{\mathsf{T}}\mathbf{Z}$ be $N(0,1)$. Also, let $\tilde{g}_a = \tilde{\mathbf{u}}^{\mathsf{T}}\mathbf{Z}_a$ for $a \in \{0, 1\}$. Since $\mathbf{Z}_a$ are given by conditioning $\mathbf{Z}$ only along $\mathbf{u}_*$, $\tilde{g}_a \sim N(0, 1)$ for $a \in \{0, 1\}$. In particular, the component along $\tilde{u}$ of $\mathbf{Z}_B$ (call it $\tilde{g}_B$) is $N(0, 1)$ and that of $\mathbf{Z}_D$ (call it $\tilde{g}_D$) is the difference of two iid $N(0, 1)$ variables. Thus, $\mathrm{Var}[\tilde{g}_B] = 1$ and $\mathrm{Var}[\tilde{g}_D] = 2$. Moreover, due to orthogonality all these gaussian variables corresponding to $\tilde{\mathbf{u}}$ are independent of those corresponding to $\mathbf{u}_*$ defined earlier. Now let $\mathbf{u} = \alpha\mathbf{u}_* + \beta\tilde{\mathbf{u}}$, where $\beta = \sqrt{1 - \alpha^2}$ be any unit vector. From the above we have,

$$\frac{\mathrm{Var}\left[\mathbf{u}^{\mathsf{T}}\mathbf{Z}_D\right]}{\mathrm{Var}\left[\mathbf{u}^{\mathsf{T}}\mathbf{Z}_B\right]} = \frac{\mathrm{Var}\left[\alpha g_D^* + \beta\tilde{g}_D\right]}{\mathrm{Var}\left[\alpha g_B^* + \beta\tilde{g}_B\right]} = \frac{\alpha^2\,\mathrm{Var}\left[g_D^*\right] + \beta^2\,\mathrm{Var}\left[\tilde{g}_D\right]}{\alpha^2\,\mathrm{Var}\left[g_B^*\right] + \beta^2\,\mathrm{Var}\left[\tilde{g}_B\right]} = \frac{2\alpha^2(1 - \kappa_1) + \alpha^2\kappa_2 + 2\beta^2}{\alpha^2(1 - \kappa_1) + \beta^2}$$

$$= 2 + \frac{\alpha^2\kappa_2}{1 - \alpha^2\kappa_1} \qquad\qquad (6)$$

where the last equality uses $\beta = \sqrt{1 - \alpha^2}$. Letting $\mathbf{u} = \mathbf{\Gamma}\mathbf{r}/\|\mathbf{\Gamma}\mathbf{r}\|_2$ we obtain that $\alpha = \frac{\langle\mathbf{\Gamma}\mathbf{r}, \mathbf{\Gamma}\mathbf{r}_*\rangle}{\|\mathbf{\Gamma}\mathbf{r}\|_2\|\mathbf{\Gamma}\mathbf{r}_*\|_2} = \gamma(\mathbf{r})$ completing the proof. $\qquad\square$

**Lemma 4.5.** $\underset{\|\mathbf{r}\|_2=1}{\mathrm{argmax}}\rho(\mathbf{r}) = \mathbf{\Sigma}_B^{-1/2}\mathsf{PrincipalEigenVector}(\mathbf{\Sigma}_B^{-1/2}\mathbf{\Sigma}_D\mathbf{\Sigma}_B^{-1/2})$

*Proof.* This follows directly from its generalization in Appendix B.5. $\qquad\square$

We now complete the proof of Lemma 4.1 (with $\delta$ instead of $\delta/2$ for convenience). By Lemma 3.1, taking $m \geq O\left((d/\varepsilon_1^2)\log(d/\delta)\right)$ ensures that $\|\mathbf{E}_B\|_2 \leq \varepsilon_1\lambda_{\max}$ and $\|\mathbf{E}_D\|_2 \leq \varepsilon_1\lambda_{\max}$ w.p. at least $1-\delta$ where $\mathbf{E}_B = \hat{\mathbf{\Sigma}}_B - \mathbf{\Sigma}_B$ and $\mathbf{E}_D = \hat{\mathbf{\Sigma}}_D - \mathbf{\Sigma}_D$. We start by defining $\hat{\rho(\mathbf{r})} := \frac{\mathbf{r}^\mathsf{T}\hat{\mathbf{\Sigma}}_D\mathbf{r}}{\mathbf{r}^\mathsf{T}\hat{\mathbf{\Sigma}}_B\mathbf{r}}$ which is the equivalent of $\rho$ using the estimated matrices. Observe that it can be written as $\hat{\rho}(\mathbf{r}) = \frac{\mathbf{r}^\mathsf{T}\mathbf{\Sigma}_B\mathbf{r}+\mathbf{r}^\mathsf{T}\mathbf{E}_B\mathbf{r}}{\mathbf{r}^\mathsf{T}\mathbf{\Sigma}_D\mathbf{r}+\mathbf{r}^\mathsf{T}\mathbf{E}_D\mathbf{r}}$. Using these we can obtain the following bound on $\hat{\rho}$: for any $\mathbf{r} \in \mathbb{R}^d$, $|\hat{\rho}(\mathbf{r}) - \rho(\mathbf{r})| \leq \theta\varepsilon_1|\rho(\mathbf{r})|$ w.p. at least $1 - \delta$ (*) as long as $\varepsilon_1 \leq \frac{(1-\kappa_1)}{2}\frac{\lambda_{\min}}{\lambda_{\max}}$, which we shall ensure (see Appendix B.4).

For convenience we denote the normalized projection of any vector $\mathbf{r}$ as $\tilde{\mathbf{r}} := \frac{\mathbf{\Sigma}^{1/2}\mathbf{r}}{\|\mathbf{\Sigma}^{1/2}\mathbf{r}\|_2}$. Now let $\tilde{\mathbf{r}} \in \mathbb{R}^d$ be a unit vector such that $\min\{\|\tilde{\mathbf{r}} - \tilde{\mathbf{r}}_*\|_2, \|\tilde{\mathbf{r}} + \tilde{\mathbf{r}}_*\|_2\} \geq \varepsilon_2$. Hence, using the definitions from Lemma 4.4, $|\gamma(\mathbf{r})| \leq 1 - \varepsilon_2^2/2$ while $\gamma(\mathbf{r}_*) = 1$ which implies $\rho(\mathbf{r}_*) - \rho(\mathbf{r}) \geq \kappa_3\varepsilon_2^2/2$. Note that $\rho(\mathbf{r}) \leq \rho(\mathbf{r}_*) = 2 + \kappa_3$. Choosing $\varepsilon_1 < \frac{\kappa_3}{4\theta(2+\kappa_3)}\varepsilon_2^2$, we obtain that $\rho(\mathbf{r}_*)(1 - \theta\varepsilon_1) > \rho(\mathbf{r})(1 + \theta\varepsilon_1)$. Using this along with the bound (*) we obtain that w.p. at least $1 - \delta$, $\hat{\rho}(\mathbf{r}_*) > \hat{\rho}(\mathbf{r})$ when $\varepsilon_2 > 0$. Since our algorithm returns $\hat{\mathbf{r}}$ as the maximizer of $\hat{\rho}$, w.p. at least $1 - \delta$ we get $\min\{\|\tilde{\mathbf{r}} - \tilde{\mathbf{r}}_*\|_2, \|\tilde{\mathbf{r}} + \tilde{\mathbf{r}}_*\|_2\} \leq \varepsilon_2$. Using Lemma 2.1, $\min\{\|\hat{\mathbf{r}} - \mathbf{r}_*\|_2, \|\hat{\mathbf{r}} + \mathbf{r}_*\|_2\} \leq 4\sqrt{\frac{\lambda_{\max}}{\lambda_{\min}}}\varepsilon_2$. Substituting $\varepsilon_2 = \frac{\varepsilon}{4}\sqrt{\frac{\lambda_{\min}}{\lambda_{\max}}}$, $\|\mathbf{r} - \mathbf{r}_*\|_2 \leq \varepsilon$ w.p. at least $1 - \delta$. The conditions on $\varepsilon_1$ are satisfied by taking it to be $\leq O\left(\frac{\kappa_3\varepsilon^2\lambda_{\min}}{\theta(2+\kappa_3)\lambda_{\max}}\right)$, and thus we can take $m \geq O\left((d/\varepsilon^4)\log(d/\delta)\left(\frac{\lambda_{\max}}{\lambda_{\min}}\right)^2\theta^2\left(\frac{2+\kappa_3}{\kappa_3}\right)^2\right) = O\left((d/\varepsilon^4)\log(d/\delta)\left(\frac{\lambda_{\max}}{\lambda_{\min}}\right)^4 q^4\right)$, using Defn. 4.3. This completes the proof.

## 5 Proof Sketches for Theorems 1.3 and 1.5

**Theorem 1.3: Case** $N(\mathbf{0}, \mathbf{I})$, $f(\mathbf{x}) = \mathsf{pos}(\mathbf{r}_*^\mathsf{T}\mathbf{x})$, $k \neq q/2$. The algorithm (Algorithm 3) and the proof is in Appendix A. We argue that a vector sampled uniformly at random from a bag is distributed as $\omega\mathbf{X}_1 + (1 - \omega)\mathbf{X}_0$ where $\mathbf{X}_a \sim N(\mathbf{0}, \mathbf{I})$ conditioned on $f(\mathbf{X}_a) = a$ and $\omega$ is an independent $\{0, 1\}-$Bernoulli r.v. s.t. $p(\omega = 1) = k/q$. This along with the fact that uncorrelated Gaussians are independent, allows us to show that the expectation is $\mathbf{0}$ in any direction orthogonal to $\mathbf{r}_*$ and to compute the expectation in the direction of $\mathbf{r}_*$. We then use Lemma 3.1 to get the sample complexity expression.

**Theorem 1.5: Case** $N(\boldsymbol{\mu}, \mathbf{\Sigma})$, $f(\mathbf{x}) = \mathsf{pos}(\mathbf{r}_*^\mathsf{T}\mathbf{x} + c_*)$. The algorithm (Algorithm 4) and the detailed proof is given in Appendix C. We start by generalizing the high probability geometric error bound in Lemma 4.1 to this case (Lemma C.1 proven in Appendix C.1) and appropriately generalize $\kappa_1, \kappa_2, \kappa_3$ and $\theta$. The rest of the proof is similar to Section 4.1. An extra factor of $O(\ell^2)$ is introduced to the sample complexity from Lemma 3.1. Next, assuming the geometric bound, we give a high probability bound on the generalization error in Lemma C.2. In this analysis, we bound the $\mathsf{BagErr}_{\mathsf{sample}}(h, \mathcal{M})$ where $h(\mathbf{x}) = \mathsf{pos}(\hat{\mathbf{r}}^\mathsf{T}\mathbf{x} + c_*)$ with $\hat{\mathbf{r}}$ being the geometric estimate of $\mathbf{r}_*$ and $\mathcal{M}$ is a sample of bags. This introducing the dependencies on $(\sqrt{\lambda_{\max}} + \|\boldsymbol{\mu}\|_2)/\sqrt{\lambda_{\min}}$ as well as on $\Phi(\ell)$. Our subroutine to find $\hat{c}$ ensures that $h^*(\mathbf{x}) = \mathsf{pos}(\hat{\mathbf{r}}^\mathsf{T}\mathbf{x} + \hat{c})$ satisfies $\mathsf{BagErr}_{\mathsf{sample}}(h^*, \mathcal{M}) \geq \mathsf{BagErr}_{\mathsf{sample}}(h, \mathcal{M})$. We then use Theorem 2.2 to bound the generalization error of $h^*$. Lemmas C.1 C.2 together imply Theorem 1.5.

## 6 Experimental Results

*General Gaussian.* We empirically evaluate our algorithmic technique on centered and general Gaussian distributions for learning homogeneous LTFs. For homogeneous LTFs the general case algorithm (Alg. 4 in Appendix C) boils down to Alg. 2 in Sec. 4. The experimental LLP datasets are created using samples from both balanced as well as unbalanced bag oracles. In particular, for dimension $d \in \{10, 50\}$, and each pair $(q, k) \in \{(2, 1), (3, 1), (10, 5), (10, 8), (50, 25), (50, 35)\}$

and $m =$ we create 25 datasets as follows: for each dataset (i) sample a random unit vector $\mathbf{r}^*$ and let $f(\mathbf{x}) := \mathsf{pos}\left(\mathbf{r}^{*\mathsf{T}}\mathbf{x}\right)$, (ii) sample $\boldsymbol{\mu}$ and $\boldsymbol{\Sigma}$ randomly (see Appendix F for details), (iii) sample $m = 2000$ training bags from $\mathsf{Ex}(f, N(\boldsymbol{\mu}, \boldsymbol{\Sigma}), q, k)$, (iv) sample 1000 test instances $(\mathbf{x}, f(\mathbf{x}))$, $\mathbf{x} \leftarrow N(\boldsymbol{\mu}, \boldsymbol{\Sigma})$. We fix $\boldsymbol{\mu} = \mathbf{0}$ for the centered Gaussian case.

For comparison we include the random LTF algorithm in which we sample 100 random LTFs and return the one that satisfies the most bags. In addition, we evaluate the Algorithm of [25] on $(q, k) = (2, 1)$, and the Algorithm of [26] on $(q, k) = (3, 1)$. We measure the accuracy of each method on the test set of each dataset. The algorithms of [25, 26] are considerably slower and we use 200 training bags for them. The results for centered Gaussian are in Table 1a and for the general Gaussian are in Table 1b. We observe that our algorithms perform significantly better in terms of accuracy than the comparative methods in all the bag distribution settings. Further, our algorithms have significantly lower error bounds (see Appendix F).

Notice in Tables 1a and 1b that the test accuracy for Algorithm 2 decreases with an increase in $q$ and $d$. This is consistent with the sample complexity expressions in Thm. 1.4 and Thm. 1.5. Also, notice that the test accuracy for Algorithm 2 for general Gaussian (Table 1b) is usually lesser than the same for centered Gaussian (Table 1a). This supports the theoretical result that the sample complexity increases with the increase in $l$.

Appendix F has additional details and further experiments for the $N(\mathbf{0}, \mathbf{I})$ with homogeneous LTFs, $N(\boldsymbol{\mu}, \boldsymbol{\Sigma})$ with non-homogeneous LTFs as well as on noisy label distributions.

Table 1: Algorithm A2 vs. rand. LTF (R) vs SDP algorithms (S)

| $d$ | $q$ | $k$ | A2 | R | S | $d$ | $q$ | $k$ | A2 | R | S |
|---|---|---|---|---|---|---|---|---|---|---|---|
| 10 | 2 | 1 | 98.12 | 78.26 | 88.40 | 10 | 2 | 1 | 98.18 | 78.32 | 90.10 |
| 10 | 3 | 1 | 98.27 | 77.16 | 67.31 | 10 | 3 | 1 | 97.92 | 75.14 | 70.80 |
| 10 | 10 | 5 | 97.9 | 78.66 | - | 10 | 10 | 5 | 97.86 | 70.41 | - |
| 10 | 10 | 8 | 97.87 | 77.64 | - | 10 | 10 | 8 | 97.4 | 69.86 | - |
| 10 | 50 | 25 | 97.87 | 76.67 | - | 10 | 50 | 25 | 97.57 | 70.48 | - |
| 10 | 50 | 35 | 97.9 | 77.17 | - | 10 | 50 | 35 | 97.6 | 62.86 | - |
| 50 | 2 | 1 | 95.64 | 61.25 | 57.83 | 50 | 2 | 1 | 94.99 | 58.68 | 61.12 |
| 50 | 3 | 1 | 95.21 | 61.15 | 58.69 | 50 | 3 | 1 | 95.6 | 59.8 | 62.39 |
| 50 | 10 | 5 | 95.59 | 55.06 | - | 50 | 10 | 5 | 95.27 | 57.43 | - |
| 50 | 10 | 8 | 94.34 | 63.17 | - | 50 | 10 | 8 | 94.44 | 61.82 | - |
| 50 | 50 | 25 | 95.16 | 55.76 | - | 50 | 50 | 25 | 94.97 | 53.98 | - |
| 50 | 50 | 35 | 94.74 | 61.02 | - | 50 | 50 | 35 | 94.33 | 56.97 | - |

(a) $N(\mathbf{0}, \boldsymbol{\Sigma})$ feature-vectors.        (b) $N(\boldsymbol{\mu}, \boldsymbol{\Sigma})$ feature-vectors.

## 7 Conclusion and Future work

Our work shows that LTFs can be efficiently properly learnt in the LLP setting from random bags with given label proportion whose feature-vectors are sampled independently from a Gaussian space, conditioned on their underlying labels. For the simple case of $N(\mathbf{0}, \mathbf{I})$ distribution and bags with unbalanced labels we provide a mean estimation based algorithm. For the general scenarios we develop a more sophisticated approach using the principal component of a matrix formed from certain covariance matrices. To resolve the ambiguity between the obtained solutions we employ novel generalization error bounds from bag satisfaction to instance classification. We also show that subgaussian concentration bounds are applicable on the thresholded Gaussians, yielding efficient sample complexity bounds. Our experimental results validate the performance guarantees of our algorithmic techniques.

In future work, classes of distributions other than Gaussian could be similarly investigated. Classifiers other than LTFs are also interesting to study in the LLP setting.

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

# A Proof of Theorem 1.3

For the setting of Theorem 1.3 we provide Algorithm 3.

---

**Algorithm 3** PAC Learner for homogenous LTFs from unbalanced bags over $N(\mathbf{0}, \mathbf{I})$

---

**Input:** $\mathsf{Ex}(f, \mathcal{D}, q, k), m$, where $f(\mathbf{x}) = \mathsf{pos}\left(\mathbf{r}_*^\mathsf{T}\mathbf{x}\right)$, $\|\mathbf{r}_*\|_2 = 1$, $k \neq q/2$.
1. Compute $\hat{\boldsymbol{\mu}}_B$ using MeanCovsEstimator with $m$ samples.
2. If $k > q/2$ **Return:** $\hat{\boldsymbol{\mu}}_B/\|\hat{\boldsymbol{\mu}}_B\|_2$ else **Return:** $-\hat{\boldsymbol{\mu}}_B/\|\hat{\boldsymbol{\mu}}_B\|_2$

---

Define $\mathbf{X}_a \sim N(\mathbf{0}, \mathbf{I})$ conditioned on $f(\mathbf{X}_a) = a$ for $a \in \{0, 1\}$, and let $\mathbf{X}_B$ denotes the random feature-vector u.a.r sampled from a randomly sampled bag. By the definition of the bag oracle, $\mathbf{X}_B := \omega \mathbf{X}_1 + (1 - \omega)\mathbf{X}_0$, where $\omega$ is an independent $\{0, 1\}$-Bernoulli r.v. s.t. $p(\omega = 1) = k/q$. Let $g^* := \mathbf{r}_*^\mathsf{T}\mathbf{X}_B$. $g^* \sim N(0, 1)$ and let $g_a^* = \mathbf{r}_*^\mathsf{T}\mathbf{X}_a$, $a \in \{0, 1\}$. Since $\mathbf{r}_*$ is a unit vector, $g_a^*$ is a half-gaussian and by direct integration we obtain $\mathbb{E}\left[g_a^*\right] = (-1)^{1-a}\sqrt{2/\pi}$ for $a \in \{0, 1\}$. Thus,

$$\mathbb{E}[g^*] = \frac{k}{q}\mathbb{E}[g_1^*] + \left(1 - \frac{k}{q}\right)\mathbb{E}[g_0^*] = \eta(q, k) := \left(\frac{2k}{q} - 1\right)\sqrt{\frac{2}{\pi}}, 0 \leq \eta(q, k) \leq 1$$

On the other hand, let $g^\perp = \mathbf{r}^\mathsf{T}\mathbf{X}$ s.t. $\mathbf{r}^\mathsf{T}\mathbf{r}_* = 0$ and $\|\mathbf{r}\|_2 = 1$, and let $g_a^\perp = \mathbf{r}^\mathsf{T}\mathbf{X}_a$, $a \in \{0, 1\}$. Since $\mathbf{X}_a$ ($a \in \{0, 1\}$) is given by conditioning a standard Gaussian vector only on the component in the direction of $\mathbf{r}_*$, its component along any direction orthogonal to $\mathbf{r}_*$ is a one-dimensional standard Gaussian. Therefore, $g_a^\perp$ are iid $N(0, 1)$ ($a \in \{0, 1\}$), and so is $g^\perp$, implying $\mathbb{E}\left[g^\perp\right] = 0$. Thus, the value of $\mathbf{r}^\mathsf{T}\mathbb{E}\left[\mathbf{X}_B\right]$ is (i) $\eta(q, k)$ if $\mathbf{r} = \mathbf{r}_*$, and (ii) 0 if $\mathbf{r} \perp \mathbf{r}_*$. In other words, $\boldsymbol{\mu}_B = \mathbb{E}\left[\mathbf{X}_B\right] = \eta(q, k)\mathbf{r}_*$, and $\|\boldsymbol{\mu}_B\|_2 = |\eta(q, k)|$. Hence, if $\eta(q, k) > 0$ then $\mathbf{r}_* = \boldsymbol{\mu}_B/\|\boldsymbol{\mu}_B\|_2$ else $\mathbf{r}_* = -\boldsymbol{\mu}_B/\|\boldsymbol{\mu}_B\|_2$

With $m = O\left(\eta(q, k)^2(d/\varepsilon^2)\log(d/\delta)\right) = O\left((d/\varepsilon^2)\log(d/\delta)\right)$, the following lemma along with Lemma 2.3 completes the proof of Theorem 1.3.

**Lemma A.1.** *Algorithm 3 returns a normal vector $\hat{\mathbf{r}}$ such that $\|\mathbf{r}_* - \hat{\mathbf{r}}\|_2 \leq \varepsilon$ w.p. at least $1 - \delta$ when $m \geq O\left((d/\varepsilon^2)\log(d/\delta)\right)$ for $\varepsilon, \delta > 0$.*

*Proof.* First, we can assume $\varepsilon \leq 2$ since the distance between two unit vectors is at most 2 and therefore the lemma is trivially true for $\varepsilon > 2$. By Lemma 3.1, taking $m \geq= O\left((d/\varepsilon^2)\log(d/\delta)\right) = O\left(\eta(q, k)^2(d/\varepsilon^2)\log(d/\delta)\right)$ ensures $\|\hat{\boldsymbol{\mu}}_B - \boldsymbol{\mu}_B\|_2 \leq \varepsilon|\eta(q, k)|/4 = \varepsilon\|\boldsymbol{\mu}_B\|_2/4$ w.p. $1 - \delta$. Therefore, by triangle inequality, $|\|\hat{\boldsymbol{\mu}}_B\|_2 - \|\boldsymbol{\mu}_B\|_2| \leq \varepsilon\|\boldsymbol{\mu}_B\|_2/4 \Rightarrow \|\hat{\boldsymbol{\mu}}_B\|_2/\|\boldsymbol{\mu}_B\|_2 \in [1 - \frac{\varepsilon}{4}, 1 + \frac{\varepsilon}{4}]$. Now, $\mathbf{r}_* = \mathsf{sign}(\eta(q, k))\boldsymbol{\mu}_B/\|\boldsymbol{\mu}_B\|_2$ and the algorithm returns $\hat{\mathbf{r}} := \mathsf{sign}(\eta(q, k))\hat{\boldsymbol{\mu}}_B/\|\hat{\boldsymbol{\mu}}_B\|_2$.

$$
\begin{aligned}
\|\hat{\mathbf{r}} - \mathbf{r}_*\|_2 = \left\|\frac{\hat{\boldsymbol{\mu}}_B}{\|\hat{\boldsymbol{\mu}}_B\|_2} - \frac{\boldsymbol{\mu}_B}{\|\boldsymbol{\mu}_B\|_2}\right\|_2 &\leq \frac{\left\|\hat{\boldsymbol{\mu}}_B\|\boldsymbol{\mu}_B\|_2 - \boldsymbol{\mu}_B\|\boldsymbol{\mu}_B\|_2 + \boldsymbol{\mu}_B\|\boldsymbol{\mu}_B\|_2 - \boldsymbol{\mu}_B\|\hat{\boldsymbol{\mu}}_B\|_2\right\|_2}{\|\hat{\boldsymbol{\mu}}_B\|_2\|\boldsymbol{\mu}_B\|_2} \\
&\leq \frac{\|\hat{\boldsymbol{\mu}}_B - \boldsymbol{\mu}_B\|_2\|\boldsymbol{\mu}_B\|_2 + \|\boldsymbol{\mu}_B\|_2|\|\boldsymbol{\mu}_B\|_2 - \|\hat{\boldsymbol{\mu}}_B\|_2|}{\|\hat{\boldsymbol{\mu}}_B\|_2\|\boldsymbol{\mu}_B\|_2} \\
&\leq \frac{\frac{\varepsilon}{2}\|\boldsymbol{\mu}_B\|_2}{\|\boldsymbol{\mu}_B\|_2 - \frac{\varepsilon}{4}\|\boldsymbol{\mu}_B\|_2} \leq \frac{2\varepsilon}{4 - \varepsilon} \leq \varepsilon \quad \text{for } \varepsilon \leq 2.
\end{aligned}
$$

$\square$

# B Useful Tools

## B.1 Chernoff Bound

We state the well known Chernoff Bound.

**Theorem B.1.** *Let $X_1, \ldots, X_n$ be iid $\{0, 1\}$-valued random variables. Let $S$ be their sum and $\mu = \mathbb{E}[S]$. Then, for any $\delta > 0$,*

$$\Pr[S \geq (1 + \delta)\mu] \leq \exp(-\delta^2\mu/(2 + \delta)).$$

## B.2 Relationship between $\mathbf{\Sigma}_B$ and $\mathbf{\Sigma}_D$

**Lemma B.2.** *If $\mathbf{\Sigma}_B, \mathbf{\Sigma}_D$ are as defined in the preliminaries and $\mathbf{X}_a := \mathbf{X}|f(\mathbf{X}) = a$, then*

$$\mathbf{\Sigma}_D = 2\mathbf{\Sigma}_B + \frac{2}{q-1}\left(\frac{k}{q}\right)\left(1 - \frac{k}{q}\right)(\mathbb{E}[\mathbf{X}_1] - \mathbb{E}[\mathbf{X}_0])(\mathbb{E}[\mathbf{X}_1] - \mathbb{E}[\mathbf{X}_0])^T$$

*Proof.* Let $\mathbf{X}_B$ be a random-feature vector sampled uniformly from a random bag sampled from $\mathcal{O}$. Hence, with probability $k/q$ it is sampled from $\mathbf{X}_1$ and with a probability of $1 - k/q$ it is sampled from $\mathbf{X}_0$. Hence,

$$\boldsymbol{\mu}_B = \mathbb{E}[\mathbf{X}_B] = \frac{k}{q}\mathbb{E}[\mathbf{X}_1] + \left(1 - \frac{k}{q}\right)\mathbb{E}[\mathbf{X}_0]$$

$$\mathbf{\Sigma}_B = \mathbb{E}[\mathbf{X}_B\mathbf{X}_B^T] - \mathbb{E}[\mathbf{X}_B]\mathbb{E}[\mathbf{X}_B]^T \text{ where } \mathbb{E}[\mathbf{X}\mathbf{X}^T] = \frac{k}{q}\mathbb{E}[\mathbf{X}_1\mathbf{X}_1^T] + \left(1 - \frac{k}{q}\right)\mathbb{E}[\mathbf{X}_0\mathbf{X}_0^T]$$

Let $\mathbf{X}_D = \mathbf{X} - \mathbf{X}'$ where $(\mathbf{X}, \mathbf{X}')$ are a random pair of feature-vectors sampled (without replacement) from a random bag sampled from $\mathcal{O}$. Hence, with probability $\binom{k}{2}/\binom{q}{2}$ it is the difference of two vectors sampled independently from $\mathbf{X}_1$, with probability $\binom{q-k}{2}/\binom{q}{2}$ it is the difference of two vectors sampled independently from $\mathbf{X}_0$, with probability $k(q-k)/2\binom{q}{2}$, it is the difference of one vector sampled from $X_1$ and another sampled independently from $X_0$ and with probability $k(q-k)/2\binom{q}{2}$, it is the difference of one vector sampled from $X_0$ and another sampled independently from $X_1$. Then,

$$\mathbf{\Sigma}_D = \mathbb{E}[\mathbf{X}_D\mathbf{X}_D^T] = \frac{1}{\binom{q}{2}}\left[\binom{k}{2}\mathbb{E}[(\mathbf{X}_1 - \mathbf{X}_1')(\mathbf{X}_1 - \mathbf{X}_1')^T] + \binom{q-k}{2}\mathbb{E}[(\mathbf{X}_0 - \mathbf{X}_0')(\mathbf{X}_0 - \mathbf{X}_0')^T]\right.$$

$$\left. + \frac{k(q-k)}{2}\mathbb{E}[(\mathbf{X}_1 - \mathbf{X}_0')(\mathbf{X}_1 - \mathbf{X}_0')^T] + \frac{k(q-k)}{2}\mathbb{E}[(\mathbf{X}_0 - \mathbf{X}_1')(\mathbf{X}_0 - \mathbf{X}_1')^T]\right]$$

Due to independence, we obtain for $a \in \{0, 1\}$

$$\mathbb{E}[(\mathbf{X}_a - \mathbf{X}_a')(\mathbf{X}_a - \mathbf{X}_a')^T] = 2\mathbb{E}[\mathbf{X}_a\mathbf{X}_a^T] - 2\mathbb{E}[\mathbf{X}_a]\mathbb{E}[\mathbf{X}_a]^T$$

$$\mathbb{E}[(\mathbf{X}_a - \mathbf{X}_{1-a}')(\mathbf{X}_a - \mathbf{X}_{1-a}')^T] = \mathbb{E}[\mathbf{X}_1\mathbf{X}_1^T] + \mathbb{E}[\mathbf{X}_0\mathbf{X}_0^T] - \mathbb{E}[\mathbf{X}_1]\mathbb{E}[\mathbf{X}_0]^T - \mathbb{E}[\mathbf{X}_0]\mathbb{E}[\mathbf{X}_1]^T$$

Hence,

$$\mathbf{\Sigma}_D = \frac{1}{\binom{q}{2}}\left[\left(2\binom{k}{2} + k(q-k)\right)\mathbb{E}[\mathbf{X}_1\mathbf{X}_1^T] + \left(2\binom{q-k}{2} + k(q-k)\right)\mathbb{E}[\mathbf{X}_0\mathbf{X}_0^T]\right.$$

$$- \left(k(k-1)\mathbb{E}[\mathbf{X}_1]\mathbb{E}[\mathbf{X}_1]^T + (q-k)(q-k-1)\mathbb{E}[\mathbf{X}_0]\mathbb{E}[\mathbf{X}_0]^T\right)$$

$$\left. + \left(k(q-k)\left(\mathbb{E}[\mathbf{X}_1]\mathbb{E}[\mathbf{X}_0]^T + \mathbb{E}[\mathbf{X}_0]\mathbb{E}[\mathbf{X}_1]^T\right)\right)\right]$$

Simplifying $\mathbf{\Sigma}_D - 2\mathbf{\Sigma}_B$, we get

$$\mathbf{\Sigma}_D - 2\mathbf{\Sigma}_B = \frac{2}{q-1}\left(\frac{k}{q}\right)\left(1 - \frac{k}{q}\right)\left[(\mathbb{E}[\mathbf{X}_1] - \mathbb{E}[\mathbf{X}_0])(\mathbb{E}[\mathbf{X}_1] - \mathbb{E}[\mathbf{X}_0])^T\right]$$

$\square$

## B.3 Bound on $\gamma(\mathbf{r})$ when $\mathbf{r}^T\mathbf{\Sigma}_B\mathbf{r}_* = 0$

**Lemma B.3.** *If $\mathbf{r}^T\mathbf{\Sigma}_B\mathbf{r}_* = 0$ then $\gamma(\mathbf{r}) \leq 1 - \left(\frac{\lambda_{\min}}{\lambda_{\max}}\right)^2 \frac{1 - \max(0, \kappa_1(q,k,\ell))}{1 - \min(0, \kappa_1(q,k,\ell))}$. Further if $\ell = 0$, then $|\gamma(\mathbf{r})| \leq 1 - \left(\frac{\lambda_{\min}}{\lambda_{\max}}\right)^2 (1 - \kappa_1(q,k))$.*

*Proof.* We begin by observing the for any $\mathbf{r} \in \mathbb{R}^d$, $\|\mathbf{\Sigma}^{1/2}\mathbf{r}\|_2 = 1$, $\text{Var}[\mathbf{r}^T\mathbf{Z}_B] = 1 - \kappa_1(q,k,\ell)$. Hence, $\lambda_{\min}(\mathbf{\Sigma}_B) \geq (1 - \max(0, \kappa_1(q,k,\ell)))\lambda_{\min}$ and $\lambda_{\max}(\mathbf{\Sigma}_B) \leq (1 - \min(0, \kappa_1(q,k,\ell)))\lambda_{\max}$.

Thus, by Cautchy-Schwartz inequality,

$$\|\mathbf{\Sigma}_B^{1/2}\mathbf{\Sigma}^{-1/2}\|_2 \le \sqrt{1 - \min(0, \kappa_1(q, k, \ell))}\sqrt{\frac{\lambda_{\max}}{\lambda_{\min}}}$$

$$\|\mathbf{\Sigma}^{1/2}\mathbf{\Sigma}_B^{-1/2}\|_2 \le \frac{1}{\sqrt{1 - \max(0, \kappa_1(q, k, \ell))}}\sqrt{\frac{\lambda_{\max}}{\lambda_{\min}}}$$

Define $\tilde{\mathbf{r}} = \frac{\mathbf{\Sigma}^{1/2}\mathbf{r}}{\|\mathbf{\Sigma}^{1/2}\mathbf{r}\|_2}$ for any $\mathbf{r} \in \mathbb{R}^d$. Observe that $\gamma(\mathbf{r}) = \tilde{\mathbf{r}}^T \tilde{\mathbf{r}}_*$. Now since $\mathbf{r}^T \mathbf{\Sigma}_B \mathbf{r}_* = 0$, by substitution $(\mathbf{\Sigma}_B^{1/2}\mathbf{\Sigma}^{-1/2}\tilde{\mathbf{r}})^T(\mathbf{\Sigma}_B^{1/2}\mathbf{\Sigma}^{-1/2}\tilde{\mathbf{r}}_*) = 0$. Thus, using Cautchy-Schwartz again

$$\|\mathbf{\Sigma}_B^{1/2}\mathbf{\Sigma}^{-1/2}\tilde{\mathbf{r}} - \mathbf{\Sigma}_B^{1/2}\mathbf{\Sigma}^{-1/2}\tilde{\mathbf{r}}_*\|_2 = \sqrt{\|\mathbf{\Sigma}_B^{1/2}\mathbf{\Sigma}^{-1/2}\tilde{\mathbf{r}}\|_2^2 + \|\mathbf{\Sigma}_B^{1/2}\mathbf{\Sigma}^{-1/2}\tilde{\mathbf{r}}_*\|_2^2}$$

$$\ge \frac{\sqrt{\|\tilde{\mathbf{r}}\|_2^2 + \|\tilde{\mathbf{r}}_*\|_2^2}}{\|\mathbf{\Sigma}^{1/2}\mathbf{\Sigma}_B^{-1/2}\|_2} \ge \sqrt{2(1 - \max(0, \kappa_1(q, k, \ell)))}\sqrt{\frac{\lambda_{\min}}{\lambda_{\max}}}$$

Again using Cautchy-Schwartz,

$$\|\tilde{\mathbf{r}} - \tilde{\mathbf{r}}_*\|_2 \ge \frac{\|\mathbf{\Sigma}_B^{1/2}\mathbf{\Sigma}^{-1/2}\tilde{\mathbf{r}} - \mathbf{\Sigma}_B^{1/2}\mathbf{\Sigma}^{-1/2}\tilde{\mathbf{r}}_*\|_2}{\|\mathbf{\Sigma}_B^{1/2}\mathbf{\Sigma}^{-1/2}\|_2}$$

$$\ge \frac{\lambda_{\min}}{\lambda_{\max}}\sqrt{\frac{2(1 - \max(0, \kappa_1(q, k, \ell)))}{1 - \min(0, \kappa_1(q, k, \ell))}}$$

Thus,

$$\gamma(\mathbf{r}) = \tilde{\mathbf{r}}^T \tilde{\mathbf{r}}_* \le 1 - \left(\frac{\lambda_{\min}}{\lambda_{\max}}\right)^2 \frac{1 - \max(0, \kappa_1(q, k, \ell))}{1 - \min(0, \kappa_1(q, k, \ell))}$$

If $\ell = 0$, then $\kappa_1(q, k, 0) = \kappa_1(q, k) \ge 0$. Hence,

$$\gamma(\mathbf{r}) \le 1 - \left(\frac{\lambda_{\min}}{\lambda_{\max}}\right)^2 (1 - \kappa_1(q, k))$$

$\square$

## B.4 Bounding error in $\hat{\rho}$

**Lemma B.4.** *If $\|\mathbf{E}_B\|_2 \le \varepsilon\lambda_{\max}$ and $\|\mathbf{E}_D\|_2 \le \varepsilon\lambda_{\max}$ where $\mathbf{E}_B = \hat{\mathbf{\Sigma}}_B - \hat{\mathbf{\Sigma}}_B$ and $\mathbf{E}_D = \hat{\mathbf{\Sigma}}_D - \hat{\mathbf{\Sigma}}_D$ then,*

$$|\hat{(\rho(\mathbf{r}))} - \rho(\mathbf{r})| \le |\rho(\mathbf{r})|\theta\varepsilon$$

*for $\theta$ as defined in Defn. 4.3 or Defn. C.3.*

*Proof.* Let $a(\mathbf{r}) := \mathbf{r}^\mathsf{T}\mathbf{\Sigma}_D\mathbf{r}, b(\mathbf{r}) := \mathbf{r}^\mathsf{T}\mathbf{\Sigma}_B\mathbf{r}, e(\mathbf{r}) := \mathbf{r}^\mathsf{T}\mathbf{E}_D\mathbf{r}, f(\mathbf{r}) := \mathbf{r}^\mathsf{T}\mathbf{E}_B\mathbf{r}$. Using the bounds on the spectral norms of $\mathbf{E}_B$ and $\mathbf{E}_D$, we get that $|e(\mathbf{r})| \le \varepsilon\lambda_{\max}$ and $|f(\mathbf{r})| \le \varepsilon\lambda_{\max}$. Also, using variances in Lemma C.4, $a(\mathbf{r}) \ge \lambda_{\min}(2 - \max(0, 2\kappa_1 - \kappa_2)) \ge 0$ and $b(\mathbf{r}) \ge \lambda_{\min}(1 - \max(0, \kappa_1)) \ge 0$. Hence, we can conclude that

$$\left|\frac{e(\mathbf{r})}{a(\mathbf{r})}\right| \le \varepsilon\frac{\lambda_{\max}}{\lambda_{\min}}\left(\frac{1}{2 - \max(0, 2\kappa_1 - \kappa_2)}\right) \text{ and } \left|\frac{f(\mathbf{r})}{b(\mathbf{r})}\right| \le \varepsilon\frac{\lambda_{\max}}{\lambda_{\min}}\left(\frac{1}{1 - \max(0, \kappa_1)}\right)$$

Observe that $\hat{\rho}(\mathbf{r})/\rho(\mathbf{r}) = \frac{1 + e(\mathbf{r})/a(\mathbf{r})}{1 + b(\mathbf{r})/f(\mathbf{r})}$. Hence,

$$\frac{1 - \varepsilon\frac{\lambda_{\max}}{\lambda_{\min}}\left(\frac{1}{2 - \max(0, 2\kappa_1 - \kappa_2)}\right)}{1 + \varepsilon\frac{\lambda_{\max}}{\lambda_{\min}}\left(\frac{1}{1 - \max(0, \kappa_1)}\right)} \le \frac{\hat{\rho}(\mathbf{r})}{\rho(\mathbf{r})} \le \frac{1 + \varepsilon\frac{\lambda_{\max}}{\lambda_{\min}}\left(\frac{1}{2 - \max(0, 2\kappa_1 - \kappa_2)}\right)}{1 - \varepsilon\frac{\lambda_{\max}}{\lambda_{\min}}\left(\frac{1}{1 - \max(0, \kappa_1)}\right)}$$

Now, whenever $\varepsilon \le \frac{1 - \max(0, \kappa_1)}{2}\frac{\lambda_{\min}}{\lambda_{\max}}, 1 - \varepsilon\frac{\lambda_{\max}}{\lambda_{\min}}\left(\frac{1}{1 - \max(0, \kappa_1)}\right) \ge 1/2$ and,

$1 + \varepsilon\frac{\lambda_{\max}}{\lambda_{\min}}\left(\frac{1}{1 - \max(0, \kappa_1)}\right) \le 3/2$. Thus, we obtain that

$$\left|\frac{\hat{\rho}(\mathbf{r}) - \rho(\mathbf{r})}{\rho(\mathbf{r})}\right| \le \theta\varepsilon$$

where $\theta$ is as defined in Defn. C.3. If we substitute $\ell = 0$, we get that $\kappa_1 \ge 0$ and we get $\theta$ as defined in Defn. 4.3. $\square$

## B.5 Ratio maximisation as a PCA problem

**Theorem B.5.** *If* $\mathbf{A}$ *and* $\mathbf{B}$ *are positive definite matrices, then for all* $\mathbf{r} \in \mathbb{R}^d$

1. $\frac{\mathbf{r}^\mathsf{T}\mathbf{A}\mathbf{r}}{\mathbf{r}^\mathsf{T}\mathbf{B}\mathbf{r}} = \tilde{\mathbf{r}}^\mathsf{T}\mathbf{B}^{-1/2}\mathbf{A}\mathbf{B}^{-1/2}\tilde{\mathbf{r}}$, $\tilde{\mathbf{r}} = \frac{\mathbf{B}^{1/2}\mathbf{r}}{\|\mathbf{B}^{1/2}\mathbf{r}\|_2}$

2. $\underset{\|\mathbf{r}\|_2=1}{\operatorname{argmax}} \frac{\mathbf{r}^\mathsf{T}\mathbf{A}\mathbf{r}}{\mathbf{r}^\mathsf{T}\mathbf{B}\mathbf{r}} = \frac{\mathbf{B}^{-1/2}\tilde{\mathbf{r}}^*}{\|\mathbf{B}^{-1/2}\tilde{\mathbf{r}}^*\|_2}$ *where* $\tilde{\mathbf{r}}^* = \underset{\|\tilde{\mathbf{r}}\|_2=1}{\operatorname{argmax}} \tilde{\mathbf{r}}^\mathsf{T}\mathbf{B}^{-1/2}\mathbf{A}\mathbf{B}^{-1/2}\tilde{\mathbf{r}}$

*Proof.* The first statement comes from the substitution. Now, $\frac{\mathbf{r}^\mathsf{T}\mathbf{A}\mathbf{r}}{\mathbf{r}^\mathsf{T}\mathbf{B}\mathbf{r}}$ is homogeneoous in $\mathbf{r}$. Let $a \simeq b \Rightarrow a = kb, k \in \mathbb{R}$

$$
\begin{aligned}
\underset{\|\mathbf{r}\|_2=1}{\operatorname{argmax}} \frac{\mathbf{r}^\mathsf{T}\mathbf{A}\mathbf{r}}{\mathbf{r}^\mathsf{T}\mathbf{B}\mathbf{r}} &\simeq \underset{\mathbf{r}}{\operatorname{argmax}} \frac{\mathbf{r}^\mathsf{T}\mathbf{A}\mathbf{r}}{\mathbf{r}^\mathsf{T}\mathbf{B}\mathbf{r}} \\
&\simeq \underset{\mathbf{B}^{-1/2}\tilde{\mathbf{r}}}{\operatorname{argmax}} \frac{\tilde{\mathbf{r}}^\mathsf{T}\mathbf{B}^{-1/2}\mathbf{A}\mathbf{B}^{-1/2}\tilde{\mathbf{r}}}{\|\tilde{\mathbf{r}}\|_2^2} \\
&\simeq \mathbf{B}^{-1/2}\underset{\tilde{\mathbf{r}}}{\operatorname{argmax}} \frac{\tilde{\mathbf{r}}^\mathsf{T}\mathbf{B}^{-1/2}\mathbf{A}\mathbf{B}^{-1/2}\tilde{\mathbf{r}}}{\|\tilde{\mathbf{r}}\|_2^2} \\
&\simeq \mathbf{B}^{-1/2}\underset{\|\tilde{\mathbf{r}}\|_2=1}{\operatorname{argmax}} \tilde{\mathbf{r}}^\mathsf{T}\mathbf{B}^{-1/2}\mathbf{A}\mathbf{B}^{-1/2}\tilde{\mathbf{r}}
\end{aligned}
$$

Hence, $\underset{\|\mathbf{r}\|_2=1}{\operatorname{argmax}} \frac{\mathbf{r}^\mathsf{T}\mathbf{A}\mathbf{r}}{\mathbf{r}^\mathsf{T}\mathbf{B}\mathbf{r}} = \frac{\mathbf{B}^{-1/2}\tilde{\mathbf{r}}^*}{\|\mathbf{B}^{-1/2}\tilde{\mathbf{r}}^*\|_2}$ where $\tilde{\mathbf{r}}^* = \underset{\|\tilde{\mathbf{r}}\|_2=1}{\operatorname{argmax}} \tilde{\mathbf{r}}^\mathsf{T}\mathbf{B}^{-1/2}\mathbf{A}\mathbf{B}^{-1/2}\tilde{\mathbf{r}}$ $\qquad\square$

## B.6 Proof of Lemma 2.1

*Proof.* Let $\mathbf{r}_1' = \mathbf{A}\mathbf{r}_1$ and $\mathbf{r}_2' = \mathbf{B}\mathbf{r}_2$. As $\mathbf{r}_1$ and $\mathbf{r}_2$ are unit vectors, $\frac{1}{\|\mathbf{A}^{-1}\|_2} \leq \|\mathbf{r}_1'\|_2 \leq \|\mathbf{A}\|_2$.

$$
\begin{aligned}
\|\mathbf{r}_1' - \mathbf{r}_2'\|_2 &= \|\mathbf{A}\mathbf{r}_1 - \mathbf{A}\mathbf{r}_2 + \mathbf{A}\mathbf{r}_2 - \mathbf{B}\mathbf{r}_2\|_2 \\
&\leq \|\mathbf{A}\|_2\|\mathbf{r}_1 - \mathbf{r}_2\|_2 + \|\mathbf{A} - \mathbf{B}\|_2\|\mathbf{r}_2\|_2 \\
&\leq \|\mathbf{A}\|_2(\varepsilon_2 + \varepsilon_1)
\end{aligned}
$$

$$
\begin{aligned}
\left\|\frac{\mathbf{r}_1'}{\|\mathbf{r}_1'\|_2} - \frac{\mathbf{r}_2'}{\|\mathbf{r}_2'\|_2}\right\|_2 &= \frac{\|\|\mathbf{r}_2'\|_2\mathbf{r}_1' - \|\mathbf{r}_1'\|_2\mathbf{r}_2'\|_2}{\|\mathbf{r}_1'\|_2\|\mathbf{r}_2'\|_2} = \frac{\|\|\mathbf{r}_2'\|_2\mathbf{r}_1' - \|\mathbf{r}_1'\|_2\mathbf{r}_1' + \|\mathbf{r}_1'\|_2\mathbf{r}_1' - \|\mathbf{r}_1'\|_2\mathbf{r}_2'\|_2}{\|\mathbf{r}_1'\|_2\|\mathbf{r}_2'\|_2} \\
&\leq \frac{\|\|\mathbf{r}_2'\|_2 - \|\mathbf{r}_1'\|_2\|\|\mathbf{r}_1'\|_2 + \|\mathbf{r}_1'\|_2\|\mathbf{r}_1' - \mathbf{r}_2'\|_2}{\|\mathbf{r}_1'\|_2\|\mathbf{r}_2'\|_2} \leq \frac{2(\|\mathbf{A}\|_2(\varepsilon_2 + \varepsilon_1))}{\frac{1}{\|\mathbf{A}^{-1}\|_2} - \|\mathbf{A}\|_2(\varepsilon_2 + \varepsilon_1)} \\
&\leq \frac{2\frac{\lambda_{\max}(A)}{\lambda_{\min}(A)}(\varepsilon_2 + \varepsilon_1))}{1 - \frac{\lambda_{\max}(A)}{\lambda_{\min}(A)}(\varepsilon_2 + \varepsilon_1)} \leq 4\frac{\lambda_{\max}(A)}{\lambda_{\min}(A)}(\varepsilon_2 + \varepsilon_1) \text{ if } \frac{\lambda_{\max}(A)}{\lambda_{\min}(A)}(\varepsilon_2 + \varepsilon_1) \leq \frac{1}{2}
\end{aligned}
$$

$\qquad\square$

# C  Proof of Theorem 1.5

For the setting of Theorem 1.5, we provide Algorithm 4. It uses as a subroutine a polynomial time procedure PrincipalEigenVector for the principal eigen-vector of a symmetric matrix.

For notation, let $\boldsymbol{\Gamma} := \boldsymbol{\Sigma}^{1/2}$, then by Lemma E.13, we can write our linear threshold function $\operatorname{pos}(\mathbf{r}_*^\mathsf{T}\mathbf{X} + c_*) = \operatorname{pos}(\mathbf{u}_*^\mathsf{T}\mathbf{Z} - \ell)$ where $\mathbf{Z} \sim N(\mathbf{0}, \boldsymbol{I})$ where $\ell := -\frac{c_* + \mathbf{r}_*^\mathsf{T}\boldsymbol{\mu}}{\|\boldsymbol{\Gamma}\mathbf{r}_*\|_2}$, $\mathbf{u}_* := \boldsymbol{\Gamma}\mathbf{r}_*/\|\boldsymbol{\Gamma}\mathbf{r}_*\|_2$. For any $\mathbf{r} \in \mathbb{R}^d$, define $\mathbf{u} := \boldsymbol{\Gamma}\mathbf{r}/\|\boldsymbol{\Gamma}\mathbf{r}\|_2$. Let $\phi$ and $\Phi$ be the standard gaussian pdf and cdf respectively.

The following geometric bound is obtained by the algorithm.

**Lemma C.1.** *For any* $\varepsilon, \delta \in (0, 1)$*, if* $m \geq O\left((d/\varepsilon^4)O(\ell^2)\log(d/\delta)\left(\frac{\lambda_{\max}}{\lambda_{\min}}\right)^4 q^4\right)$*, then* $\hat{\mathbf{r}}$ *computed in Step 3 of Alg. 4 satisfies* $\min\{\|\hat{\mathbf{r}} - \mathbf{r}_*\|_2, \|\hat{\mathbf{r}} + \mathbf{r}_*\|_2\} \leq \varepsilon$*, w.p.* $1 - \delta/2$*.*

---

**Algorithm 4** PAC Learner for LTFs in over $N(\boldsymbol{\mu}, \boldsymbol{\Sigma})$

---

**Input:** $\mathcal{O} = \mathsf{Ex}(f, \mathcal{D} = N(\boldsymbol{\mu}, \boldsymbol{\Sigma}), q, k), m, s$, where $f(\mathbf{x}) = \mathsf{pos}\left(\mathbf{r}_*^\mathsf{T}\mathbf{x} + c_*\right), \|\mathbf{r}_*\|_2 = 1$.

1. Compute $\hat{\boldsymbol{\Sigma}}_B, \hat{\boldsymbol{\Sigma}}_D$ using MeanCovsEstimator with $m$ samples.
2. $\bar{\mathbf{r}} = \hat{\boldsymbol{\Sigma}}_B^{-1/2}\mathsf{PrincipalEigenVector}(\hat{\boldsymbol{\Sigma}}_B^{-1/2}\hat{\boldsymbol{\Sigma}}_D\hat{\boldsymbol{\Sigma}}_B^{-1/2})$ if $\hat{\boldsymbol{\Sigma}}_B^{-1/2}$ exists, else exit.
3. Let $\hat{\mathbf{r}} = \bar{\mathbf{r}}/\|\bar{\mathbf{r}}\|_2$.
4. If $k = q/2$
    - a. Sample a collection $\mathcal{M}$ of $s$ bags from $\mathcal{O}$.
    - b. For each bag $B_j$ in $\mathcal{M}$
        - i. Project each vector $\mathbf{x}_j^{(i)}$ in $B_j$ on $\hat{\mathbf{r}}$.
        - ii. Order these projections in a descending order $\hat{\mathbf{r}}^\mathsf{T}\mathbf{x}_j^{(1)} > ... > \hat{\mathbf{r}}^\mathsf{T}\mathbf{x}_j^{(q)}$.
        - ii. Define $h_j = \mathsf{pos}(\hat{\mathbf{r}}^\mathsf{T}\mathbf{x} + \hat{\mathbf{r}}^\mathsf{T}\mathbf{x}_j^{(k)})$
    - c. **Return** $h^* \in \{h_1, ..., h_s\}$ which has lower $\mathsf{BagErr}_{\text{sample}}(h^*, \mathcal{M})$.
5. else
    - a. Sample a collection $\mathcal{M}$ of $s$ bags from $\mathcal{O}$.
    - b. For each bag $B_j$ in $\mathcal{M}$
        - i. Project each vector $\mathbf{x}_j^{(i)}$ in $B_j$ on $\hat{\mathbf{r}}$.
        - ii. Order these projections in a descending order $\hat{\mathbf{r}}^\mathsf{T}\mathbf{x}_j^{(1)} > ... > \hat{\mathbf{r}}^\mathsf{T}\mathbf{x}_j^{(q)}$.
        - ii. Define $h_j = \mathsf{pos}(\hat{\mathbf{r}}^\mathsf{T}\mathbf{x} + \hat{\mathbf{r}}^\mathsf{T}\mathbf{x}_j^{(k)})$
    - c. For each bag $B_j$ in $\mathcal{M}$
        - i. Project each vector $\mathbf{x}_j^{(i)}$ in $B_j$ on $-\hat{\mathbf{r}}$.
        - ii. Order these projections in a descending order $-\hat{\mathbf{r}}^\mathsf{T}\mathbf{x}_j^{(1)} > ... > -\hat{\mathbf{r}}^\mathsf{T}\mathbf{x}_j^{(q)}$.
        - ii. Define $\tilde{h}_j = \mathsf{pos}(-\hat{\mathbf{r}}^\mathsf{T}\mathbf{x} - \hat{\mathbf{r}}^\mathsf{T}\mathbf{x}_j^{(k)})$
    - d. **Return** $h^* \in \{h_1, ..., h_s, \tilde{h}_1, ..., \tilde{h}_s\}$ which has lower $\mathsf{BagErr}_{\text{sample}}(h^*, \mathcal{M})$.

---

The above lemma, whose proof is deferred to Sec. C.1, is used in conjunction with the following lemma.

**Lemma C.2.** *Let $\varepsilon, \delta \in (0, 1)$ and suppose that $\hat{\mathbf{r}}$ computed in Step 3 of Alg. 4 satisfies if $k \neq q/2$ $\min\{\|\hat{\mathbf{r}} - \mathbf{r}_*\|_2, \|\hat{\mathbf{r}} + \mathbf{r}_*\|_2\} \leq \varepsilon,$. Then, with $s \geq O\left(d(\log q + \log(1/\delta))/\varepsilon^2\right)$, $h^*$ computed in Step 4.c or Step 5.d satisfies*

$$\Pr_{\mathcal{D}}\left[h^*(\mathbf{x}) \neq f(\mathbf{x})\right] \leq \frac{8q\varepsilon}{\Phi(\ell)(1 - \Phi(\ell))}\left(c_0\sqrt{\tfrac{\lambda_{\max}}{\lambda_{\min}}} + c_1\tfrac{\|\boldsymbol{\mu}\|_2}{\sqrt{\lambda_{\min}}}\right)$$

*and if $k = q/2$,*

$$\min(\Pr_{\mathcal{D}}\left[h^*(\mathbf{x}) \neq f(\mathbf{x})\right], \Pr_{\mathcal{D}}\left[(1 - h^*(\mathbf{x})) \neq f(\mathbf{x})\right]) \leq \frac{8q\varepsilon}{\Phi(\ell)(1 - \Phi(\ell))}\left(c_0\sqrt{\tfrac{\lambda_{\max}}{\lambda_{\min}}} + c_1\tfrac{\|\boldsymbol{\mu}\|_2}{\sqrt{\lambda_{\min}}}\right)$$

*w.p. $1 - \delta/2$, where $c_0, c_1$ are the constants from Lemma 2.3.*

With the above we complete the proof of Theorem 1.5 as follows.

*Proof.* (of Theorem 1.5) Let the parameters $\delta, \varepsilon$ be as given in the statement of the theorem. We use $O\left(\varepsilon\frac{(\Phi(\ell)(1-\Phi(\ell)))\sqrt{\lambda_{\min}}}{q(\sqrt{\lambda_{\max}} + \|\boldsymbol{\mu}\|_2)}\right)$ for the error bound in Lemma C.1. We now take $m$ to be of $m = O\left((d/\varepsilon^4)\frac{O(\ell^2)}{(\Phi(\ell)(1-\Phi(\ell)))^2}\log(d/\delta)\left(\frac{\lambda_{\max}}{\lambda_{\min}}\right)^4\left(\frac{\sqrt{\lambda_{\max}} + \|\boldsymbol{\mu}\|_2}{\sqrt{\lambda_{\min}}}\right)^4 q^8\right)$ in Alg. 4 we obtain the following bound: $\min\{\|\hat{\mathbf{r}} - \mathbf{r}_*\|_2, \|\hat{\mathbf{r}} + \mathbf{r}_*\|_2\} \leq \varepsilon\frac{(\Phi(\ell)(1-\Phi(\ell)))\sqrt{\lambda_{\min}}}{q(c_0\sqrt{\lambda_{\max}} + c_1\|\boldsymbol{\mu}\|_2)}$ with probability $1 - \delta/2$. Using $s \geq O\left(d(\log(q) + \log(1/\delta))\frac{q^2(\sqrt{\lambda_{\max}} + \|\boldsymbol{\mu}\|_2)^2}{\varepsilon^2(\Phi(\ell)(1-\Phi(\ell))^2\lambda_{\min}}\right)$, Lemma C.2 yields the desired misclassification error bound of $\varepsilon$ on $h^*$ w.p. $1 - \delta$. $\qquad\square$

*Proof.* (of Lemma C.2) Define $h(\mathbf{x}) := \mathsf{pos}(\hat{\mathbf{r}}^\mathsf{T}\mathbf{X} + c_*)$ and $\tilde{h}(\mathbf{x}) := \mathsf{pos}(-\hat{\mathbf{r}}^\mathsf{T}\mathbf{X} + c_*)$. Applying Lemma 2.3, we obtain that at least one of $h, \tilde{h}$ has an instance misclassification error of at most

$O(\varepsilon(\sqrt{\lambda_{\max}/\lambda_{\min}} + \|\boldsymbol{\mu}\|_2/\lambda_{\min}))$. WLOG assume that $h$ satisfies this error bound i.e., $\Pr_{\mathcal{D}}[f(\mathbf{x}) \neq h(\mathbf{x})] \leq \varepsilon(c_0\sqrt{\lambda_{\max}/\lambda_{\min}} + c_1\|\boldsymbol{\mu}\|_2/\sqrt{\lambda_{\min}}) =: \varepsilon'$. Note that, $\Pr_{\mathcal{D}}[f(\mathbf{x}) = 1] = \Phi(\ell), \Pr_{\mathcal{D}}[f(\mathbf{x}) = 0] = 1 - \Phi(\ell)$. Thus,

$$\Pr_{\mathcal{D}}[h(x) \neq f(x) \mid f(\mathbf{x}) = 1] \leq \varepsilon'/\Phi(\ell), \qquad \Pr_{\mathcal{D}}[[h(x) \neq f(x) \mid f(\mathbf{x}) = 0] \leq \varepsilon'/(1 - \Phi(\ell)).$$

Therefore, taking a union bound we get that the probability that a random bag from the oracle contains a feature vector on which $f$ and $h$ disagree is at most $q\varepsilon'/\Phi(\ell)(1 - \Phi(\ell))$. Applying Chernoff bound (see Appendix B.1) we obtain that with probability at least $1 - \delta/6$, $\mathsf{BagErr}_{\mathrm{sample}}(h, \mathcal{M}) \leq 2q\varepsilon'/\Phi(\ell)(1 - \Phi(\ell))$. Therefore, $h$ satisfies $\mathsf{BagErr}_{\mathrm{sample}}(h^*, \mathcal{M}) \leq 2q\varepsilon'/\Phi(\ell)(1 - \Phi(\ell))$. Hence, there exists at least one $h_j(\mathbf{x}) = \mathsf{pos}(\hat{\mathbf{r}}^\top\mathbf{x} + \hat{\mathbf{r}}^\top\mathbf{x}_j^{(k)})$ will satisfy $\mathsf{BagErr}_{\mathrm{sample}}(h_j, \mathcal{M}) \leq 2q\varepsilon'/\Phi(\ell)(1 - \Phi(\ell))$. Since, $h^*(\mathbf{x})$ has the minimum sample bag error among all $h_j(\mathbf{x})$, $\mathsf{BagErr}_{\mathrm{sample}}(h^*, \mathcal{M}) \leq 2q\varepsilon'/\Phi(\ell)(1 - \Phi(\ell))$.

On the other hand, applying Theorem 2.2, except with probability $\delta/3$, $\Pr_{\mathcal{D}}[f(\mathbf{x}) \neq h^*(\mathbf{x})] \leq 8q\varepsilon'/\Phi(\ell)(1 - \Phi(\ell)) = \frac{8q\varepsilon}{\Phi(\ell)(1-\Phi(\ell))}(c_0\sqrt{\lambda_{\max}/\lambda_{\min}} + c_1\|\boldsymbol{\mu}\|_2/\sqrt{\lambda_{\min}})$ if $k \neq q/2$ and $\min(\Pr_{\mathcal{D}}[f(\mathbf{x}) \neq h^*(\mathbf{x})], \Pr_{\mathcal{D}}[f(\mathbf{x}) \neq (1 - h^*(\mathbf{x}))]) \leq 8q\varepsilon'/\Phi(\ell)(1 - \Phi(\ell)) = \frac{8q\varepsilon}{\Phi(\ell)(1-\Phi(\ell))}(c_0\sqrt{\lambda_{\max}/\lambda_{\min}} + c_1\|\boldsymbol{\mu}\|_2/\sqrt{\lambda_{\min}})$ if $k = q/2$. Therefore, except with probability $\delta/2$, the bound in Lemma C.2 holds. $\qquad\square$

### C.1 Proof of Lemma C.1

We use these to generalize a few quantities we had defined earlier. We obtain their bounds using E.2,

**Definition C.3.** *Define:*

$$\kappa_1 := \left(\frac{\phi(\ell)\left(\frac{k}{q} - (1 - \Phi(\ell))\right)}{\Phi(\ell)(1 - \Phi(\ell))}\right)^2 - \frac{\ell\phi(\ell)\left(\frac{k}{q} - (1 - \Phi(\ell))\right)}{\Phi(\ell)(1 - \Phi(\ell))}, \qquad \kappa_1 \geq -\ell^2/4$$

$$\kappa_2 := \frac{2}{q-1}\frac{k}{q}\left(1 - \frac{k}{q}\right)\left(\frac{\phi(\ell)}{\Phi(\ell)(1 - \Phi(\ell))}\right)^2, \qquad \kappa_2 \geq 2\ell^2/q^2 \qquad \text{when } \ell > 1$$

$$\kappa_3 := \frac{\kappa_2}{(1 - \kappa_1)(1 - \max(0, \kappa_1))}, \qquad \kappa_3 \geq \frac{2\ell^2}{q^2(1 + \ell^2/4)} \qquad \text{when } \ell > 1$$

$$\theta := \frac{2\lambda_{\max}}{\lambda_{\min}}\left(\frac{1}{2 - \max(0, 2\kappa_1 - \kappa_2)} + \frac{1}{1 - \max(0, \kappa_1)}\right), \qquad \frac{3\lambda_{\max}}{\lambda_{\min}} \leq \theta \leq \frac{3\lambda_{\max}}{(1 - K_2)\lambda_{\min}}$$

*Where $K_i$'s are some finite functions of $K$.*

Similar to Lemma 4.4, we again show in the following lemma that $\hat{\mathbf{r}}$ in the algorithms is indeed $\pm\mathbf{r}_*$ if the covariance estimates were the actual covariances.

**Lemma C.4.** *The ratio $\rho(\mathbf{r}) := \mathbf{r}^\top\boldsymbol{\Sigma}_D\mathbf{r}/\mathbf{r}^\top\boldsymbol{\Sigma}_B\mathbf{r}$ is maximized when $\mathbf{r} = \pm\mathbf{r}_*$. Moreover,*

$$\rho(\mathbf{r}) = 2 + \frac{\gamma(\mathbf{r})^2\kappa_2}{1 - \gamma(\mathbf{r})^2\kappa_1} \quad \text{where } \gamma(\mathbf{r}) := \frac{\mathbf{r}^\top\boldsymbol{\Sigma}\mathbf{r}_*}{\sqrt{\mathbf{r}^\top\boldsymbol{\Sigma}\mathbf{r}}\sqrt{\mathbf{r}_*^\top\boldsymbol{\Sigma}\mathbf{r}_*}} \text{ and}$$

$$\mathbf{r}^\top\boldsymbol{\Sigma}_B\mathbf{r} = \mathbf{r}^\top\boldsymbol{\Sigma}\mathbf{r}(1 - \gamma(\mathbf{r})^2\kappa_1), \quad \mathbf{r}^\top\boldsymbol{\Sigma}_D\mathbf{r} = \mathbf{r}^\top\boldsymbol{\Sigma}\mathbf{r}(2 - \gamma(\mathbf{r})^2(2\kappa_1 - \kappa_2))$$

*Proof.* Using the transformations to $\mathbf{Z}$, we can let $\mathbf{X}_B = \boldsymbol{\Gamma}\mathbf{Z}_B$ as a random feature-vector sampled uniformly from a random bag sampled from $\mathcal{O}$. Also, let $\mathbf{X}_D = \boldsymbol{\Gamma}\mathbf{Z}_D$ be the difference of two random feature vectors sampled uniformly without replacement from a random bag sampled from $\mathcal{O}$. Observe that the ratio $\rho(\mathbf{r}) = \mathrm{Var}[\mathbf{r}^\top\mathbf{X}_D]/\mathrm{Var}[\mathbf{r}^\top\mathbf{X}_B] = \mathrm{Var}[\mathbf{u}^\top\mathbf{Z}_D]/\mathrm{Var}[\mathbf{u}^\top\mathbf{Z}_B]$.

Let $g^* := \mathbf{u}_*^\top\mathbf{Z}$ which is $N(0, 1)$. For $a \in \{0, 1\}$, let $\mathbf{Z}_a$ be $\mathbf{Z}$ conditioned on $\mathsf{pos}\left(\mathbf{u}_*^\top\mathbf{Z} - \ell\right) = a$. Let $g_a^* := \mathbf{u}_*^\top\mathbf{Z}_a$, $a \in \{0, 1\}$. $g_0^*$ lower-tailed one-sided truncated normal distributions truncated at $\ell$ and $g_1^*$ upper-tailed one-sided truncated normal distributions truncated at $\ell$. Hence, $\mathbb{E}[g_1^*] = \phi(\ell)/(1 - \Phi(\ell))$, $\mathbb{E}[g_0^*] = -\phi(\ell)/\Phi(\ell)$, $\mathbb{E}[g_1^{*2}] = 1 + \ell\phi(\ell)/(1 - \Phi(\ell))$, $\mathbb{E}[g_0^{*2}] = 1 - \ell\phi(\ell)/\Phi(\ell)$.

With this setup, letting $g_B^* := \mathbf{u}_*^\mathsf{T}\mathbf{Z}_B$ and $g_D^* := \mathbf{u}_*^\mathsf{T}\mathbf{Z}_D$ we obtain (using Lemma B.2 in Appendix B.2)

$$\mathrm{Var}[g_B^*] = 1 - \kappa_1, \quad \mathrm{Var}[g_D^*] = 2(1 - \kappa_1) + \kappa_2$$

Now let $\tilde{\mathbf{u}}$ be a unit vector orthogonal to $\mathbf{u}_*$. Let $\tilde{g} = \tilde{\mathbf{u}}^\mathsf{T}\mathbf{Z}$ be $N(0,1)$. Also, let $\tilde{g}_a = \tilde{\mathbf{u}}^\mathsf{T}\mathbf{Z}_a$ for $a \in \{0,1\}$. Since $\mathbf{Z}_a$ are given by conditioning $\mathbf{Z}$ only along $\mathbf{u}_*$, $\tilde{g}_a \sim N(0,1)$ for $a \in \{0,1\}$. In particular, the component along $\tilde{u}$ of $\mathbf{Z}_B$ (call it $\tilde{g}_B$) is $N(0,1)$ and that of $\mathbf{Z}_D$ (call it $\tilde{g}_D$) is the difference of two iid $N(0,1)$ variables. Thus, $\mathrm{Var}[\tilde{g}_B] = 1$ and $\mathrm{Var}[\tilde{g}_D] = 2$. Moreover, due to orthogonality all these gaussian variables corresponding to $\tilde{\mathbf{u}}$ are independent of those corresponding to $\mathbf{u}_*$ defined earlier.

Now let $\mathbf{u} = \alpha\mathbf{u}_* + \beta\tilde{\mathbf{u}}$, where $\beta = \sqrt{1-\alpha^2}$ be any unit vector. From the above we have,

$$\frac{\mathrm{Var}\left[\mathbf{u}^\mathsf{T}\mathbf{Z}_D\right]}{\mathrm{Var}\left[\mathbf{u}^\mathsf{T}\mathbf{Z}_B\right]} = \frac{\mathrm{Var}\left[\alpha g_D^* + \beta\tilde{g}_D\right]}{\mathrm{Var}\left[\alpha g_B^* + \beta\tilde{g}_B\right]} = \frac{\alpha^2\,\mathrm{Var}\left[g_D^*\right] + \beta^2\,\mathrm{Var}\left[\tilde{g}_D\right]}{\alpha^2\,\mathrm{Var}\left[g_B^*\right] + \beta^2\,\mathrm{Var}\left[\tilde{g}_B\right]}$$

$$= \frac{2\alpha^2(1-\kappa_1) + \alpha^2\kappa_2 + 2\beta^2}{\alpha^2(1-\kappa_1) + \beta^2}$$

$$= 2 + \frac{\alpha^2\kappa_2}{1 - \alpha^2\kappa_1} \tag{7}$$

where the last equality uses $\beta = \sqrt{1-\alpha^2}$. Letting $\mathbf{u} = \mathbf{\Gamma r}/\|\mathbf{\Gamma r}\|_2$ we obtain that $\alpha = \frac{\langle\mathbf{\Gamma r}, \mathbf{\Gamma r}_*\rangle}{\|\mathbf{\Gamma r}\|_2\|\mathbf{\Gamma r}_*\|_2} = \gamma(\mathbf{r})$ completing the proof. $\qquad\square$

Lemma 4.5 shows that ratio maximization can be treated as an eigenvalue decomposition problem of the matrix $\mathbf{\Sigma}_B^{-1/2}\mathbf{\Sigma}_D\mathbf{\Sigma}_B^{-1/2}$. We now prove that the error in the estimate of $\hat{\mathbf{r}}$ given to us by the algorithm is bounded if the error in the covariance estimates are bounded. The sample complexity of computing these estimates gives the sample complexity of our algorithm.

We now complete the proof of Lemma C.1 (with $\delta$ instead of $\delta/2$ for convenience). By Lemma 3.1, taking $m \geq O\left((d/\varepsilon_1^2)O(\ell^2)\log(d/\delta)\right)$ ensures that $\|\mathbf{E}_B\|_2 \leq \varepsilon_1\lambda_{\max}$ and $\|\mathbf{E}_D\|_2 \leq \varepsilon_1\lambda_{\max}$ w.p. at least $1 - \delta$ where $\mathbf{E}_B = \hat{\mathbf{\Sigma}}_B - \mathbf{\Sigma}_B$ and $\mathbf{E}_D = \hat{\mathbf{\Sigma}}_D - \mathbf{\Sigma}_D$. We start by defining $\hat{\rho}(\mathbf{r}) := \frac{\mathbf{r}^\mathsf{T}\hat{\mathbf{\Sigma}}_D\mathbf{r}}{\mathbf{r}^\mathsf{T}\hat{\mathbf{\Sigma}}_B\mathbf{r}}$ which is the equivalent of $\rho$ using the estimated matrices. Observe that it can be written as $\hat{\rho}(\mathbf{r}) = \frac{\mathbf{r}^\mathsf{T}\mathbf{\Sigma}_B\mathbf{r} + \mathbf{r}^\mathsf{T}\mathbf{E}_B\mathbf{r}}{\mathbf{r}^\mathsf{T}\mathbf{\Sigma}_D\mathbf{r} + \mathbf{r}^\mathsf{T}\mathbf{E}_D\mathbf{r}}$. Using these we can obtain the following bound on $\hat{\rho}$: for any $\mathbf{r} \in \mathbb{R}^d$, $|\hat{\rho}(\mathbf{r}) - \rho(\mathbf{r})| \leq \theta\varepsilon_1|\rho(\mathbf{r})|$ w.p. at least $1 - \delta$ (*) as long as $\varepsilon_1 \leq \frac{(1-\max(0,\kappa_1))}{2}\frac{\lambda_{\min}}{\lambda_{\max}}$, which we shall ensure (see Appendix B.4).

For convenience we denote the normalized projection of any vector $\mathbf{r}$ as $\tilde{\mathbf{r}} := \frac{\mathbf{\Sigma}^{1/2}\mathbf{r}}{\|\mathbf{\Sigma}^{1/2}\mathbf{r}\|_2}$. Now let $\tilde{\mathbf{r}} \in \mathbb{R}^d$ be a unit vector such that $\min\{\|\tilde{\mathbf{r}} - \tilde{\mathbf{r}}_*\|_2, \|\tilde{\mathbf{r}} + \tilde{\mathbf{r}}_*\|_2\} \geq \varepsilon_2$. Hence, using the definitions from Lemma C.4, $|\gamma(\mathbf{r})| \leq 1 - \varepsilon_2^2/2$ while $\gamma(\mathbf{r}_*) = 1$ which implies $\rho(\mathbf{r}_*) - \rho(\mathbf{r}) \geq \kappa_3\varepsilon_2^2/2$. Note that $\rho(\mathbf{r}) \leq \rho(\mathbf{r}_*) = 2 + \kappa_3(1 - \max(0,\kappa_1))$. Choosing $\varepsilon_1 < \frac{\kappa_3}{2\theta(2+\kappa_3(1-\max(0,\kappa_1)))}\varepsilon_2^2$, we obtain that $\rho(\mathbf{r}_*)(1 - \theta\varepsilon_1) > \rho(\mathbf{r})(1 + \theta\varepsilon_1)$. Using this along with the bound (*) we obtain that w.p. at least $1 - \delta$, $\hat{\rho}(\mathbf{r}_*) > \hat{\rho}(\mathbf{r})$ when $\varepsilon_2 > 0$. Since our algorithm returns $\hat{\mathbf{r}}$ as the maximizer of $\hat{\rho}$, w.p. at least $1 - \delta$ we get $\min\{\|\tilde{\mathbf{r}} - \tilde{\mathbf{r}}_*\|_2, \|\tilde{\mathbf{r}} + \tilde{\mathbf{r}}_*\|_2\} \leq \varepsilon_2$. Using Lemma 2.1, $\min\{\|\hat{\mathbf{r}} - \mathbf{r}_*\|_2, \|\hat{\mathbf{r}} + \mathbf{r}_*\|_2\} \leq 4\sqrt{\frac{\lambda_{\max}}{\lambda_{\min}}}\varepsilon_2$. Substituting $\varepsilon_2 = \frac{\varepsilon}{4}\sqrt{\frac{\lambda_{\min}}{\lambda_{\max}}}$, $\|\mathbf{r} - \mathbf{r}_*\|_2 \leq \varepsilon$ w.p. at least $1 - \delta$. The conditions on $\varepsilon_1$ are satisfied by taking it to be $\leq O\left(\frac{\kappa_3\varepsilon^2\lambda_{\min}}{\theta(2+\kappa_3(1-\max(0,\kappa_1)))\lambda_{\max}}\right)$, and thus we can take $m \geq O\left((d/\varepsilon^4)O(\ell^2)\log(d/\delta)\left(\frac{\lambda_{\max}}{\lambda_{\min}}\right)^2\theta^2\left(\frac{2+\kappa_3(1-\max(0,\kappa_1))}{\kappa_3}\right)^2\right) = O\left((d/\varepsilon^4)O(\ell^2)\log(d/\delta)\left(\frac{\lambda_{\max}}{\lambda_{\min}}\right)^4 q^4\right)$ since $1/\kappa_3 \leq q^2(1+\ell^2/4)/\ell^2$ whenever $\ell > 1$(Defn. C.3). This completes the proof of Lemma C.1.

## D  Generalization error Bounds

We show that if a hypothesis LTF $h$ satisfies close to 1 fraction of sufficient number of bags sampled from a bag oracle then with high probability $h$ is a good approximator for the target LTF $f$ (or its

complement in the case of balanced bags). The first step is to prove a generalization bound from the sample bag-level accuracy to the oracle bag-level accuracy.

**Theorem D.1.** *Let $\mathcal{O} := \mathsf{Ex}(f, \mathcal{D}, q, k)$ be any bag oracle for an LTF $f$ in $d$-dimensions, and let $\mathcal{M}$ be a collection of $m$ bags sampled iid from the oracle. Then, there is an absolute constant $C_0 \leq 1000$ s.t. w.p. at least $1 - \delta$,*

$$\mathsf{BagErr}_{\mathrm{oracle}}(h, f, \mathcal{D}, q, k) \leq \mathsf{BagErr}_{\mathrm{sample}}(h, \mathcal{M}) + \varepsilon \tag{8}$$

*when $m \geq C_0 d \left(\log q + \log(1/\delta)\right) / \varepsilon^2$, for any $\delta, \varepsilon > 0$.*

*Proof.* The proof follows from the arguments similar to the ones used in Appendix M of [26] to prove bag satisfaction generalization bounds. Consider a bag loss function of the form $\ell(\phi^\zeta(h, B), (\phi^\zeta(f, B))$, where $\phi^\zeta(g, B) := \zeta(\mathbf{y}_{g,B})$ where $\mathbf{y}_{g,B}$ is the vector of $(g(\mathbf{x}))_{\mathbf{x} \in B}$, for $\zeta : \{0, 1\}^q \to \mathbb{R}$. The result of [35] showed generalization error bounds when (i) $\zeta$ is 1-Lipschitz w.r.t. to $\infty$-norm, and (ii) $\ell$ is 1-Lipschitz in the first coordinate. We can thus apply their results using the bound of $(d + 1)$ on the VC-dimension of LTFs in $d$-dimensions to show that the above bound on $m$ holds (with $C_0/8$ instead of $C_0$) for the generalization of the following bag error:

$$|\gamma(B, f, t) - \gamma(B, h, t)|,$$

where

$$\gamma(B, g, t) := \begin{cases} 0 & \text{if } \sum_{\mathbf{x} \in B} g(\mathbf{x}) \leq t \\ 1 & \text{otherwise.} \end{cases} \tag{9}$$

for $t \in \{0, \ldots, q-1\}$. We can bound our bag satisfaction error by the sum of the generalization errors of $|\gamma(B, f, k) - \gamma(B, h, k)|$, and $|\gamma(B, f, k-1) - \gamma(B, h, k-1)|$, which can each be bounded by $\varepsilon/2$ thus completing the proof. □

Next we show that if the oracle-level bag accuracy of $h$ is high then this translates to $h$ is being a low error instance-level approximator for the target LTF $f$ (or its complement in the case of balanced bags). With the setup as used in the previous theorem, let us define define the regions $S_a := \{\mathbf{x} \text{ s.t } f(\mathbf{x}) = a\}$ for $a \in \{0, 1\}$, and $S_{ab} := \{\mathbf{x} \text{ s.t } f(\mathbf{x}) = a, h(\mathbf{x}) = b\}$. Let $\mu$ be the measure induced by $\mathcal{D}$, $\mu_a$ and $\mu_{ab}$ be the respectively conditional measures induced on $S_a$ and $S_{ab}$. The oracle $\mathcal{O}$ for a random bag $B$, samples $k$ points iid from $(S_1, \mu_1)$ and $(q - k)$ points from $(S_0, \mu_0)$.

we have the following theorem.

**Theorem D.2.** *Suppose $k \in \{1, \ldots, q\}$, and $0 < \mathsf{BagAcc}_{\mathrm{oracle}}(h, f, \mathcal{D}, q, k) \leq \varepsilon' < 1/(4q)$, then*

(i) $\Pr_{\mathcal{D}}[f(\mathbf{x}) \neq h(\mathbf{x})] \leq \varepsilon$ *if* $k \neq q/2$, *and*
(ii) $\Pr_{\mathcal{D}}[f(\mathbf{x}) \neq h(\mathbf{x})] \leq \varepsilon$ *or* $\Pr[f(\mathbf{x}) \neq (1 - h(\mathbf{x}))] \leq \varepsilon$, *if* $k = q/2$,

*where $\varepsilon = 4\varepsilon'$.*

Before we prove the above them, we need the following lemma bounding the probability that two independent binomial random variables take the same value. Let Binomial$(n, p)$ be the sum of $n$ iid $\{0, 1\}$ random variables each with expectation $p$.

**Lemma D.3.** *Let $u, v \in \{1, \ldots, q\}$, and $X_1 \sim$ Binomial$(u, p_1)$ and $X_2 \sim$ Binomial$(v, p_2)$ be independent binomial distributions for some probabilities $p_1$ and $p_2$. Let $p^* = \min\{\max\{p_1, (1 - p_1)\}, \max\{p_2, (1 - p_2)\}\}$, then, $\Pr[X_1 \neq X_2] \geq 1 - \sqrt{p^*}$.*

*Proof.* We first begin by bounding a binomial coefficient by the sum of its adjacent binomial coefficients. Let $n \geq 1$ and $n > r > 0$. We begin with the standard identity and proceed further:

$$\binom{n}{r} = \binom{n-1}{r} + \binom{n-1}{r-1}$$

$$\leq \frac{n}{r+1}\binom{n-1}{r} + \frac{n}{n-r+1}\binom{n-1}{r-1} \tag{10}$$

$$= \binom{n}{r+1} + \binom{n}{r-1} \tag{11}$$

Moreover, it is trivially true that $\binom{n}{0} \leq \binom{n}{1}$ and $\binom{n}{n} \leq \binom{n}{n-1}$, therefore (11) holds even for $r \in \{0, n\}$ whenever the binomial coefficients exist. Now, for some probability $p$ define $\nu(p, n, r) := \binom{n}{r} p^r (1-p)^{1-r}$ which is pdf at $r$ of Binomial$(n, p)$. The above implies the following:

$$\nu(p, n, r) \leq p' \left( \nu(p, n, r-1) + \nu(p, n, r+1) \right) \tag{12}$$

where $p' = \max\{p/(1-p), (1-p)/p\}$. Using the fact that $\nu(p, n, r-1) + \nu(p, n, r) + \nu(p, n, r+1) \leq 1$, we obtain that

$$(1/p')\nu(p, n, r) + \nu(p, n, r) \leq 1$$
$$\Rightarrow \quad \nu(p, n, r) \leq (p'/1 + p') = \max\{p, 1-p\} \tag{13}$$

The above allows us to complete the proof of the lemma as follows.

$$\Pr\left[X_1 = X_2\right]$$
$$= \sum_{r=0}^{\min\{u, v\}} \nu(p_1, u, r)\nu(p_2, v, r)$$
$$\leq \left( \sum_{r=0}^{u} \nu(p_1, u, r)^2 \right)^{\frac{1}{2}} \left( \sum_{r=0}^{v} \nu(p_2, v, r)^2 \right)^{\frac{1}{2}}$$
$$\leq \left( \max_r \{\nu(p_1, u, r)\} \sum_{r=0}^{u} \nu(p_1, u, r) \right)^{\frac{1}{2}}$$
$$\cdot \left( \max_r \{\nu(p_2, v, r)\} \sum_{r=0}^{u} \nu(p_2, v, r) \right)^{\frac{1}{2}}$$
$$\leq (\max\{p_1, 1-p_1\})^{\frac{1}{2}} (\max\{p_2, 1-p_2\})^{\frac{1}{2}}$$
$$\leq \sqrt{p^*}, \tag{14}$$

where we use Cauchy-Schwarz for the first inequality. $\square$

*Proof.* (of Theorem D.2) We now consider three cases, for each one we shall prove points (i) and (ii) of the theorem.

*Case* $\Pr_{\mathcal{D}}[f(\mathbf{x}) \neq h(\mathbf{x})] \geq 1 - \varepsilon$. This condition means that $\mu(S_{10}) + \mu(S_{01}) \geq 1 - \varepsilon$, which implies that at least one of $\mu_1(S_{10}), \mu_0(S_{01})$ is $\geq 1 - \varepsilon$. Assume that $\mu_0(S_{01}) \geq 1 - \varepsilon$ (the other case is analogous).

Consider unbalanced bags i.e., $k \neq q/2$. Now in case that $\mu_1(S_{10}) \geq 1 - \varepsilon$, all the points sampled from $S_1$ are sampled from $S_{10}$ w.p. $(1 - k\varepsilon)$ and those sampled from $S_0$ are all sampled from $S_{01}$ w.p. $(1 - (q - k)\varepsilon)$. Therefore, with probability at least $(1 - q\varepsilon)$ all the points are sampled from $S_{10} \cup S_{01}$. Since $k \neq q/2$ this implies that $h$ does not satisfy the random bags with probability at least $(1 - q\varepsilon) > \varepsilon'$. If $\mu_1(S_{10}) \leq \varepsilon$, we can show with similar arguments that with probability $\geq (1 - q\varepsilon) > \varepsilon'$ no points are sampled from $S_{10}$ and $q - k$ points are sampled from $S_{01}$ which means (since $k \geq 1$) that $h$ does not satisfy the bag. Finally, let $\mu_1(S_{10}) \in (\varepsilon, 1 - \varepsilon)$. In this case, the number of points sampled from $S_{10}$ is distributed as Binomial$(k, \mu_1(S_{10}))$ and those sampled from $S_{01}$ is independently distributed as Binomial$(k, \mu_0(S_{01}))$. If these two numbers are different then $h$ does not satisfy the bag. We can apply Lemma D.3 and using the bounds on $\mu_1(S_{10})$, we get that $p^* \leq 1 - \varepsilon$. Therefore, the probability of $h$ not satisfying a randomly sampled bag is at least

$$1 - \sqrt{1 - \varepsilon} \geq \frac{(1 - (1 - \varepsilon))}{1 + \sqrt{1 - \varepsilon}} \geq \varepsilon/2 > \varepsilon'. \tag{15}$$

For balanced bags, the case $\Pr_{\mathcal{D}}[f(\mathbf{x}) \neq h(\mathbf{x})] \geq 1 - \varepsilon$ implies that $\Pr[f(\mathbf{x}) \neq (1 - h(\mathbf{x}))] \leq \varepsilon$, so the condition (ii) of the theorem holds.

*Case* $\varepsilon < \Pr_{\mathcal{D}}[f(\mathbf{x}) \neq h(\mathbf{x})] < 1 - \varepsilon$. This case violates the conditions (i) and (ii). First observe that, $\mu_1(S_{10})$ and $\mu_0(S_{01})$ both cannot be $\geq (1 - \varepsilon)$ or $\leq \varepsilon$, otherwise $\mu(S_{10}) + \mu(S_{01})$ is either $\geq 1 - \varepsilon$ or $\leq \varepsilon$ violating the assumption of this case. For the subcase that $\mu_1(S_{10}) \leq \varepsilon$ and $\mu_0(S_{01}) \geq (1 - \varepsilon)$,

we can show using arguments similar to the previous case that w.p. $\geq (1 - q\varepsilon)$ no points are sampled from $S_{10}$ and $q - k$ points are sampled from $S_{01}$, and thus, $h$ will not satisfy a random bag with probability at least $(1 - q\varepsilon) > \varepsilon'$. The subcase when $\mu_1(S_{01}) \leq \varepsilon$ and $\mu_0(S_{10}) \geq (1 - \varepsilon)$ is analogous. Finally, we have the subcase that $\mu_1(S_{10})$ or $\mu_0(S_{01})$ both lie in the range $(\varepsilon, 1 - \varepsilon)$. Now, the number of points sampled from $S_{10}$ is distributed as $\text{Binomial}(k, \mu_1(S_{10}))$ and those sampled from $S_{01}$ is independently distributed as $\text{Binomial}(k, \mu_0(S_{01}))$. If these two numbers are different then $h$ does not satisfy the bag. We can apply Lemma D.3 and using the bounds on one of $\mu_1(S_{10})$ or $\mu_0(S_{01})$, we get that $p^* \leq 1 - \varepsilon$. Therefore, the probability of $h$ not satisfying a randomly sampled bag is at least $1 - \sqrt{1 - \varepsilon} \geq \varepsilon/2 > \varepsilon'$, using (15).

$\square$

## D.1 Proof of Theorem 2.2

The proof follows directly from Theorems D.1 and D.2.

## D.2 Proof of Lemma 2.3

*Proof.* We have,

$$\Pr[\mathsf{pos}(\mathbf{r}^\mathsf{T}\mathbf{X} + c) \neq \mathsf{pos}(\hat{\mathbf{r}}^\mathsf{T}\mathbf{X} + c)] = \Pr[\mathsf{pos}(\mathbf{r}^\mathsf{T}\tilde{\mathbf{X}} + \|\mathbf{\Gamma}\mathbf{r}\|_2\zeta) \neq \mathsf{pos}(\hat{\mathbf{r}}^\mathsf{T}\tilde{\mathbf{X}} + \|\mathbf{\Gamma}\hat{\mathbf{r}}\|_2\hat{\zeta})]$$
$$= \Pr[\mathsf{pos}(\mathbf{a}^\mathsf{T}\mathbf{Z} + \zeta) \neq \mathsf{pos}(\hat{\mathbf{a}}^\mathsf{T}\mathbf{Z} + \hat{\zeta})] \qquad (16)$$

where $\mathbf{\Gamma} = \mathbf{\Sigma}^{1/2}$, $\mathbf{a} = \mathbf{\Gamma}\mathbf{r}/\|\mathbf{\Gamma}\mathbf{r}\|_2$ and $\hat{\mathbf{a}} = \mathbf{\Gamma}\hat{\mathbf{r}}/\|\mathbf{\Gamma}\hat{\mathbf{r}}\|_2$ and $\mathbf{Z} = \mathbf{\Gamma}^{-1}\tilde{\mathbf{X}} \sim N(\mathbf{0}, \mathbf{I})$ and $\tilde{\mathbf{X}} + \boldsymbol{\mu} = \mathbf{X} \sim N(\boldsymbol{\mu}, \mathbf{\Sigma})$, $\zeta = (c + \mathbf{r}^\mathsf{T}\boldsymbol{\mu})/\|\mathbf{\Gamma}\mathbf{r}\|_2$ and $\hat{\zeta} = (c + \hat{\mathbf{r}}^\mathsf{T}\boldsymbol{\mu})/\|\mathbf{\Gamma}\hat{\mathbf{r}}\|_2$. By Lemma 2.1, $\|\mathbf{a} - \hat{\mathbf{a}}\|_2 \leq 4\sqrt{\frac{\lambda_{\max}}{\lambda_{\min}}}\varepsilon$.

Now, the RHS of (16) can be bounded as,

$$\Pr[\mathsf{pos}(\mathbf{a}^\mathsf{T}\mathbf{Z} + \zeta) \neq \mathsf{pos}(\mathbf{a}^\mathsf{T}\mathbf{Z} + \hat{\zeta})] + \Pr[\mathsf{pos}(\mathbf{a}^\mathsf{T}\mathbf{Z} + \hat{\zeta}) \neq \mathsf{pos}(\hat{\mathbf{a}}^\mathsf{T}\mathbf{Z} + \hat{\zeta})] \qquad (17)$$

Now, $g := \mathbf{a}^\mathsf{T}\mathbf{Z} \sim N(0, 1)$. Thus, the first term in the above is bounded by the probability that $g$ lies in a range of length $|\zeta - \hat{\zeta}| = \|\boldsymbol{\mu}\|_2\|\mathbf{r} - \hat{\mathbf{r}}\|_2/\|\mathbf{\Gamma}\mathbf{r}\|_2 \leq \varepsilon\|\boldsymbol{\mu}\|_2/\sqrt{\lambda_{\min}}$. This probability is at most $\varepsilon\|\boldsymbol{\mu}\|_2/\sqrt{2\pi\lambda_{\min}}$.

For the second term, that is at most the probability that $\mathsf{pos}(\mathbf{a}^\mathsf{T}\mathbf{Z}) \neq \mathsf{pos}(\hat{\mathbf{a}}^\mathsf{T}\mathbf{Z})$. Now $\|\hat{\mathbf{a}} - \mathbf{a}\|_2 \leq 4\sqrt{\frac{\lambda_{\max}}{\lambda_{\min}}}\varepsilon \Rightarrow \angle\hat{\mathbf{a}}, \mathbf{a} \leq \pi 4\sqrt{\frac{\lambda_{\max}}{\lambda_{\min}}}\varepsilon \Rightarrow \Pr[\mathsf{pos}(\mathbf{a}^\mathsf{T}\mathbf{Z}) \neq \mathsf{pos}(\hat{\mathbf{a}}^\mathsf{T}\mathbf{Z})] \leq 2\sqrt{\frac{\lambda_{\max}}{\lambda_{\min}}}\varepsilon$. Hence,

$$\Pr[\mathsf{pos}(\mathbf{r}^\mathsf{T}\mathbf{X} + c) \neq \mathsf{pos}(\hat{\mathbf{r}}^\mathsf{T}\mathbf{X} + c)] \leq \varepsilon\left(2\sqrt{\frac{\lambda_{\max}}{\lambda_{\min}}} + \frac{\|\boldsymbol{\mu}\|_2}{\sqrt{2\pi\lambda_{\min}}}\right)$$

$\square$

# E Subgaussian concentration with thresholded Gaussian random variables

Let $\Phi(.)$ be the standard Gaussian cdf i.e., $\Phi(t) := \Pr_{X \sim N(0,1)}[X \geq t]$. We also define $\overline{\Phi}(t) := \Pr[X > t] = 1 - \Phi(t)$. We begin by defining the subgaussian norm of a random variable.

**Definition E.1.** *The subgaussian norm of a random variable $X$ denoted by $\|X\|_{\psi_2}$ and is defined as: $\|X\|_{\psi_2} := \inf\{t > 0 : \mathbb{E}\left[\exp\left(X^2/t^2\right)\right] \leq 2\}$. Further, there is an absolute constant $K_0$ such that $\|X\|_{\psi_2} \leq K_0 K$ if $X$ satisfies,*

$$\Pr\left[|X| \geq t\right] \leq 2\exp\left(-t^2/K^2\right), \qquad \text{for all } t \geq 0. \qquad (18)$$

Let $X \sim N(0, 1)$. It is easy to see that $\mathbb{E}\left[\exp(X^2/2^2)\right] = (1/\sqrt{2\pi})\int_{-\infty}^{\infty}\exp(-x^2/4)dx = \sqrt{2}(1/\sqrt{2\pi})\int_{-\infty}^{\infty}\exp(-z^2/2)dz = \sqrt{2}$. Thus, $X \sim N(0, 1)$ is subgaussian with subgaussian norm $\leq 2$. In the analysis of this section, we shall use the following proposition from [32].

**Proposition E.2** (Prop. 2.1.2 of [32]). *Let $X \sim N(0,1)$. Then, for any $t > 0$,*

$$\left(\frac{1}{t} - \frac{1}{t^3}\right) \frac{1}{\sqrt{2\pi}} \exp\left(-t^2/2\right) \leq \overline{\Phi}(t) \leq \frac{1}{t\sqrt{2\pi}} \exp\left(-t^2/2\right).$$

*In particular for $t \geq 1$,*

$$\overline{\Phi}(t) \leq \frac{1}{\sqrt{2\pi}} \exp\left(-t^2/2\right).$$

Using the above, and by symmetry it is easy to see that for $t \geq 1$, $\Pr[|X| > t] \leq \left(\sqrt{2/\pi}\right) \exp\left(-t^2/2\right) \leq 2 \cdot \exp\left(-t^2/2\right)$. On the other hand, $2 \cdot \exp\left(-t^2/2\right) \geq 1$ for $0 \leq t < 1$. Thus,

$$\Pr[|X| > t] \leq 2 \cdot \exp\left(-t^2/2\right), \quad \forall\, t \geq 0. \tag{19}$$

Consider the normal distribution conditioned on a threshold defined by letting $\mathcal{D}_\ell$ be the distribution of $\{X \sim N(0,1) \mid X > \ell\}$. We shall show that $\tilde{X} \sim \mathcal{D}_\ell$ is a subgaussian random variable. Let us handle the (relatively) easy case of $\ell \leq 0$ first.

**Lemma E.3.** *Let $\tilde{X} \sim \mathcal{D}_\ell$ for some $\ell \leq 0$. Then, $\Pr\left[|\tilde{X}| > t\right] \leq 2 \cdot \exp\left(-t^2/2\right)$, for any $t > 0$.*

*Proof.* Let $E$ be the event that $\tilde{X} \geq 0$. Conditioned on $E$, $\tilde{X}$ is distributed as $|X|$ where $X \sim N(0,1)$ and (19) implies that

$$\Pr\left[|\tilde{X}| > t \mid E\right] \leq 2 \cdot \exp\left(-t^2/2\right), \quad \forall\, t > 0.$$

On the other hand, conditioned on $\overline{E}$, $\tilde{X}$ is sampled as $-|Z|$ where $\{Z \sim N(0,1) \mid |Z| \leq -\ell = |\ell|\}$. Thus,

$$\Pr\left[|\tilde{X}| > t \mid \overline{E}\right] = \Pr_{Z \sim N(0,1)}[|Z| > t \mid |Z| \leq |\ell|]. \tag{20}$$

Now, if $t > |\ell|$ then $\Pr\left[|\tilde{X}| > t \mid \overline{E}\right] = 0$, otherwise if $t \leq \ell$ the LHS of (20) is,

$$\Pr_{Z \sim N(0,1)}[|Z| > t \mid |Z| \leq |\ell|] = 1 - \Pr[|Z| \leq t \mid |Z| \leq |\ell|] = 1 - \frac{\Pr[|Z| \leq t]}{\Pr[|Z| \leq \ell]}$$
$$\leq 1 - \Pr[|Z| \leq t] = \Pr[|Z| > t]$$

which is bounded by $2 \cdot \exp\left(-t^2/2\right)$ and combining the probability bounds conditioned on $E$ and $\overline{E}$ we complete the proof. $\qquad\square$

The case of $\ell > 0$ is proved below.

**Lemma E.4.** *Let $\tilde{X} \sim \mathcal{D}_\ell$ for some $\ell > 0$. Then, $\Pr\left[|\tilde{X}| > t\right] \leq 2 \cdot \exp\left(-t^2/K_1^2\right)$, for any $t > 0$, where $K_1 = \max\{\sqrt{20}, |\ell|\sqrt{10}\}$.*

*Proof.* Let us first explicitly define the pdf of $\tilde{X}$ as:

$$f_{\mathcal{D}_\ell}(x) = \begin{cases} 0 & \text{if } x \leq \ell \\ f_{N(0,1)}(x)/\overline{\Phi}(\ell) & \text{otherwise,} \end{cases} \tag{21}$$

where $f_{N(0,1)}$ is the pdf of $N(0,1)$. We prove this in two cases.

*Case $\ell \leq 2$.* In this case, one can use the lower bound from Prop. E.2 to show that $\overline{\Phi}(\ell) \geq \overline{\Phi}(2) > 1/50$ by explicit calculation. Thus, $f_{\mathcal{D}_\ell}(x) \leq 50 f_{N(0,1)}(x)$ for $x > 0$. We can now obtain the desired bound as follows. Letting $X \sim N(0,1)$, for any $t > 0$,

$$\begin{aligned} \Pr\left[\left|\tilde{X}\right| > t\right] &= \Pr\left[\tilde{X} > t\right] \\ &\leq 50 \Pr[X > t] \leq 50 \Pr[|X| > t] \leq 100 \exp\left(-t^2/2\right) \end{aligned} \tag{22}$$

using (19). Now, it is easy to check that for $t \geq 3$,

$$\frac{\exp\left(-\frac{t^2}{20}\right)}{\exp\left(-\frac{t^2}{2}\right)} \geq \exp\left(9 \cdot \left(\frac{1}{2} - \frac{1}{20}\right)\right) \geq \exp(4) > 50$$

$$\Rightarrow 2 \cdot \exp\left(-\frac{t^2}{20}\right) > 100 \cdot \exp\left(-\frac{t^2}{2}\right) \tag{23}$$

On the other hand, for $0 \leq t < 3$, $2\exp\left(-t^2/20\right) > 1$. Thus,

$$\Pr\left[\left|\tilde{X}\right| > t\right] \leq 2 \cdot \exp\left(-t^2/20\right)$$

*Case $\ell > 2$.* In this case using the easily verifiable facts that hold for $\ell > 2$:

- $(1/\ell - 1/\ell^3) > 1/(2\ell)$ and
- $\exp\left(-3\ell^2/2\right) \leq \frac{1}{2\ell\sqrt{2\pi}}$,

Prop. E.2 yields

$$\overline{\Phi}(\ell) \geq \left(\frac{1}{\ell} - \frac{1}{\ell^3}\right) \frac{1}{\sqrt{2\pi}}\exp\left(-\ell^2/2\right) \geq \frac{1}{2\ell\sqrt{2\pi}}\exp\left(-\ell^2/2\right) \geq \exp\left(-2\ell^2\right). \tag{24}$$

Observe that $\Pr\left[\left|\tilde{X}\right| > t\right] = \Pr_{Z \sim N(0,1)}\left[Z > t \mid Z \geq \ell\right]$. If $t < \ell$, then this probability vanishes. Otherwise $t \geq \ell > 2$, and from Prop. E.2, $\Pr[Z > t] \leq \exp(-t^2/2)$ and therefore $\Pr\left[\left|\tilde{X}\right| > t\right]$ can be bounded by

$$\leq \frac{\Pr[Z > t]}{\Pr[Z \geq \ell]} \leq 2\overline{\Phi}(\ell)^{-1}\exp\left(-t^2/2\right) \leq 2\exp\left(-t^2/2 + 2\ell^2\right) \tag{25}$$

using (24). Now, if $t^2 = 5\ell^2 + \kappa$ for some $\kappa \geq 0$, then using $\ell > 2$ we have

$$-\frac{t^2}{2} + 2\ell^2 = -\frac{\ell^2 + \kappa}{2} \leq -1 - \frac{\kappa}{5\ell^2} = -\frac{5\ell^2 + \kappa}{5\ell^2} \leq -\frac{t^2}{5\ell^2} \leq -\frac{t^2}{10\ell^2}. \tag{26}$$

Therefore, $2\exp\left(-t^2/2 + 2\ell^2\right) \leq 2\exp\left(-t^2/(10\ell^2)\right)$ for $t^2 \geq 5\ell^2$. On the other hand,

$$2\exp\left(-t^2/(10\ell^2)\right) > 2e^{-1/2} > 1, \qquad \text{when } t^2 < 5\ell^2.$$

Thus, in this case the following holds for all $t > 0$:

$$\Pr\left[\left|\tilde{X}\right| > t\right] \leq 2\exp\left(-t^2/(10\ell^2)\right) \tag{27}$$

completing the proof. $\qquad\square$

The above results also apply to "complements" of the thresholded Gaussians. In particular, let $\overline{\mathcal{D}}_\ell$ be the distribution of $\{X \sim N(0,1) \mid X \leq \ell \Leftrightarrow -X \geq -\ell\}$ which is equivalently $\{-X \sim N(0,1) \mid X \geq -\ell\}$ to which the above analysis can be directly be applied. This yields, that if $\tilde{X} \sim \overline{\mathcal{D}}_\ell$ then for any $t > 0$,

$$\Pr\left[|\tilde{X}| > t\right] \leq 2 \cdot \exp\left(-t^2/2\right), \qquad\qquad \text{if } \ell \geq 0, \tag{28}$$

$$\Pr\left[|\tilde{X}| > t\right] \leq 2 \cdot \exp\left(-t^2/K_1^2\right), \qquad\qquad \text{if } \ell < 0, \tag{29}$$

where $K_1 = \max\{\sqrt{20}, |\ell|\sqrt{10}\}$.

### E.1 Bag mean and convariance estimation error bounds

In this section we shall be concerned with random variables $\tilde{X}$ which are sampled from $\mathcal{D}_\ell$ with probability $p$ and from $\overline{\mathcal{D}}_\ell$ with probability $(1 - p)$. Let us denote this distribution by $\hat{\mathcal{D}}(p, \ell)$. Using Lemmas E.3, E.4 and (28), (29), we obtain the following lemma.

**Lemma E.5.** *Let $\tilde{X} \sim \hat{\mathcal{D}}(p, \ell)$ for some $p \in (0, 1)$. Then for any $t > 0$,*

$$\Pr\left[|\tilde{X}| > t\right] \leq 2 \cdot \exp\left(-t^2/K_1^2\right) \tag{30}$$

*where $K_1 = \max\{\sqrt{20}, |\ell|\sqrt{10}\}$. In particular, $\|\tilde{X}\|_{\psi_2} = O(|\ell|)$.*

We however, shall also require similar bounds for the mean-zero version of such distributions. To begin with we state an easy lemma bounding the mean of $\tilde{X}$.

**Lemma E.6.** *Let $\tilde{X} \sim \hat{\mathcal{D}}(p, \ell)$ for some $p \in (0, 1)$. Then, $\left|\mathbb{E}\left[\tilde{X}\right]\right| \leq \gamma_\ell = \max\{\gamma_0, 2\ell\}$, where $\gamma_0 > 0$ is some constant.*

*Proof.* Let us consider the case of $\ell > 0$ (and $\ell \leq 0$ follows analogously). When $\ell < 2$, it is easy to see that desired expectation is $O(1)$. Further, the expectation over $\overline{\mathcal{D}}_\ell$ is $O(\ell)$ for any $\ell > 0$, since it is a convex combination of the expectation of a half gaussian which has $O(1)$ expectation, and a gaussian truncated from below at $0$ and above at $\ell$, which has $O(\ell)$ expectation. To complete the argument we need to bound the expectation over $\mathcal{D}_\ell$. Using (21) and $(1/\sqrt{2\pi}) \int_\ell^\infty x \exp(-x^2/2)\, dx = (1/\sqrt{2\pi})\exp(-\ell^2/2)$, we obtain $\mathbb{E}_{X \sim \mathcal{D}_\ell}[X] = (1/\sqrt{2\pi})\exp(-\ell^2/2)\overline{\Phi}(\ell)^{-1}$ and the lower bound from Prop. E.2 along with $\ell \geq 2$ yields an upper bound of $2\ell$ on the expectation. $\qquad\square$

With the setup as in Lemma E.5 define $\hat{X} := \tilde{X} - \mathbb{E}\left[\tilde{X}\right]$. Clearly, $\mathbb{E}\left[\hat{X}\right] = 0$. Further,

$$\left|\hat{X}\right| > t \Rightarrow \left|\tilde{X}\right| + \left|\mathbb{E}\left[\tilde{X}\right]\right| > t \Rightarrow \left|\tilde{X}\right| > t - \gamma_\ell.$$

Therefore,

$$\Pr\left[|\hat{X}| > t\right] \leq \Pr\left[|\tilde{X}| > t - \gamma_\ell\right] \leq 2 \cdot \exp\left(-(t - \gamma_\ell)^2/K_1^2\right) \tag{31}$$

Let $K_2 = \max\{2K_1, \sqrt{2}\gamma_\ell\}$. Now,

$$t \geq 2\gamma_\ell \Rightarrow |t| \leq 2|t - \gamma_\ell| \quad \Rightarrow \quad \frac{t^2}{4K_1^2} \leq \frac{(t - \gamma_\ell)^2}{K_1^2}$$
$$\Rightarrow \quad 2 \cdot \exp\left(-(t - \gamma_\ell)^2/K_1^2\right) \leq 2 \cdot \exp\left(-t^2/K_2^2\right) \tag{32}$$

On the other hand, when $0 \leq t < 2\gamma_\ell$, $t^2/K_2^2 \geq 1/2$, and thus

$$2 \cdot \exp\left(-t^2/K_2^2\right) \geq 2e^{-1/2} > 1.$$

Thus,

$$\Pr\left[|\hat{X}| > t\right] \leq 2 \cdot \exp\left(-t^2/K_2^2\right) \tag{33}$$

for all $t > 0$, where $K_2 = O(|\ell|)$.

### E.1.1 Concentration of mean estimate using Hoeffding's Bound

Let us first state the Hoeffding's concentration bound for subgaussian random variables.

**Theorem E.7** (Theorem 2.6.2 of [32])**.** *Let $X_1, \ldots, X_N$ be independent, mean-zero, sub-gaussian random variables. Then, for every $\varepsilon \geq 0$,*

$$\Pr\left[\left|\frac{\sum_{i=1}^N X_i}{N}\right| \geq \varepsilon\right] \leq 2 \cdot \exp\left(\frac{-c\varepsilon^2 N^2}{\sum_{i=1}^N \|X_i\|_{\psi_2}^2}\right), \tag{34}$$

*where $c > 0$ is some absolute constant.*

For the rest of this section we shall fix $\ell \in \mathbb{R}$ and $p \in (0, 1)$. Consider vector valued random variable $\mathbf{X} = (X^{(1)}, \ldots, X^{(d)})$ with independent coordinates where

- $X^{(1)} = \tilde{X} - \mathbb{E}[\tilde{X}]$ where $\tilde{X} \sim \hat{\mathcal{D}}(p, \ell)$. From the previous subsection, we have $\|X^{(1)}\|_{\psi_2} = O(|\ell|)$.

- For $i = 2, \ldots, d$, $X^{(i)} \sim N(0, 1)$ and therefore $\|X^{(i)}\|_{\psi_2} = O(1)$.

Using the above bounds on the subgaussian norms, and applying Theorem E.7 to bound the error in each coordinate by $\varepsilon/\sqrt{d}$ and taking a union bound we obtain the following lemma.

**Lemma E.8.** *Let $\mathbf{X}_1, \ldots, \mathbf{X}_N$ be $N$ iid samples of $\mathbf{X}$. Then for every $\varepsilon \geq 0$,*

$$\Pr\left[\left\|\frac{\sum_{i=1}^{N}\mathbf{X}_i}{N}\right\|_2 \geq \varepsilon\right] \leq 2 \cdot \exp\left(\frac{-c_0 \varepsilon^2 N}{dO(\ell^2)}\right) + 2(d-1) \cdot \exp\left(-c_0 \varepsilon^2 N/d\right) \tag{35}$$

*for some absolute constant $c_0 > 0$. In particular, if $N > O\left((d/\varepsilon^2)O(\ell^2)\log(d/\delta)\right)$,*

$$\Pr\left[\left\|\frac{\sum_{i=1}^{N}\mathbf{X}_i}{N}\right\|_2 \geq \varepsilon\right] \leq \delta,$$

*for any $\delta > 0$.*

### E.1.2 Concentration of covariance estimate

Consider the vector random variable $\mathbf{X}$ defined in the previous subsection. It is mean-zero and so by Defn. 3.4.1 and Lemma 3.4.2 of [32],

$$\sup_{\mathbf{x} \in S^{d-1}} \|\langle \mathbf{x}, \mathbf{X} \rangle\|_{\psi_2} = O(\ell), \tag{36}$$

using the bounds on the subgaussian norms of the coordinates of $\mathbf{X}$ given in the previous subsection. Using this we can directly apply Proposition 2.1 of [31] to obtain the following lemma.

**Lemma E.9.** *Let $\mathbf{X}_1, \ldots, \mathbf{X}_N$ be $N$ iid samples of $\mathbf{X}$, then if $N > O\left((d/\varepsilon^2)O(\ell^2)\log(1/\delta)\right)$,*

$$\Pr\left[\left\|\frac{\sum_{i=1}^{N}\mathbf{X}_i\mathbf{X}_i^\mathsf{T}}{N} - \mathbb{E}\left[\mathbf{X}\mathbf{X}^\mathsf{T}\right]\right\|_2 \geq \varepsilon\right] \leq \delta, \tag{37}$$

*for any $\varepsilon, \delta > 0$.*

### E.1.3 Mean and covariance estimate bounds for non-centered vector r.v.s

**Distribution $\mathcal{D}_{\text{asymvec}}(p, \ell)$.** We revisit the definition of $\mathbf{X}$ in Sec. E.1.1 and instead define distribution $\mathcal{D}_{\text{asymvec}}(p, \ell)$ over $\mathbf{Z} = (Z^{(1)}, \ldots, Z^{(d)})$ with independent coordinates by taking $Z^{(1)} = \tilde{X}$ where $\tilde{X} \sim \hat{\mathcal{D}}(p, \ell)$ and for $i = 2, \ldots, d$, $Z^{(i)} \sim N(0, 1)$.

Clearly, $\mathbf{X} = \mathbf{Z} - \mathbb{E}[\mathbf{Z}]$. For convenience, we shall use the following notation:

$$\boldsymbol{\mu}_Z := \mathbb{E}[\mathbf{Z}] \qquad \text{and,} \qquad \hat{\boldsymbol{\mu}}_Z := \frac{\sum_{i=1}^{N}\mathbf{Z}_i}{N}, \tag{38}$$

$$\boldsymbol{\Sigma}_Z := \mathbb{E}[(\mathbf{Z} - \boldsymbol{\mu}_Z)(\mathbf{Z} - \boldsymbol{\mu}_Z)^\mathsf{T}] \qquad \text{and,} \qquad \hat{\boldsymbol{\Sigma}}_Z := \frac{\sum_{i=1}^{N}(\mathbf{Z}_i - \hat{\boldsymbol{\mu}}_Z)(\mathbf{Z}_i - \hat{\boldsymbol{\mu}}_Z)^\mathsf{T}}{N}, \tag{39}$$

where $\mathbf{Z}_i$ is an iid sample of $\mathbf{Z}$ and $\mathbf{X}_i = \mathbf{Z}_i - \boldsymbol{\mu}_Z$, for $i = 1, \ldots, N$. We have the following lemma.

**Lemma E.10.** *For any $\varepsilon, \delta \in (0, 1)$, if $N > O\left((d/\varepsilon^2)O(\ell^2)\log(d/\delta)\right)$, then w.p. $1 - \delta$ the following hold simultaneously,*

$$\|\hat{\boldsymbol{\mu}}_Z - \boldsymbol{\mu}_Z\|_2 \leq \varepsilon/2, \tag{40}$$

$$\|\hat{\boldsymbol{\Sigma}}_Z - \boldsymbol{\Sigma}_Z\|_2 \leq \varepsilon \tag{41}$$

*Proof.* We begin by applying Lemmas E.8 and E.9 to $\mathbf{X}$ and the iid samples $\mathbf{X}_1, \ldots, \mathbf{X}_N$ so that their conditions hold with $\varepsilon/2$ and $\delta/2$. Taking a union bound we get that the following simultaneously hold with probability at least $1 - \delta$:

$$\left\| \frac{\sum_{i=1}^{N} \mathbf{X}_i}{N} \right\|_2 \leq \varepsilon/2, \tag{42}$$

$$\left\| \frac{\sum_{i=1}^{N} \mathbf{X}_i \mathbf{X}_i^{\mathsf{T}}}{N} - \mathbb{E}\left[\mathbf{X}\mathbf{X}^{\mathsf{T}}\right] \right\|_2 \leq \varepsilon/2. \tag{43}$$

By the definitions above, (42) directly implies (40).

Now, observe that $\boldsymbol{\Sigma}_Z = \mathbb{E}[\mathbf{X}\mathbf{X}^{\mathsf{T}}]$. On the other hand, letting $\boldsymbol{\zeta} := \hat{\boldsymbol{\mu}}_Z - \boldsymbol{\mu}_Z = (\sum_{i=1}^{N} \mathbf{X}_i)/N$ we simplify $\hat{\boldsymbol{\Sigma}}_Z$ as,

$$
\begin{aligned}
\frac{\sum_{i=1}^{N} (\mathbf{Z}_i - \hat{\boldsymbol{\mu}}_Z)(\mathbf{Z}_i - \hat{\boldsymbol{\mu}}_Z)^{\mathsf{T}}}{N} &= \frac{\sum_{i=1}^{N} (\mathbf{X}_i - \boldsymbol{\zeta})(\mathbf{X}_i - \boldsymbol{\zeta})^{\mathsf{T}}}{N} \\
&= \frac{\sum_{i=1}^{N} \mathbf{X}_i \mathbf{X}_i^{\mathsf{T}} - \mathbf{X}_i \boldsymbol{\zeta}^{\mathsf{T}} - \boldsymbol{\zeta} \mathbf{X}_i^{\mathsf{T}} + \boldsymbol{\zeta}\boldsymbol{\zeta}^{\mathsf{T}}}{N} \\
&= \frac{\sum_{i=1}^{N} \mathbf{X}_i \mathbf{X}_i^{\mathsf{T}}}{N} - 2\boldsymbol{\zeta}\boldsymbol{\zeta}^{\mathsf{T}} + \boldsymbol{\zeta}\boldsymbol{\zeta}^{\mathsf{T}} \\
&= \frac{\sum_{i=1}^{N} \mathbf{X}_i \mathbf{X}_i^{\mathsf{T}}}{N} - \boldsymbol{\zeta}\boldsymbol{\zeta}^{\mathsf{T}} \tag{44}
\end{aligned}
$$

Thus, the LHS of (41) is at most,

$$
\begin{aligned}
\left\| \frac{\sum_{i=1}^{N} \mathbf{X}_i \mathbf{X}_i^{\mathsf{T}}}{N} - \mathbb{E}\left[\mathbf{X}\mathbf{X}^{\mathsf{T}}\right] - \boldsymbol{\zeta}\boldsymbol{\zeta}^{\mathsf{T}} \right\|_2 &\leq \left\| \frac{\sum_{i=1}^{N} \mathbf{X}_i \mathbf{X}_i^{\mathsf{T}}}{N} - \mathbb{E}\left[\mathbf{X}\mathbf{X}^{\mathsf{T}}\right] \right\|_2 + \left\| \boldsymbol{\zeta}\boldsymbol{\zeta}^{\mathsf{T}} \right\|_2 \\
&\leq \varepsilon/2 + \varepsilon^2/4 \leq \varepsilon, \tag{45}
\end{aligned}
$$

since we have shown that $\|\boldsymbol{\zeta}\|_2 = \|\hat{\boldsymbol{\mu}}_Z - \boldsymbol{\mu}_Z\|_2 \leq \varepsilon/2$. □

Finally, we prove a version of the above lemma under a symmteric psd transformation. Let $\mathbf{A}$ be a psd matrix s.t. $\lambda_{\max}$ is the maximum eigenvalue of $\mathbf{A}^2 = \mathbf{A}\mathbf{A}$ i.e., $\sqrt{\lambda_{\max}}$ is the maximum eigenvalue of $\mathbf{A}$. Then, if we define $\tilde{\mathbf{Z}} := \mathbf{A}\mathbf{Z}$ and $\tilde{\mathbf{Z}}_i$ as iid samples of $\tilde{\mathbf{Z}}$, $i = 1, \ldots, N$ and analogous to (38) and (39), define $\boldsymbol{\mu}_{\tilde{Z}}$, $\hat{\boldsymbol{\mu}}_{\tilde{Z}}$, $\boldsymbol{\Sigma}_{\tilde{Z}}$ and $\hat{\boldsymbol{\Sigma}}_{\tilde{Z}}$, we have the following lemma which follows directly from Lemma E.10 the $\sqrt{\lambda_{\max}}$ upper bound on the operator norm of $\mathbf{A}$.

**Lemma E.11.** *For any $\varepsilon, \delta \in (0, 1)$, if $N > O\left((d/\varepsilon^2)O(\ell^2)\log(d/\delta)\right)$, then w.p. $1 - \delta$ the following hold simultaneously,*

$$\|\hat{\boldsymbol{\mu}}_{\tilde{Z}} - \boldsymbol{\mu}_{\tilde{Z}}\|_2 \leq \varepsilon\sqrt{\lambda_{\max}}/2, \tag{46}$$

$$\|\hat{\boldsymbol{\Sigma}}_{\tilde{Z}} - \boldsymbol{\Sigma}_{\tilde{Z}}\|_2 \leq \varepsilon\lambda_{\max} \tag{47}$$

### E.2 Estimating Covariance of differences

First we begin this subsection with a simple observation. If $X$ and $Y$ are two random variables such that ,

$$\Pr\left[|X| > t\right], \Pr\left[|Y| > t\right] \leq 2 \cdot \exp\left(-t^2/K_1^2\right), \quad \forall t > 0, \tag{48}$$

then $X - Y$ is a random variable such that,

$$\Pr\left[|X - Y| > t\right] \leq \Pr\left[|X| > t/2\right] + \Pr\left[|Y| > t/2\right] \leq 4 \cdot \exp\left(-t^2/(2K_1)^2\right).$$

It is easy to see that,

$$4 \cdot \exp\left(-t^2/(2K_1)^2\right) \leq 2 \cdot \exp\left(-t^2/(4K_1)^2\right),$$

when $t^2 \geq 8K_1^2$. On the other hand, when $t^2 < 8K_1^2$, $2 \cdot \exp\left(-t^2/(4K_1)^2\right) > 1$. Thus,

$$\Pr\left[|X - Y| > t\right] \leq 2 \cdot \exp\left(-t^2/(4K_1)^2\right) \tag{49}$$

In this subsection shall consider $\tilde{X} \sim \mathcal{D}(p, q, \ell)$ to be defined as follows, for some $\ell \in \mathbb{R}$, $p, q \in [0, 1)$ s.t. $p + q < 1$. $\tilde{X} = U - V$ where:

- with probability $p$, $U \sim \mathcal{D}_\ell$ and $V \sim \mathcal{D}_\ell$ independently,
- with probability $q$, $U \sim \overline{\mathcal{D}}_\ell$ and $V \sim \overline{\mathcal{D}}_\ell$ independently,
- with probability $(1 - p - q)/2$, $U \sim \mathcal{D}_\ell$ and $V \sim \overline{\mathcal{D}}_\ell$ independently,
- with probability $(1 - p - q)/2$, $U \sim \overline{\mathcal{D}}_\ell$ and $V \sim \mathcal{D}_\ell$ independently.

From the above it is clear that $\mathbb{E}[\tilde{X}] = 0$. Further, from (49) and Lemmas E.3, E.4 and (28), (29),

$$\Pr\left[|\tilde{X}| > t\right] \leq 2 \cdot \exp\left(-t^2/K_1^2\right) \tag{50}$$

for $t > 0$, where $K_1 = \max\{4\sqrt{20}, 4|\ell|\sqrt{10}\}$. In particular, $\|\tilde{X}\|_{\psi_2} = O(\ell)$.

Let us now define a distribution $\mathcal{D}_{\text{diffvec}}(p, q, \ell)$ vector valued random variable with independent coordinates $\mathbf{X} = (X^{(1)}, \ldots, X^{(d)})$, where

- $X^{(1)} \sim \mathcal{D}(p, q, \ell)$, for some $\ell \in \mathbb{R}$, $p, q \in [0, 1)$ s.t. $p + q < 1$.
- $\mathbf{X}^{(j)}$ is the difference of two iid $N(0, 1)$ random variables, for $j = 2, \ldots, d$. In particular, the subgaussian norm of these coordinates is $O(1)$.

From, the above it is clear that $\mathbb{E}[\mathbf{X}] = \mathbf{0}$, and by Defn. 3.4.1 and Lemma 3.4.2 of [32], (36) is applicable to $\mathbf{X}$ as defined above. Thus, letting $\hat{\mathbf{X}} := \mathbf{A}\mathbf{X}$, where $\mathbf{A}$ is as used in the previous subsection we have,

**Lemma E.12.** *For $\mathbf{X}$ defined above, the statement of Lemma E.9 is applicable, and (37) implies that the following holds:*

$$\left\|\frac{\sum_{i=1}^{N} \hat{\mathbf{X}}_i \hat{\mathbf{X}}_i^\top}{N} - \mathbb{E}\left[\hat{\mathbf{X}}\hat{\mathbf{X}}^\top\right]\right\|_2 \leq \varepsilon \lambda_{\max} \tag{51}$$

*with probability $1 - \delta$.*

### E.3   Proof of Lemma 3.1

Our proof shall utilize the following normalization of LTFs in a Gaussian space.

**Lemma E.13.** *Suppose $f(\mathbf{X}) = \mathsf{pos}\left(\mathbf{r}^\top \mathbf{X} + c\right)$, where $\|\mathbf{r}\|_2 > 0$ and $\mathbf{X} \sim N(\boldsymbol{\mu}, \boldsymbol{\Sigma})$ s.t. $\boldsymbol{\Sigma}$ is positive definite. Let $\boldsymbol{\Gamma} := \boldsymbol{\Sigma}^{1/2}$ be symmetric p.d., and $\mathbf{U}$ be any orthonormal transformation satisfying $\mathbf{U}\boldsymbol{\Gamma}\mathbf{r}/\|\boldsymbol{\Gamma}\mathbf{r}\|_2 = \mathbf{e}_1$ where $\mathbf{e}_1$ is the vector with 1 in the first coordinate and 0 in the rest. Then, letting $\mathbf{Z} \sim N(\mathbf{0}, \mathbf{I})$ so that $\mathbf{X} = \boldsymbol{\Gamma}\mathbf{U}^\top \mathbf{Z} + \boldsymbol{\mu}$,*

$$f(\mathbf{X}) = \mathsf{pos}\left(\mathbf{r}^\top \mathbf{X} + c\right) = \mathsf{pos}\left(\mathbf{e}_1^\top \mathbf{Z} + \ell\right) \tag{52}$$

*where $\ell = \left(\mathbf{r}^\top \boldsymbol{\mu} + c\right)/\|\boldsymbol{\Gamma}\mathbf{r}\|_2$.*

*Proof.* We have $\mathbf{X} = \hat{\mathbf{X}} + \boldsymbol{\mu}$ where $\hat{\mathbf{X}} \sim N(\mathbf{0}, \boldsymbol{\Sigma})$. Thus, $\hat{\mathbf{X}} = \boldsymbol{\Gamma}\mathbf{U}^\top \mathbf{Z} \Rightarrow \mathbf{Z} = \mathbf{U}\boldsymbol{\Gamma}^{-1}\hat{\mathbf{X}}$, using $\mathbf{U}^\top = \mathbf{U}^{-1}$. Now, $f(\mathbf{X})$ can be written as

$$\mathsf{pos}\left(\frac{\mathbf{r}^\top(\hat{\mathbf{X}} + \boldsymbol{\mu}) + c}{\|\boldsymbol{\Gamma}\mathbf{r}\|_2}\right) = \mathsf{pos}\left(\frac{\mathbf{r}^\top \boldsymbol{\Gamma}\mathbf{U}^\top \mathbf{U}\boldsymbol{\Gamma}^{-1}\hat{\mathbf{X}} + \mathbf{r}^\top\boldsymbol{\mu} + c}{\|\boldsymbol{\Gamma}\mathbf{r}\|_2}\right)$$

$$= \mathsf{pos}\left(\frac{(\mathbf{U}\boldsymbol{\Gamma}\mathbf{r})^\top \mathbf{Z}}{\|\boldsymbol{\Gamma}\mathbf{r}\|_2} + \frac{\mathbf{r}^\top\boldsymbol{\mu} + c}{\|\boldsymbol{\Gamma}\mathbf{r}\|_2}\right)$$

$$= \mathsf{pos}\left(\mathbf{e}_1^\top \mathbf{Z} + \frac{\mathbf{r}^\top\boldsymbol{\mu} + c}{\|\boldsymbol{\Gamma}\mathbf{r}\|_2}\right) \tag{53}$$

which completes the proof. □

*Proof.* (of Lemma 3.1) Using the normalization in Lemma E.13, we can write $\mathbf{X} = \mathbf{A}\mathbf{Z} + \boldsymbol{\mu}$ where $\mathbf{A} = \boldsymbol{\Sigma}^{1/2}\mathbf{U}^\top$ such that $\mathbf{X} \sim N(\boldsymbol{\mu}, \boldsymbol{\Sigma}) \equiv \mathbf{Z} \sim N(\mathbf{0}, \mathbf{I})$. From the condition in (52) we can write the samples in Step 2 of Alg. 1 as $\mathbf{x}_i = \mathbf{A}\mathbf{z}_i + \boldsymbol{\mu}$ where $\mathbf{z}_i$ are sampled from $\mathcal{D}_{\text{asymvec}}(k/q, -\ell)$ (see Sec.

E.1.3) for $\ell$ as given in Lemma E.13. Note that the maximum eigenvalue of $\mathbf{A}^2$ is $\lambda_{\max}$ which is the maximum eigenvalue of $\boldsymbol{\Sigma}$. Thus, one can apply Lemma E.11 to $\hat{\boldsymbol{\mu}}_B$ and $\hat{\boldsymbol{\Sigma}}_B$.

Further, the difference vectors sampled in Step 6 can be written as $\overline{\mathbf{x}}_i = \mathbf{A}\overline{\mathbf{z}}_i$ where $\overline{\mathbf{z}}_i$ are sampled from $\mathcal{D}_{\text{diffvec}}(p, p', -\ell)$ (see Sec. E.2) where $p = \binom{k}{2}/\binom{q}{2}$ is the probability of sampling a pair of 1-labeled feature-vectors from a bag, and $p' = \binom{q-k}{2}/\binom{q}{2}$ is that of sampling a pair of 0-labeled feature-vectors. Thus, one can apply Lemma E.12 to $\hat{\boldsymbol{\Sigma}}_D$.

Using both the above applications with the error probability $\delta/2$ and using union bound we complete the proof. $\qquad\square$

# F    Experimental Details and Results

## F.1    Implementation Details

The implementations of the algorithms in this paper (Algs. 2, 3, 4) and of the random LTF algorithm are in python using numpy libraries. The code for the SDP algorithms of [25, 26] for bag sizes 2 and 3 is adapted from the publicly available codebase[1]. The experimental code was executed on a 16-core CPU and 128 GB RAM machine running linux in a standard python environment.

## F.2    Experimental Results

In the following $d$ denotes the dimension of the feature-vectors, $q$ the size of the bags, with $k/q \in (0,1)$ the bag label proportion, and $m$ be the number of sampled bags in the training dataset. The instance-level test set is of size 1000 in all the experiments, and the reported metric is the accuracy over the test set.

**Standard Gaussian without LTF offset.** Here we primarily wish to evaluate the Algorithm 3 using unbalanced bag oracles such that $k \neq q/2$. For $d \in \{10, 50\}$, $(q,k) \in \{(3,1), (10,8), (50,35)\}$ and $m \in \{100, 500, 2000\}$ we create 25 datasets. In each LLP dataset, we (i) sample a random unit vector $\mathbf{r}^*$ and let $f(\mathbf{x}) := \text{pos}\left(\mathbf{r}^{*\mathsf{T}}\mathbf{x}\right)$, (ii) sample $m$ training bags from $\text{Ex}(f, N(\mathbf{0}, \mathbf{I}), q, k)$, (iii) sample 1000 test instances $(\mathbf{x}, f(\mathbf{x}))$, $\mathbf{x} \leftarrow N(\mathbf{0}, \mathbf{I})$. We also evaluate the Algorithm 2 on these datasets. For comparison we have the random LTF algorithm (R) in which we sample 100 random LTFs and return the one that performs best on the training set. The results are reported in Table 5. We also evaluate the SDP algorithm (S) in [26] for $(q,k) = (3,1)$ using $m \in \{50, 100, 200\}$ (since the SDP algorithms do not scale to larger number of bags) whose results are reported in Table 6.

Table 2 reports a concise set of comparative scores of Algorithms 2, 3 and random LTF (R) using 2000 bags and of the SDP algorithms (S) with 200 bags.

**Centered and general Gaussian.** Here we evaluate Algorithms 2 and 4, and we have both balanced as well as unbalanced bag oracles. In particular, for $d \in \{10, 50\}$, $(q,k) \in \{(2,1), (3,1), (10,5), (10,8), (50,25), (50,35)\}$ and $m \in \{100, 500, 2000\}$ we create 25 datasets similar to the previous case, except that for each dataset we first sample $\boldsymbol{\mu}$ and $\boldsymbol{\Sigma}$ and use $N(\mathbf{0}, \boldsymbol{\Sigma})$ for sampling feature-vectors in the centered Gaussian case and use $N(\boldsymbol{\mu}, \boldsymbol{\Sigma})$ for sampling feature-vectors in the general Gaussian case. We perform the following set of experiments in each case. For the cases when bags are balanced, i.e. $(q,k) \in \{(2,1), (10,5), (50,25)\}$, for each our Algorithms 2 and 4 we evaluate their two possible solutions on the test data and report the better number.

- **With LTF offset.** We sample $(\mathbf{r}_*, c_*)$ and create a dataset using $\text{pos}(\mathbf{r}_*^{\mathsf{T}}\mathbf{x} + c_*)$ as the labeling function. Table 11 reports the test accuracy scores for Algorithm 4 and random LTF (R) with $m = \{100, 500, 2000\}$ for centered and general Gaussians. Table 12 reports the corresponding scores for the SDP algorithm (S) [25, 26] with $m = \{50, 100, 200\}$ and $(q,k) \in \{(2,1), (3,1)\}$. Table 4 provides concise comparative scores with $m = 2000$ for Algorithm 4 and random LTF and $m = 200$ for the SDP algorithm (S).

- **Without LTF offset.** We sample an $\mathbf{r}_*$ and create a dataset using $\text{pos}(\mathbf{r}_*^{\mathsf{T}}\mathbf{x})$ as the labeling function. Table 7 reports the test accuracy scores for Algorithm 2 and random LTF (R)

---

[1]https://github.com/google-research/google-research/tree/master/Algorithms_and_Hardness_for_Learning_Linear_Thresholds_from_Label_Proportions (license included in the repository)

for centered and general Gaussians. Table 12 reports the scores for the SDP algorithm (S) [25, 26] on $(q, k) \in \{(2, 1), (3, 1)\}$ with $m = \{50, 100, 200\}$. Table 3 provides concise comparative scores with $m = 2000$ for Algorithm 4 and random LTF and $m = 200$ for the SDP algorithm (S).

*Noisy Labels.* We also experiment in a model with label noise. Here, the label of any instance can be independently flipped with some probability $p$, as a result the *true* bag label sum $k^*$ has distribution over $\{0, \dots, q\}$. In this case the SDP algorithms are not applicable and we omit them. Tables 9 and 10 give the test accuracy scores for Algorithm 2 and rand. LTF (R) with label flip probability $p = \{0.1, 0.25, 0.5\}$ for centered and general Gaussians. Like the balanced case, here also we evaluate both the solutions of Algorithm 2 on the test data and report the better number.

We observe that our algorithms perform significantly better in terms of accuracy than the comparative methods in all the bag distribution settings. Further, our algorithms have much lower error bounds on their accuracy scores. For the standard gaussian case, Algorithm 2 outperforms Algorithm 3 for larger $m$, possibly since with larger number of bags Algorithm 2 (which has higher sample complexity) can perform to its full potential. Conversely, we can observe that with larger bag sizes and dimensions, Algorithm 3 outperforms Algorithm 2 for smaller $m$.

For the noisy cases, from Tables 9 and 10 we observe that while the test accuracy degrades with large noise, it is fairly robust to small amounts of noise. This robustness is intuitive and we provide an explanation for the same in Appendix H (Lemma H.2).

Table 2: Comparison of Algorithms A3, A2, rand. LTF (R) and SDP (S) on $N(\mathbf{0}, \mathbf{I})$ feature-vectors.

| $d$ | $q$ | $k$ | A3 | A2 | R | S |
|-----|-----|-----|-------|-------|-------|-------|
| 10 | 3 | 1 | 95.52 | **98.17** | 76.0 | 68.04 |
| 10 | 10 | 8 | **98.58** | 97.68 | 75.46 | - |
| 10 | 50 | 35 | **99.06** | 97.95 | 74.18 | - |
| 50 | 3 | 1 | 89.26 | **95.15** | 59.86 | 57.55 |
| 50 | 10 | 8 | **96.73** | 94.36 | 61.58 | - |
| 50 | 50 | 35 | **97.82** | 94.94 | 61.04 | - |

Table 3: Comparision of Algorithm A2, rand. LTF (R) and SDP algorithms (S) without offset

| $d$ | $q$ | $k$ | A2 | R | S | | $d$ | $q$ | $k$ | A2 | R | S |
|-----|-----|-----|--------|-------|-------|---|-----|-----|-----|--------|-------|-------|
| 10 | 2 | 1 | **98.12** | 78.26 | 88.40 | | 10 | 2 | 1 | **98.18** | 78.32 | 90.10 |
| 10 | 3 | 1 | **98.27** | 77.16 | 67.31 | | 10 | 3 | 1 | **97.92** | 75.14 | 70.80 |
| 10 | 10 | 5 | **97.9** | 78.66 | - | | 10 | 10 | 5 | **97.86** | 70.41 | - |
| 10 | 10 | 8 | **97.87** | 77.64 | - | | 10 | 10 | 8 | **97.4** | 69.86 | - |
| 10 | 50 | 25 | **97.87** | 76.67 | - | | 10 | 50 | 25 | **97.57** | 70.48 | - |
| 10 | 50 | 35 | **97.9** | 77.17 | - | | 10 | 50 | 35 | **97.6** | 62.86 | - |
| 50 | 2 | 1 | **95.64** | 61.25 | 57.83 | | 50 | 2 | 1 | **94.99** | 58.68 | 61.12 |
| 50 | 3 | 1 | **95.21** | 61.15 | 58.69 | | 50 | 3 | 1 | **95.6** | 59.8 | 62.39 |
| 50 | 10 | 5 | **95.59** | 55.06 | - | | 50 | 10 | 5 | **95.27** | 57.43 | - |
| 50 | 10 | 8 | **94.34** | 63.17 | - | | 50 | 10 | 8 | **94.44** | 61.82 | - |
| 50 | 50 | 25 | **95.16** | 55.76 | - | | 50 | 50 | 25 | **94.97** | 53.98 | - |
| 50 | 50 | 35 | **94.74** | 61.02 | - | | 50 | 50 | 35 | **94.33** | 56.97 | - |

(a) $N(\mathbf{0}, \mathbf{\Sigma})$ feature-vectors.      (b) $N(\boldsymbol{\mu}, \mathbf{\Sigma})$ feature-vectors.

Table 4: Comparision of Algorithm A4, rand. LTF (R) and SDP algorithms (S) with offset

| d | q | k | A4 | R | S |
|---|---|---|---|---|---|
| 10 | 2 | 1 | **92.49** | 77.23 | 82.76 |
| 10 | 3 | 1 | **93.67** | 74.96 | 69.62 |
| 10 | 10 | 5 | **94.43** | 76.42 | - |
| 10 | 10 | 8 | **92.5** | 65.87 | - |
| 10 | 50 | 25 | **92.83** | 70.34 | - |
| 10 | 50 | 35 | **92.92** | 65.75 | - |
| 50 | 2 | 1 | **94.84** | 59.34 | 58.25 |
| 50 | 3 | 1 | **94.7** | 58.36 | 59.37 |
| 50 | 10 | 5 | **95.15** | 57.67 | - |
| 50 | 10 | 8 | **92.49** | 56.46 | - |
| 50 | 50 | 25 | **94.51** | 54.53 | - |
| 50 | 50 | 35 | **94.45** | 57.76 | - |

(a) $N(\mathbf{0}, \boldsymbol{\Sigma})$ feature-vectors.

| d | q | k | A4 | R | S |
|---|---|---|---|---|---|
| 10 | 2 | 1 | **93.08** | 78.24 | 88.01 |
| 10 | 3 | 1 | **94.62** | 71.49 | 67.57 |
| 10 | 10 | 5 | **93.1** | 70.59 | - |
| 10 | 10 | 8 | **94.47** | 67.85 | - |
| 10 | 50 | 25 | **93.93** | 65.03 | - |
| 10 | 50 | 35 | **93.31** | 66.04 | - |
| 50 | 2 | 1 | **94.17** | 59.66 | 61.01 |
| 50 | 3 | 1 | **94.41** | 59.92 | 64.24 |
| 50 | 10 | 5 | **94.06** | 56.12 | - |
| 50 | 10 | 8 | **92.3** | 59.67 | - |
| 50 | 50 | 25 | **93.41** | 55.72 | - |
| 50 | 50 | 35 | **94.08** | 56.97 | - |

(b) $N(\boldsymbol{\mu}, \boldsymbol{\Sigma})$ feature-vectors.

Table 5: Our algorithms A1 and A2 vs. rand. LTF (R) on $N(\mathbf{0}, \mathbf{I})$ feature-vectors.

| d | q | k | m | A3 | A2 | R |
|---|---|---|---|---|---|---|
| 10 | 3 | 1 | 100 | $82.37_{\pm0.13}$ | $90.2_{\pm0.07}$ | $73.46_{\pm0.68}$ |
| 10 | 3 | 1 | 500 | $91.2_{\pm0.03}$ | $96.34_{\pm0.01}$ | $73.26_{\pm0.34}$ |
| 10 | 3 | 1 | 2000 | $95.52_{\pm0.01}$ | $98.17_{\pm0.0}$ | $76.0_{\pm0.15}$ |
| 10 | 10 | 8 | 100 | $94.28_{\pm0.02}$ | $87.48_{\pm0.59}$ | $74.73_{\pm0.2}$ |
| 10 | 10 | 8 | 500 | $96.96_{\pm0.01}$ | $95.29_{\pm0.01}$ | $74.76_{\pm0.13}$ |
| 10 | 10 | 8 | 2000 | $98.58_{\pm0.0}$ | $97.68_{\pm0.0}$ | $75.46_{\pm0.12}$ |
| 10 | 50 | 35 | 100 | $95.79_{\pm0.01}$ | $89.99_{\pm0.07}$ | $73.34_{\pm0.5}$ |
| 10 | 50 | 35 | 500 | $98.26_{\pm0.0}$ | $95.63_{\pm0.03}$ | $72.83_{\pm0.22}$ |
| 10 | 50 | 35 | 2000 | $99.06_{\pm0.0}$ | $97.95_{\pm0.01}$ | $74.18_{\pm0.22}$ |
| 50 | 3 | 1 | 100 | $67.75_{\pm0.11}$ | $61.21_{\pm0.63}$ | $55.34_{\pm0.11}$ |
| 50 | 3 | 1 | 500 | $80.78_{\pm0.05}$ | $89.45_{\pm0.02}$ | $56.72_{\pm0.17}$ |
| 50 | 3 | 1 | 2000 | $89.26_{\pm0.02}$ | $95.15_{\pm0.01}$ | $59.86_{\pm0.09}$ |
| 50 | 10 | 8 | 100 | $85.7_{\pm0.02}$ | $60.4_{\pm0.53}$ | $56.26_{\pm0.12}$ |
| 50 | 10 | 8 | 500 | $94.09_{\pm0.01}$ | $86.84_{\pm0.04}$ | $60.37_{\pm0.11}$ |
| 50 | 10 | 8 | 2000 | $96.73_{\pm0.0}$ | $94.36_{\pm0.01}$ | $61.58_{\pm0.07}$ |
| 50 | 50 | 35 | 100 | $90.21_{\pm0.02}$ | $62.8_{\pm0.52}$ | $55.84_{\pm0.1}$ |
| 50 | 50 | 35 | 500 | $95.68_{\pm0.01}$ | $89.09_{\pm0.03}$ | $59.69_{\pm0.15}$ |
| 50 | 50 | 35 | 2000 | $97.82_{\pm0.0}$ | $94.94_{\pm0.01}$ | $61.04_{\pm0.14}$ |

Table 6: SDP Algorithm (S) on standard gaussian feature vectors

| d | m | S |
|---|---|---|
| 10 | 50 | $67.86_{\pm6.49}$ |
| 10 | 100 | $66.25_{\pm5.87}$ |
| 10 | 200 | $68.04_{\pm6.82}$ |
| 50 | 50 | $58.09_{\pm3.24}$ |
| 50 | 100 | $56.46_{\pm1.83}$ |
| 50 | 200 | $57.55_{\pm3.27}$ |

Table 7: Our algorithms A2 vs. rand. LTF (R) without offset

| $d$ | $q$ | $k$ | $m$ | A2 | R |
|---|---|---|---|---|---|
| 10 | 2 | 1 | 100 | $91.06_{\pm0.07}$ | $77.32_{\pm0.27}$ |
| 10 | 2 | 1 | 500 | $96.77_{\pm0.01}$ | $76.22_{\pm0.36}$ |
| 10 | 2 | 1 | 2000 | $98.12_{\pm0.01}$ | $78.26_{\pm0.12}$ |
| 10 | 3 | 1 | 100 | $91.2_{\pm0.1}$ | $74.25_{\pm0.34}$ |
| 10 | 3 | 1 | 500 | $96.2_{\pm0.01}$ | $76.12_{\pm0.29}$ |
| 10 | 3 | 1 | 2000 | $98.27_{\pm0.0}$ | $77.16_{\pm0.24}$ |
| 10 | 10 | 5 | 100 | $90.29_{\pm0.11}$ | $66.84_{\pm0.9}$ |
| 10 | 10 | 5 | 500 | $96.08_{\pm0.01}$ | $74.16_{\pm0.62}$ |
| 10 | 10 | 5 | 2000 | $97.9_{\pm0.01}$ | $78.66_{\pm0.17}$ |
| 10 | 10 | 8 | 100 | $89.18_{\pm0.2}$ | $74.49_{\pm0.33}$ |
| 10 | 10 | 8 | 500 | $95.57_{\pm0.02}$ | $77.34_{\pm0.19}$ |
| 10 | 10 | 8 | 2000 | $97.87_{\pm0.01}$ | $77.64_{\pm0.19}$ |
| 10 | 50 | 25 | 100 | $90.48_{\pm0.05}$ | $62.52_{\pm0.64}$ |
| 10 | 50 | 25 | 500 | $95.89_{\pm0.02}$ | $64.87_{\pm0.79}$ |
| 10 | 50 | 25 | 2000 | $97.87_{\pm0.0}$ | $76.67_{\pm0.19}$ |
| 10 | 50 | 35 | 100 | $89.69_{\pm0.06}$ | $70.79_{\pm0.23}$ |
| 10 | 50 | 35 | 500 | $95.87_{\pm0.01}$ | $76.09_{\pm0.07}$ |
| 10 | 50 | 35 | 2000 | $97.9_{\pm0.01}$ | $77.17_{\pm0.15}$ |
| 50 | 2 | 1 | 100 | $69.54_{\pm0.52}$ | $55.16_{\pm0.12}$ |
| 50 | 2 | 1 | 500 | $90.94_{\pm0.02}$ | $57.9_{\pm0.15}$ |
| 50 | 2 | 1 | 2000 | $95.64_{\pm0.0}$ | $61.25_{\pm0.15}$ |
| 50 | 3 | 1 | 100 | $66.99_{\pm0.38}$ | $54.6_{\pm0.12}$ |
| 50 | 3 | 1 | 500 | $90.63_{\pm0.02}$ | $59.2_{\pm0.17}$ |
| 50 | 3 | 1 | 2000 | $95.21_{\pm0.0}$ | $61.15_{\pm0.12}$ |
| 50 | 10 | 5 | 100 | $61.92_{\pm0.36}$ | $53.49_{\pm0.07}$ |
| 50 | 10 | 5 | 500 | $90.29_{\pm0.03}$ | $55.28_{\pm0.15}$ |
| 50 | 10 | 5 | 2000 | $95.59_{\pm0.0}$ | $55.06_{\pm0.17}$ |
| 50 | 10 | 8 | 100 | $60.66_{\pm0.35}$ | $56.92_{\pm0.08}$ |
| 50 | 10 | 8 | 500 | $87.01_{\pm0.03}$ | $61.72_{\pm0.09}$ |
| 50 | 10 | 8 | 2000 | $94.34_{\pm0.01}$ | $63.17_{\pm0.09}$ |
| 50 | 50 | 25 | 100 | $59.33_{\pm0.58}$ | $54.31_{\pm0.05}$ |
| 50 | 50 | 25 | 500 | $89.24_{\pm0.03}$ | $54.53_{\pm0.15}$ |
| 50 | 50 | 25 | 2000 | $95.16_{\pm0.01}$ | $55.76_{\pm0.18}$ |
| 50 | 50 | 35 | 100 | $60.68_{\pm0.55}$ | $56.61_{\pm0.15}$ |
| 50 | 50 | 35 | 500 | $88.63_{\pm0.03}$ | $57.73_{\pm0.08}$ |
| 50 | 50 | 35 | 2000 | $94.74_{\pm0.01}$ | $61.02_{\pm0.09}$ |

(a) Centered Gaussian

| $d$ | $q$ | $k$ | $m$ | A2 | R |
|---|---|---|---|---|---|
| 10 | 2 | 1 | 100 | $91.25_{\pm0.19}$ | $74.75_{\pm0.46}$ |
| 10 | 2 | 1 | 500 | $96.26_{\pm0.02}$ | $76.86_{\pm0.23}$ |
| 10 | 2 | 1 | 2000 | $98.18_{\pm0.01}$ | $78.32_{\pm0.23}$ |
| 10 | 3 | 1 | 100 | $90.44_{\pm0.12}$ | $70.64_{\pm1.09}$ |
| 10 | 3 | 1 | 500 | $96.04_{\pm0.04}$ | $73.32_{\pm1.1}$ |
| 10 | 3 | 1 | 2000 | $97.92_{\pm0.01}$ | $75.14_{\pm0.47}$ |
| 10 | 10 | 5 | 100 | $89.36_{\pm0.23}$ | $66.08_{\pm0.98}$ |
| 10 | 10 | 5 | 500 | $95.48_{\pm0.02}$ | $69.46_{\pm0.99}$ |
| 10 | 10 | 5 | 2000 | $97.86_{\pm0.01}$ | $70.41_{\pm0.86}$ |
| 10 | 10 | 8 | 100 | $85.97_{\pm0.3}$ | $63.39_{\pm0.84}$ |
| 10 | 10 | 8 | 500 | $94.02_{\pm0.1}$ | $69.71_{\pm0.58}$ |
| 10 | 10 | 8 | 2000 | $97.4_{\pm0.01}$ | $69.86_{\pm0.8}$ |
| 10 | 50 | 25 | 100 | $87.92_{\pm0.28}$ | $60.65_{\pm0.6}$ |
| 10 | 50 | 25 | 500 | $95.5_{\pm0.03}$ | $61.73_{\pm0.8}$ |
| 10 | 50 | 25 | 2000 | $97.57_{\pm0.02}$ | $70.48_{\pm0.69}$ |
| 10 | 50 | 35 | 100 | $89.12_{\pm0.18}$ | $59.08_{\pm0.52}$ |
| 10 | 50 | 35 | 500 | $95.2_{\pm0.03}$ | $60.67_{\pm0.62}$ |
| 10 | 50 | 35 | 2000 | $97.6_{\pm0.01}$ | $62.86_{\pm0.75}$ |
| 50 | 2 | 1 | 100 | $69.08_{\pm0.46}$ | $55.44_{\pm0.21}$ |
| 50 | 2 | 1 | 500 | $89.46_{\pm0.12}$ | $57.35_{\pm0.32}$ |
| 50 | 2 | 1 | 2000 | $94.99_{\pm0.03}$ | $58.68_{\pm0.15}$ |
| 50 | 3 | 1 | 100 | $66.42_{\pm1.59}$ | $57.24_{\pm0.46}$ |
| 50 | 3 | 1 | 500 | $89.96_{\pm0.23}$ | $57.22_{\pm0.28}$ |
| 50 | 3 | 1 | 2000 | $95.6_{\pm0.05}$ | $59.8_{\pm0.84}$ |
| 50 | 10 | 5 | 100 | $65.76_{\pm1.07}$ | $54.71_{\pm0.13}$ |
| 50 | 10 | 5 | 500 | $90.2_{\pm0.14}$ | $56.57_{\pm0.25}$ |
| 50 | 10 | 5 | 2000 | $95.27_{\pm0.04}$ | $57.43_{\pm0.13}$ |
| 50 | 10 | 8 | 100 | $62.48_{\pm0.7}$ | $62.27_{\pm0.82}$ |
| 50 | 10 | 8 | 500 | $88.04_{\pm0.29}$ | $61.4_{\pm0.49}$ |
| 50 | 10 | 8 | 2000 | $94.44_{\pm0.06}$ | $61.82_{\pm0.54}$ |
| 50 | 50 | 25 | 100 | $65.07_{\pm0.81}$ | $54.78_{\pm0.14}$ |
| 50 | 50 | 25 | 500 | $87.72_{\pm0.28}$ | $55.22_{\pm0.22}$ |
| 50 | 50 | 25 | 2000 | $94.97_{\pm0.02}$ | $53.98_{\pm0.08}$ |
| 50 | 50 | 35 | 100 | $66.12_{\pm0.67}$ | $56.97_{\pm0.28}$ |
| 50 | 50 | 35 | 500 | $87.86_{\pm0.17}$ | $59.57_{\pm0.3}$ |
| 50 | 50 | 35 | 2000 | $94.33_{\pm0.03}$ | $56.97_{\pm0.32}$ |

(b) General Gaussian

Table 8: SDP Algorithm S without offset

| $d$ | $m$ | $q$ | S |
|---|---|---|---|
| 10 | 50 | 2 | $71.82_{\pm8.01}$ |
| 10 | 100 | 2 | $82.53_{\pm5.25}$ |
| 10 | 200 | 2 | $88.39_{\pm3.83}$ |
| 10 | 50 | 3 | $68.94_{\pm5.40}$ |
| 10 | 100 | 3 | $68.28_{\pm4.53}$ |
| 10 | 200 | 3 | $67.31_{\pm6.50}$ |
| 50 | 50 | 2 | $57.32_{\pm2.68}$ |
| 50 | 100 | 2 | $58.72_{\pm2.92}$ |
| 50 | 200 | 2 | $57.83_{\pm2.67}$ |
| 50 | 50 | 3 | $59.18_{\pm3.32}$ |
| 50 | 100 | 3 | $57.86_{\pm2.91}$ |
| 50 | 200 | 3 | $58.69_{\pm3.42}$ |

(a) Centered Gaussian

| $d$ | $m$ | $q$ | S |
|---|---|---|---|
| 10 | 50 | 2 | $77.44_{\pm6.71}$ |
| 10 | 100 | 2 | $84.31_{\pm5.43}$ |
| 10 | 200 | 2 | $90.10_{\pm4.24}$ |
| 10 | 50 | 3 | $72.08_{\pm6.72}$ |
| 10 | 100 | 3 | $71.58_{\pm8.39}$ |
| 10 | 200 | 3 | $70.80_{\pm7.67}$ |
| 50 | 50 | 2 | $59.58_{\pm3.97}$ |
| 50 | 100 | 2 | $59.95_{\pm3.24}$ |
| 50 | 200 | 2 | $61.12_{\pm4.38}$ |
| 50 | 50 | 3 | $63.90_{\pm7.61}$ |
| 50 | 100 | 3 | $62.69_{\pm6.89}$ |
| 50 | 200 | 3 | $62.39_{\pm7.83}$ |

(b) General Gaussian

Table 9: Our algorithms A2 vs. rand. LTF (R) with label flip noise (mentioned in bracket) for Centered Gaussians

| $d$ | $q$ | $k$ | $m$ | A2(0.1) | R(0.1) | A2(0.25) | R(0.25) | A2(0.5) | R(0.5) |
|---|---|---|---|---|---|---|---|---|---|
| 10 | 2 | 1 | 100 | $81.96_{\pm0.4}$ | $72.17_{\pm0.86}$ | $66.9_{\pm0.77}$ | $63.44_{\pm0.85}$ | $57.69_{\pm0.35}$ | $57.4_{\pm0.23}$ |
| 10 | 2 | 1 | 500 | $93.08_{\pm0.03}$ | $77.86_{\pm0.22}$ | $76.36_{\pm1.19}$ | $65.99_{\pm0.53}$ | $57.2_{\pm0.37}$ | $59.24_{\pm0.56}$ |
| 10 | 2 | 1 | 2000 | $97.09_{\pm0.01}$ | $78.21_{\pm0.09}$ | $91.22_{\pm0.09}$ | $77.09_{\pm0.35}$ | $57.64_{\pm0.41}$ | $60.73_{\pm0.47}$ |
| 10 | 3 | 1 | 100 | $79.94_{\pm0.54}$ | $67.69_{\pm0.85}$ | $63.58_{\pm0.88}$ | $62.53_{\pm0.78}$ | $56.24_{\pm0.3}$ | $59.83_{\pm0.36}$ |
| 10 | 3 | 1 | 500 | $92.43_{\pm0.05}$ | $74.82_{\pm0.16}$ | $74.0_{\pm0.82}$ | $67.78_{\pm1.09}$ | $58.37_{\pm0.43}$ | $58.86_{\pm0.21}$ |
| 10 | 3 | 1 | 2000 | $96.53_{\pm0.01}$ | $76.22_{\pm0.16}$ | $88.51_{\pm0.13}$ | $73.74_{\pm0.6}$ | $58.68_{\pm0.39}$ | $59.6_{\pm0.55}$ |
| 10 | 10 | 5 | 100 | $75.87_{\pm0.98}$ | $59.41_{\pm0.57}$ | $62.86_{\pm0.65}$ | $60.47_{\pm0.63}$ | $57.69_{\pm0.38}$ | $56.38_{\pm0.32}$ |
| 10 | 10 | 5 | 500 | $91.97_{\pm0.07}$ | $69.36_{\pm1.07}$ | $74.61_{\pm0.9}$ | $63.14_{\pm0.49}$ | $57.7_{\pm0.28}$ | $61.17_{\pm0.56}$ |
| 10 | 10 | 5 | 2000 | $95.78_{\pm0.01}$ | $74.52_{\pm0.67}$ | $87.93_{\pm0.24}$ | $67.22_{\pm0.75}$ | $58.04_{\pm0.39}$ | $60.28_{\pm0.73}$ |
| 10 | 10 | 8 | 100 | $72.66_{\pm0.66}$ | $71.26_{\pm0.3}$ | $61.11_{\pm0.58}$ | $67.88_{\pm0.71}$ | $56.77_{\pm0.35}$ | $60.96_{\pm0.76}$ |
| 10 | 10 | 8 | 500 | $89.6_{\pm0.1}$ | $75.16_{\pm0.25}$ | $65.65_{\pm0.74}$ | $73.61_{\pm0.45}$ | $57.7_{\pm0.32}$ | $59.26_{\pm0.5}$ |
| 10 | 10 | 8 | 2000 | $94.98_{\pm0.02}$ | $76.48_{\pm0.14}$ | $78.16_{\pm0.78}$ | $76.39_{\pm0.22}$ | $57.78_{\pm0.51}$ | $57.48_{\pm0.3}$ |
| 10 | 50 | 25 | 100 | $76.86_{\pm0.98}$ | $60.43_{\pm0.54}$ | $63.2_{\pm0.92}$ | $61.48_{\pm0.6}$ | $56.06_{\pm0.27}$ | $59.76_{\pm0.42}$ |
| 10 | 50 | 25 | 500 | $91.4_{\pm0.09}$ | $65.14_{\pm0.95}$ | $72.86_{\pm1.11}$ | $61.6_{\pm0.35}$ | $57.98_{\pm0.28}$ | $57.18_{\pm0.28}$ |
| 10 | 50 | 25 | 2000 | $96.36_{\pm0.01}$ | $70.18_{\pm0.75}$ | $87.59_{\pm0.13}$ | $65.52_{\pm0.83}$ | $57.05_{\pm0.16}$ | $58.78_{\pm0.36}$ |
| 10 | 50 | 35 | 100 | $74.32_{\pm0.7}$ | $69.02_{\pm0.44}$ | $60.0_{\pm0.6}$ | $65.95_{\pm0.7}$ | $58.81_{\pm0.27}$ | $58.44_{\pm0.4}$ |
| 10 | 50 | 35 | 500 | $91.23_{\pm0.05}$ | $75.32_{\pm0.25}$ | $69.86_{\pm0.96}$ | $69.33_{\pm0.37}$ | $58.76_{\pm0.54}$ | $57.96_{\pm0.22}$ |
| 10 | 50 | 35 | 2000 | $95.39_{\pm0.01}$ | $75.46_{\pm0.13}$ | $84.6_{\pm0.23}$ | $75.25_{\pm0.26}$ | $56.95_{\pm0.28}$ | $59.66_{\pm0.53}$ |
| 50 | 2 | 1 | 100 | $58.14_{\pm0.36}$ | $55.05_{\pm0.15}$ | $54.43_{\pm0.12}$ | $54.15_{\pm0.11}$ | $54.57_{\pm0.15}$ | $54.78_{\pm0.09}$ |
| 50 | 2 | 1 | 500 | $82.97_{\pm0.07}$ | $57.46_{\pm0.27}$ | $59.14_{\pm0.36}$ | $54.14_{\pm0.06}$ | $53.73_{\pm0.08}$ | $53.72_{\pm0.09}$ |
| 50 | 2 | 1 | 2000 | $91.99_{\pm0.02}$ | $59.86_{\pm0.26}$ | $73.26_{\pm0.41}$ | $56.35_{\pm0.17}$ | $54.2_{\pm0.08}$ | $54.7_{\pm0.13}$ |
| 50 | 3 | 1 | 100 | $59.07_{\pm0.32}$ | $54.72_{\pm0.13}$ | $54.66_{\pm0.12}$ | $53.8_{\pm0.08}$ | $53.45_{\pm0.08}$ | $55.07_{\pm0.1}$ |
| 50 | 3 | 1 | 500 | $79.51_{\pm0.22}$ | $56.96_{\pm0.17}$ | $57.44_{\pm0.15}$ | $54.99_{\pm0.1}$ | $53.55_{\pm0.08}$ | $53.41_{\pm0.06}$ |
| 50 | 3 | 1 | 2000 | $90.66_{\pm0.01}$ | $60.26_{\pm0.17}$ | $66.04_{\pm0.58}$ | $56.88_{\pm0.13}$ | $54.28_{\pm0.1}$ | $54.69_{\pm0.14}$ |
| 50 | 10 | 5 | 100 | $56.3_{\pm0.29}$ | $54.32_{\pm0.07}$ | $54.77_{\pm0.12}$ | $53.9_{\pm0.12}$ | $53.92_{\pm0.11}$ | $54.68_{\pm0.13}$ |
| 50 | 10 | 5 | 500 | $75.21_{\pm0.62}$ | $54.48_{\pm0.06}$ | $57.21_{\pm0.26}$ | $54.14_{\pm0.12}$ | $53.68_{\pm0.08}$ | $52.5_{\pm0.04}$ |
| 50 | 10 | 5 | 2000 | $90.68_{\pm0.02}$ | $55.72_{\pm0.24}$ | $65.65_{\pm0.56}$ | $56.06_{\pm0.13}$ | $53.21_{\pm0.05}$ | $55.06_{\pm0.11}$ |
| 50 | 10 | 8 | 100 | $55.76_{\pm0.18}$ | $57.33_{\pm0.15}$ | $54.34_{\pm0.08}$ | $56.62_{\pm0.17}$ | $53.72_{\pm0.05}$ | $54.93_{\pm0.1}$ |
| 50 | 10 | 8 | 500 | $65.55_{\pm0.57}$ | $60.97_{\pm0.14}$ | $55.0_{\pm0.21}$ | $58.98_{\pm0.15}$ | $54.7_{\pm0.11}$ | $53.82_{\pm0.06}$ |
| 50 | 10 | 8 | 2000 | $87.17_{\pm0.04}$ | $62.39_{\pm0.11}$ | $60.28_{\pm0.26}$ | $61.82_{\pm0.14}$ | $54.76_{\pm0.17}$ | $53.41_{\pm0.05}$ |
| 50 | 50 | 25 | 100 | $54.6_{\pm0.06}$ | $54.16_{\pm0.11}$ | $55.45_{\pm0.17}$ | $55.12_{\pm0.12}$ | $53.1_{\pm0.04}$ | $53.91_{\pm0.07}$ |
| 50 | 50 | 25 | 500 | $77.1_{\pm0.22}$ | $53.81_{\pm0.07}$ | $56.34_{\pm0.26}$ | $54.44_{\pm0.09}$ | $53.4_{\pm0.05}$ | $53.8_{\pm0.09}$ |
| 50 | 50 | 25 | 2000 | $90.28_{\pm0.03}$ | $55.11_{\pm0.1}$ | $65.29_{\pm0.42}$ | $55.26_{\pm0.08}$ | $53.43_{\pm0.08}$ | $54.43_{\pm0.16}$ |
| 50 | 50 | 35 | 100 | $55.33_{\pm0.1}$ | $55.45_{\pm0.13}$ | $54.51_{\pm0.14}$ | $54.08_{\pm0.07}$ | $52.94_{\pm0.04}$ | $54.24_{\pm0.05}$ |
| 50 | 50 | 35 | 500 | $72.95_{\pm0.43}$ | $59.5_{\pm0.16}$ | $56.18_{\pm0.16}$ | $57.85_{\pm0.1}$ | $53.71_{\pm0.11}$ | $54.28_{\pm0.12}$ |
| 50 | 50 | 35 | 2000 | $89.07_{\pm0.02}$ | $60.11_{\pm0.16}$ | $62.83_{\pm0.59}$ | $57.27_{\pm0.11}$ | $54.51_{\pm0.13}$ | $53.99_{\pm0.1}$ |

Table 10: Our algorithms A2 vs. rand. LTF (R) with label flip noise (mentioned in bracket) for General Gaussians

| $d$ | $q$ | $k$ | $m$ | A2(0.1) | R(0.1) | A2(0.25) | R(0.25) | A2(0.5) | R(0.5) |
|---|---|---|---|---|---|---|---|---|---|
| 10 | 2 | 1 | 100 | $86.68_{\pm0.46}$ | $68.35_{\pm0.96}$ | $65.88_{\pm0.83}$ | $60.62_{\pm0.66}$ | $59.48_{\pm0.48}$ | $58.82_{\pm0.37}$ |
| 10 | 2 | 1 | 500 | $93.34_{\pm0.09}$ | $77.69_{\pm0.48}$ | $82.68_{\pm0.79}$ | $67.5_{\pm0.93}$ | $60.55_{\pm0.57}$ | $61.5_{\pm0.71}$ |
| 10 | 2 | 1 | 2000 | $96.94_{\pm0.02}$ | $75.5_{\pm0.16}$ | $90.02_{\pm0.24}$ | $76.6_{\pm0.24}$ | $60.26_{\pm0.53}$ | $60.05_{\pm0.61}$ |
| 10 | 3 | 1 | 100 | $80.05_{\pm0.93}$ | $64.76_{\pm1.07}$ | $64.66_{\pm0.8}$ | $62.96_{\pm0.98}$ | $62.46_{\pm0.6}$ | $60.63_{\pm0.43}$ |
| 10 | 3 | 1 | 500 | $91.83_{\pm0.12}$ | $70.26_{\pm1.42}$ | $74.9_{\pm1.18}$ | $65.09_{\pm0.96}$ | $59.86_{\pm0.49}$ | $58.69_{\pm0.38}$ |
| 10 | 3 | 1 | 2000 | $96.12_{\pm0.03}$ | $73.05_{\pm0.97}$ | $88.23_{\pm0.11}$ | $63.38_{\pm0.71}$ | $60.22_{\pm0.51}$ | $60.06_{\pm0.39}$ |
| 10 | 10 | 5 | 100 | $79.69_{\pm1.2}$ | $65.48_{\pm0.77}$ | $63.53_{\pm0.78}$ | $63.28_{\pm0.85}$ | $64.81_{\pm0.75}$ | $59.51_{\pm0.64}$ |
| 10 | 10 | 5 | 500 | $91.9_{\pm0.11}$ | $71.2_{\pm1.0}$ | $71.65_{\pm1.33}$ | $61.41_{\pm0.62}$ | $62.1_{\pm0.85}$ | $61.56_{\pm0.45}$ |
| 10 | 10 | 5 | 2000 | $96.21_{\pm0.03}$ | $71.95_{\pm0.71}$ | $86.2_{\pm0.28}$ | $63.04_{\pm0.93}$ | $60.43_{\pm0.71}$ | $59.52_{\pm0.4}$ |
| 10 | 10 | 8 | 100 | $72.46_{\pm1.1}$ | $59.4_{\pm0.47}$ | $64.74_{\pm0.92}$ | $61.22_{\pm0.57}$ | $59.93_{\pm0.75}$ | $63.45_{\pm0.87}$ |
| 10 | 10 | 8 | 500 | $87.61_{\pm0.47}$ | $61.66_{\pm0.79}$ | $64.32_{\pm0.68}$ | $62.22_{\pm0.74}$ | $65.04_{\pm0.65}$ | $64.1_{\pm0.61}$ |
| 10 | 10 | 8 | 2000 | $94.53_{\pm0.07}$ | $63.79_{\pm0.83}$ | $77.46_{\pm1.37}$ | $61.28_{\pm0.58}$ | $61.05_{\pm0.75}$ | $64.82_{\pm0.87}$ |
| 10 | 50 | 25 | 100 | $80.2_{\pm0.85}$ | $60.49_{\pm0.5}$ | $62.72_{\pm0.93}$ | $59.3_{\pm0.62}$ | $58.45_{\pm0.49}$ | $59.32_{\pm0.48}$ |
| 10 | 50 | 25 | 500 | $91.96_{\pm0.22}$ | $61.28_{\pm0.57}$ | $72.0_{\pm1.05}$ | $58.47_{\pm0.64}$ | $59.89_{\pm0.9}$ | $58.85_{\pm0.31}$ |
| 10 | 50 | 25 | 2000 | $96.38_{\pm0.04}$ | $63.74_{\pm0.67}$ | $84.59_{\pm0.78}$ | $62.78_{\pm0.72}$ | $59.84_{\pm0.48}$ | $60.17_{\pm0.57}$ |
| 10 | 50 | 35 | 100 | $69.83_{\pm1.44}$ | $60.74_{\pm0.45}$ | $62.45_{\pm0.71}$ | $59.02_{\pm0.24}$ | $62.27_{\pm0.65}$ | $60.51_{\pm0.72}$ |
| 10 | 50 | 35 | 500 | $90.07_{\pm0.14}$ | $62.02_{\pm0.96}$ | $66.85_{\pm1.68}$ | $61.81_{\pm1.09}$ | $61.67_{\pm0.93}$ | $62.31_{\pm0.6}$ |
| 10 | 50 | 35 | 2000 | $95.24_{\pm0.03}$ | $62.2_{\pm1.02}$ | $81.07_{\pm0.65}$ | $61.24_{\pm0.5}$ | $61.13_{\pm0.7}$ | $60.29_{\pm0.49}$ |
| 50 | 2 | 1 | 100 | $61.4_{\pm0.99}$ | $57.38_{\pm0.25}$ | $57.99_{\pm0.35}$ | $55.72_{\pm0.12}$ | $56.0_{\pm0.22}$ | $54.19_{\pm0.09}$ |
| 50 | 2 | 1 | 500 | $82.7_{\pm0.37}$ | $56.61_{\pm0.15}$ | $60.44_{\pm0.73}$ | $56.8_{\pm0.14}$ | $56.3_{\pm0.32}$ | $54.22_{\pm0.08}$ |
| 50 | 2 | 1 | 2000 | $91.5_{\pm0.07}$ | $57.45_{\pm0.22}$ | $74.38_{\pm0.66}$ | $56.52_{\pm0.16}$ | $58.35_{\pm0.17}$ | $54.0_{\pm0.13}$ |
| 50 | 3 | 1 | 100 | $58.07_{\pm0.42}$ | $58.16_{\pm0.26}$ | $59.48_{\pm0.35}$ | $59.29_{\pm0.35}$ | $57.95_{\pm0.72}$ | $56.86_{\pm0.23}$ |
| 50 | 3 | 1 | 500 | $77.58_{\pm0.45}$ | $57.2_{\pm0.24}$ | $61.05_{\pm0.55}$ | $56.24_{\pm0.35}$ | $56.85_{\pm0.44}$ | $56.77_{\pm0.23}$ |
| 50 | 3 | 1 | 2000 | $89.25_{\pm0.14}$ | $55.3_{\pm0.18}$ | $71.13_{\pm0.66}$ | $56.78_{\pm0.3}$ | $56.98_{\pm0.3}$ | $55.62_{\pm0.22}$ |
| 50 | 10 | 5 | 100 | $59.26_{\pm0.5}$ | $54.08_{\pm0.1}$ | $57.16_{\pm0.31}$ | $54.97_{\pm0.08}$ | $56.82_{\pm0.36}$ | $55.3_{\pm0.16}$ |
| 50 | 10 | 5 | 500 | $75.83_{\pm0.65}$ | $55.05_{\pm0.13}$ | $59.42_{\pm0.5}$ | $54.61_{\pm0.09}$ | $59.04_{\pm0.72}$ | $54.56_{\pm0.07}$ |
| 50 | 10 | 5 | 2000 | $88.63_{\pm0.13}$ | $54.74_{\pm0.18}$ | $68.27_{\pm0.56}$ | $55.38_{\pm0.13}$ | $60.22_{\pm0.6}$ | $54.48_{\pm0.1}$ |
| 50 | 10 | 8 | 100 | $58.23_{\pm0.7}$ | $60.75_{\pm0.48}$ | $57.71_{\pm0.28}$ | $60.52_{\pm0.53}$ | $55.46_{\pm0.2}$ | $59.23_{\pm0.53}$ |
| 50 | 10 | 8 | 500 | $68.01_{\pm1.04}$ | $60.06_{\pm0.32}$ | $58.77_{\pm0.32}$ | $61.39_{\pm0.54}$ | $58.76_{\pm0.88}$ | $60.28_{\pm0.85}$ |
| 50 | 10 | 8 | 2000 | $85.53_{\pm0.17}$ | $61.66_{\pm0.42}$ | $62.99_{\pm1.22}$ | $61.56_{\pm0.67}$ | $57.54_{\pm0.44}$ | $59.17_{\pm0.54}$ |
| 50 | 50 | 25 | 100 | $59.24_{\pm0.5}$ | $54.38_{\pm0.08}$ | $59.94_{\pm0.56}$ | $53.92_{\pm0.08}$ | $57.92_{\pm0.48}$ | $54.84_{\pm0.14}$ |
| 50 | 50 | 25 | 500 | $80.22_{\pm0.81}$ | $54.26_{\pm0.1}$ | $57.32_{\pm0.4}$ | $53.97_{\pm0.12}$ | $55.14_{\pm0.22}$ | $55.18_{\pm0.2}$ |
| 50 | 50 | 25 | 2000 | $90.32_{\pm0.12}$ | $54.03_{\pm0.1}$ | $64.51_{\pm0.91}$ | $54.95_{\pm0.15}$ | $57.8_{\pm0.33}$ | $55.2_{\pm0.13}$ |
| 50 | 50 | 35 | 100 | $57.1_{\pm0.34}$ | $56.03_{\pm0.31}$ | $58.14_{\pm0.6}$ | $57.16_{\pm0.36}$ | $55.18_{\pm0.24}$ | $56.69_{\pm0.27}$ |
| 50 | 50 | 35 | 500 | $72.81_{\pm0.83}$ | $58.09_{\pm0.44}$ | $59.59_{\pm0.43}$ | $56.23_{\pm0.21}$ | $57.36_{\pm0.62}$ | $58.39_{\pm0.32}$ |
| 50 | 50 | 35 | 2000 | $88.48_{\pm0.18}$ | $55.86_{\pm0.39}$ | $61.99_{\pm0.6}$ | $57.32_{\pm0.22}$ | $57.42_{\pm0.4}$ | $58.0_{\pm0.31}$ |

Table 11: Our algorithms A4 vs. rand. LTF (R) with offset

| $d$ | $q$ | $k$ | $m$ | A4 | R | $d$ | $q$ | $k$ | $m$ | A4 | R |
|---|---|---|---|---|---|---|---|---|---|---|---|
| 10 | 2 | 1 | 100 | $89.41_{\pm 0.09}$ | $73.83_{\pm 0.52}$ | 10 | 2 | 1 | 100 | $88.36_{\pm 0.24}$ | $74.23_{\pm 0.39}$ |
| 10 | 2 | 1 | 500 | $92.2_{\pm 0.18}$ | $76.7_{\pm 0.21}$ | 10 | 2 | 1 | 500 | $92.22_{\pm 0.38}$ | $77.04_{\pm 0.3}$ |
| 10 | 2 | 1 | 2000 | $92.49_{\pm 0.26}$ | $77.23_{\pm 0.17}$ | 10 | 2 | 1 | 2000 | $93.08_{\pm 0.32}$ | $78.24_{\pm 0.24}$ |
| 10 | 3 | 1 | 100 | $89.15_{\pm 0.09}$ | $71.78_{\pm 0.77}$ | 10 | 3 | 1 | 100 | $88.47_{\pm 0.39}$ | $68.16_{\pm 0.96}$ |
| 10 | 3 | 1 | 500 | $92.5_{\pm 0.09}$ | $72.66_{\pm 0.42}$ | 10 | 3 | 1 | 500 | $93.09_{\pm 0.21}$ | $72.18_{\pm 0.4}$ |
| 10 | 3 | 1 | 2000 | $93.67_{\pm 0.13}$ | $74.96_{\pm 0.1}$ | 10 | 3 | 1 | 2000 | $94.62_{\pm 0.16}$ | $71.49_{\pm 0.3}$ |
| 10 | 10 | 5 | 100 | $89.43_{\pm 0.08}$ | $64.12_{\pm 0.75}$ | 10 | 10 | 5 | 100 | $82.44_{\pm 1.08}$ | $64.65_{\pm 0.86}$ |
| 10 | 10 | 5 | 500 | $93.31_{\pm 0.1}$ | $73.14_{\pm 0.83}$ | 10 | 10 | 5 | 500 | $90.75_{\pm 0.41}$ | $68.96_{\pm 0.72}$ |
| 10 | 10 | 5 | 2000 | $94.43_{\pm 0.14}$ | $76.42_{\pm 0.29}$ | 10 | 10 | 5 | 2000 | $93.1_{\pm 0.38}$ | $70.59_{\pm 0.78}$ |
| 10 | 10 | 8 | 100 | $85.16_{\pm 0.31}$ | $64.0_{\pm 0.88}$ | 10 | 10 | 8 | 100 | $84.11_{\pm 0.97}$ | $64.71_{\pm 0.96}$ |
| 10 | 10 | 8 | 500 | $90.73_{\pm 0.11}$ | $68.29_{\pm 0.58}$ | 10 | 10 | 8 | 500 | $92.71_{\pm 0.17}$ | $65.91_{\pm 1.16}$ |
| 10 | 10 | 8 | 2000 | $92.5_{\pm 0.17}$ | $65.87_{\pm 0.77}$ | 10 | 10 | 8 | 2000 | $94.47_{\pm 0.26}$ | $67.85_{\pm 1.05}$ |
| 10 | 50 | 25 | 100 | $88.68_{\pm 0.08}$ | $62.41_{\pm 0.47}$ | 10 | 50 | 25 | 100 | $86.78_{\pm 0.72}$ | $60.89_{\pm 0.57}$ |
| 10 | 50 | 25 | 500 | $92.41_{\pm 0.11}$ | $62.66_{\pm 0.65}$ | 10 | 50 | 25 | 500 | $91.64_{\pm 0.25}$ | $63.56_{\pm 0.75}$ |
| 10 | 50 | 25 | 2000 | $92.83_{\pm 0.15}$ | $70.34_{\pm 0.75}$ | 10 | 50 | 25 | 2000 | $93.93_{\pm 0.13}$ | $65.03_{\pm 1.14}$ |
| 10 | 50 | 35 | 100 | $87.0_{\pm 0.23}$ | $62.8_{\pm 0.64}$ | 10 | 50 | 35 | 100 | $85.69_{\pm 0.87}$ | $59.46_{\pm 0.66}$ |
| 10 | 50 | 35 | 500 | $92.26_{\pm 0.14}$ | $65.4_{\pm 0.75}$ | 10 | 50 | 35 | 500 | $91.96_{\pm 0.3}$ | $66.64_{\pm 1.08}$ |
| 10 | 50 | 35 | 2000 | $92.92_{\pm 0.19}$ | $65.75_{\pm 0.78}$ | 10 | 50 | 35 | 2000 | $93.31_{\pm 0.33}$ | $66.04_{\pm 0.8}$ |
| 50 | 2 | 1 | 100 | $67.1_{\pm 0.9}$ | $54.03_{\pm 0.12}$ | 50 | 2 | 1 | 100 | $70.05_{\pm 0.96}$ | $55.58_{\pm 0.23}$ |
| 50 | 2 | 1 | 500 | $90.51_{\pm 0.01}$ | $57.79_{\pm 0.17}$ | 50 | 2 | 1 | 500 | $90.16_{\pm 0.04}$ | $58.88_{\pm 0.31}$ |
| 50 | 2 | 1 | 2000 | $94.84_{\pm 0.02}$ | $59.34_{\pm 0.19}$ | 50 | 2 | 1 | 2000 | $94.17_{\pm 0.08}$ | $59.66_{\pm 0.24}$ |
| 50 | 3 | 1 | 100 | $59.98_{\pm 0.46}$ | $55.47_{\pm 0.07}$ | 50 | 3 | 1 | 100 | $63.86_{\pm 1.03}$ | $58.19_{\pm 0.37}$ |
| 50 | 3 | 1 | 500 | $89.64_{\pm 0.04}$ | $55.86_{\pm 0.18}$ | 50 | 3 | 1 | 500 | $86.73_{\pm 0.56}$ | $58.3_{\pm 0.36}$ |
| 50 | 3 | 1 | 2000 | $94.7_{\pm 0.01}$ | $58.36_{\pm 0.26}$ | 50 | 3 | 1 | 2000 | $94.41_{\pm 0.1}$ | $59.92_{\pm 0.42}$ |
| 50 | 10 | 5 | 100 | $63.14_{\pm 0.62}$ | $53.58_{\pm 0.08}$ | 50 | 10 | 5 | 100 | $64.76_{\pm 1.18}$ | $54.76_{\pm 0.13}$ |
| 50 | 10 | 5 | 500 | $89.88_{\pm 0.02}$ | $54.24_{\pm 0.1}$ | 50 | 10 | 5 | 500 | $86.8_{\pm 0.51}$ | $55.0_{\pm 0.14}$ |
| 50 | 10 | 5 | 2000 | $95.15_{\pm 0.01}$ | $57.67_{\pm 0.22}$ | 50 | 10 | 5 | 2000 | $94.06_{\pm 0.04}$ | $56.12_{\pm 0.23}$ |
| 50 | 10 | 8 | 100 | $55.52_{\pm 0.19}$ | $58.26_{\pm 0.2}$ | 50 | 10 | 8 | 100 | $62.47_{\pm 0.99}$ | $60.43_{\pm 0.48}$ |
| 50 | 10 | 8 | 500 | $84.76_{\pm 0.06}$ | $57.08_{\pm 0.16}$ | 50 | 10 | 8 | 500 | $82.48_{\pm 0.7}$ | $60.34_{\pm 0.8}$ |
| 50 | 10 | 8 | 2000 | $92.49_{\pm 0.03}$ | $56.46_{\pm 0.23}$ | 50 | 10 | 8 | 2000 | $92.3_{\pm 0.09}$ | $59.67_{\pm 0.57}$ |
| 50 | 50 | 25 | 100 | $61.96_{\pm 0.36}$ | $55.44_{\pm 0.21}$ | 50 | 50 | 25 | 100 | $59.94_{\pm 1.25}$ | $54.34_{\pm 0.17}$ |
| 50 | 50 | 25 | 500 | $88.96_{\pm 0.02}$ | $55.74_{\pm 0.16}$ | 50 | 50 | 25 | 500 | $88.3_{\pm 0.23}$ | $54.11_{\pm 0.14}$ |
| 50 | 50 | 25 | 2000 | $94.51_{\pm 0.01}$ | $54.53_{\pm 0.1}$ | 50 | 50 | 25 | 2000 | $93.41_{\pm 0.13}$ | $55.72_{\pm 0.23}$ |
| 50 | 50 | 35 | 100 | $61.06_{\pm 0.28}$ | $56.4_{\pm 0.19}$ | 50 | 50 | 35 | 100 | $65.29_{\pm 1.37}$ | $58.16_{\pm 0.63}$ |
| 50 | 50 | 35 | 500 | $88.32_{\pm 0.04}$ | $56.7_{\pm 0.14}$ | 50 | 50 | 35 | 500 | $84.11_{\pm 1.16}$ | $56.98_{\pm 0.34}$ |
| 50 | 50 | 35 | 2000 | $94.45_{\pm 0.01}$ | $57.76_{\pm 0.2}$ | 50 | 50 | 35 | 2000 | $94.08_{\pm 0.07}$ | $56.97_{\pm 0.4}$ |

(a) Centered Gaussian          (b) General Gaussian

Table 12: SDP Algorithm S with offset

| $d$ | $m$ | $q$ | S | $d$ | $m$ | $q$ | S |
|---|---|---|---|---|---|---|---|
| 10 | 50 | 2 | $73.16_{\pm 7.26}$ | 10 | 50 | 2 | $75.96_{\pm 5.05}$ |
| 10 | 100 | 2 | $79.09_{\pm 9.21}$ | 10 | 100 | 2 | $80.76_{\pm 6.47}$ |
| 10 | 200 | 2 | $82.76_{\pm 7.02}$ | 10 | 200 | 2 | $88.01_{\pm 5.08}$ |
| 10 | 50 | 3 | $67.80_{\pm 5.78}$ | 10 | 50 | 3 | $69.84_{\pm 6.78}$ |
| 10 | 100 | 3 | $68.52_{\pm 6.28}$ | 10 | 100 | 3 | $66.95_{\pm 5.62}$ |
| 10 | 200 | 3 | $69.62_{\pm 6.90}$ | 10 | 200 | 3 | $67.57_{\pm 6.12}$ |
| 50 | 50 | 2 | $59.30_{\pm 3.17}$ | 50 | 50 | 2 | $61.41_{\pm 3.51}$ |
| 50 | 100 | 2 | $59.04_{\pm 3.27}$ | 50 | 100 | 2 | $59.87_{\pm 4.60}$ |
| 50 | 200 | 2 | $58.25_{\pm 2.68}$ | 50 | 200 | 2 | $61.01_{\pm 3.17}$ |
| 50 | 50 | 3 | $58.17_{\pm 3.32}$ | 50 | 50 | 3 | $63.88_{\pm 6.37}$ |
| 50 | 100 | 3 | $57.94_{\pm 2.72}$ | 50 | 100 | 3 | $62.62_{\pm 5.61}$ |
| 50 | 200 | 3 | $59.37_{\pm 3.14}$ | 50 | 200 | 3 | $64.24_{\pm 6.64}$ |

(a) Centered Gaussian          (b) General Gaussian

# G   Class ratio estimation for LTFs

The work of [13] studies the problem of matching the classifier label proportion using a single sampled bag which they call *class-ratio* (CR) learning as distinct from LLP. Indeed, in LLP the goal is to learn an accurate instance-level classifier from multiple sampled bags, whereas CR-learning does not guarantee instance-level performance. Further, similar to Prop. 18 of [13], CR learning LTFs over Gaussians is easy: for a bag $B = \{\mathbf{x}^{(i)}\}_{i=1}^n$ of iid Gaussian points, a random unit vector $\mathbf{r}$ has distinct inner products $\{s_i := \mathbf{r}^\mathsf{T}\mathbf{x}^{(i)}\}_{i=1}^n$ with probability 1. The LTFs $\{\text{pos}\left(\mathbf{r}^\mathsf{T}\mathbf{x} - s\right) \mid s \in \{-\infty, s_1, \ldots, s_n\}\}$ achieve all possible target label proportions $\{j/n\}_{j=0}^n$, and one can then apply the generalization error bound in Thm. 4 of [13].

# H   Analysis of a Mixture of Label Sums

**Definition H.1** (Mixed Bag Oracle). *Given a set of bag oracles* $\text{Ex}(f, \mathcal{D}, q, k)$ *for* $k \in \{0, \ldots, q\}$ *and* $\mathbf{p} = (p_0, \ldots, p_q) \in \Delta^q$ *where* $\Delta^q$ *is a q-simplex, a mixed bag oracle* $\text{Ex}(f, \mathcal{D}, q, \mathbf{p})$ *samples a bag size k from* $\text{Multinoulli}(\mathbf{p})$ *distribution[2] and then samples a bag from* $\text{Ex}(f, \mathcal{D}, q, k)$.

Let $\mathbf{\Sigma}_D$ be the covariance matrix of difference of a pair of vectors sampled u.a.r without replacement from $\text{Ex}(f, \mathcal{D}, q, \mathbf{p})$ and $\mathbf{\Sigma}_B$ be the covariance matrix of vectors sampled u.a.r from $\text{Ex}(f, \mathcal{D}, q, \mathbf{p})$. If $\mathbf{\Sigma}_{Dk}$ is the covariance matrix of difference of a pair of vectors sampled u.a.r without replacement from $\text{Ex}(f, \mathcal{D}, q, k)$ and $\mathbf{\Sigma}_{Bk}$ be the covariance matrix of vectors sampled u.a.r from $\text{Ex}(f, \mathcal{D}, q, k)$ then we have the following

$$\mathbf{\Sigma}_B = \sum_{k=0}^{q} p_k^2 \mathbf{\Sigma}_{Bk} \qquad\qquad \mathbf{\Sigma}_D = \sum_{k=0}^{q} p_k^2 \mathbf{\Sigma}_{Dk} \qquad\qquad (54)$$

Using the above, we prove the following geometric error bound which is analogous to Lemma 4.1.

**Lemma H.2.** *For any* $\varepsilon, \delta \in (0, 1)$, *if* $m \geq O\left((d/\varepsilon^4)\log(d/\delta)(\lambda_{\max}/\lambda_{\min})^4 q^4 \left(1/\sum_{k=1}^{q-1} p_k^2\right)^2\right)$, *then* $\hat{\mathbf{r}}$ *computed in Step 3 of Alg. 2 satisfies* $\min\{\|\hat{\mathbf{r}} - \mathbf{r}_*\|_2, \|\hat{\mathbf{r}} + \mathbf{r}_*\|_2\} \leq \varepsilon$, *w.p.* $1 - \delta/2$.

## H.1   Proof of Lemma H.2

We define and bound the following useful quantities based on $q$, $\lambda_{\max}$, $\lambda_{\min}$ and $\mathbf{p}$.

**Definition H.3.** *Define, (i)* $\kappa_1(k) := \left(\frac{2k}{q} - 1\right)^2 \frac{2}{\pi}$ *so that* $0 \leq \kappa_1(k) \leq 2/\pi$, *(ii)* $\kappa_2(k) := \frac{1}{q-1}\frac{k}{q}\left(1 - \frac{k}{q}\right)\frac{16}{\pi}$ *so that* $\frac{16}{\pi q^2} \leq \kappa_2(k) \leq \frac{4}{\pi(q-1)}$ *whenever* $1 \leq k \leq q - 1$, *(iii)* $\kappa_3(\mathbf{p}) := \frac{\sum_{k=0}^{q} p_k^2 \kappa_2(k)}{\sum_{k=0}^{q} p_k^2 - \sum_{k=0}^{q} p_k^2 \kappa_1(k)}$ *so that* $\frac{16 \sum_{k=1}^{q-1} p_k^2}{\pi q^2 \sum_{k=0}^{q} p_k^2} \leq \kappa_3(\mathbf{p})$, *and (iv)* $\theta(\mathbf{p}) := \frac{2\lambda_{\max}}{\lambda_{\min}}\left(\frac{1}{2\sum_{k=0}^{q} p_k^2 - \max(0, 2\sum_{k=0}^{q}\kappa_1(k) - \sum_{k=0}^{q} p_k^2 \kappa_2(k))} + \frac{1}{\sum_{k=0}^{q} p_k^2 - \sum_{k=0}^{q} p_k^2 \kappa_1(k)}\right)$ *so that* $\theta(\mathbf{p}) \leq \frac{3\lambda_{\max}}{(1-2/\pi)\lambda_{\min}(\sum_{k=0}^{q} p_k^2)}$.

**Lemma H.4.** *The ratio* $\rho(\mathbf{r}) := \mathbf{r}^\mathsf{T}\mathbf{\Sigma}_D\mathbf{r}/\mathbf{r}^\mathsf{T}\mathbf{\Sigma}_B\mathbf{r}$ *is maximized when* $\mathbf{r} = \pm\mathbf{r}_*$. *Moreover,*

$$\rho(\mathbf{r}) = 2 + \frac{\gamma(\mathbf{r})^2 \sum_{k=0}^{q} p_k^2 \kappa_2(k)}{\sum_{k=0}^{q} p_k^2 - \gamma(\mathbf{r})^2 \sum_{k=0}^{q} p_k^2 \kappa_1(k)} \qquad where \qquad \gamma(\mathbf{r}) := \frac{\mathbf{r}^\mathsf{T}\mathbf{\Sigma}\mathbf{r}_*}{\sqrt{\mathbf{r}^\mathsf{T}\mathbf{\Sigma}\mathbf{r}}\sqrt{\mathbf{r}_*^\mathsf{T}\mathbf{\Sigma}\mathbf{r}_*}} \quad and$$

$$\mathbf{r}^\mathsf{T}\mathbf{\Sigma}_B\mathbf{r} = \mathbf{r}^\mathsf{T}\mathbf{\Sigma}\mathbf{r}\left(\sum_{k=0}^{q} p_k^2 - \gamma(\mathbf{r})^2 \sum_{k=0}^{q} p_k^2 \kappa_1(k)\right),$$

$$\mathbf{r}^\mathsf{T}\mathbf{\Sigma}_D\mathbf{r} = \mathbf{r}^\mathsf{T}\mathbf{\Sigma}\mathbf{r}\left(2\sum_{k=0}^{q} p_k^2 - 2\gamma(\mathbf{r})^2 \sum_{k=0}^{q} p_k^2 \kappa_1(k) + \gamma(\mathbf{r})^2 \sum_{k=0}^{q} p_k^2 \kappa_2(k)\right)$$

---

[2] Section 2.3.2 (p. 35) of 'Machine Learning: A Probabilistic Perspective' (by K. Murphy)

*Proof.* The proof follows directly from Lemma 4.4 which gives us the expression for $\boldsymbol{\Sigma}_{Bk}$ and $\boldsymbol{\Sigma}_{Dk}$ and (54). Once the expression for $\rho(\mathbf{r})$ is obtained, it is easy to see that since $|\gamma(\mathbf{r})| \leq 1$, $\rho(\mathbf{r})$ maximizes when $\gamma(\mathbf{r}) = \pm 1$ and thus when $\mathbf{r} = \pm \mathbf{r}_*$. $\qquad\square$

*Proof.* (of Lemma H.2) By Lemma 3.1, taking $m \geq O\left((d/\varepsilon_1^2)\log(d/\delta)\right)$ ensures that $\|\mathbf{E}_B\|_2 \leq \varepsilon_1 \lambda_{\max}$ and $\|\mathbf{E}_D\|_2 \leq \varepsilon_1 \lambda_{\max}$ w.p. at least $1 - \delta$ where $\mathbf{E}_B = \hat{\boldsymbol{\Sigma}}_B - \boldsymbol{\Sigma}_B$ and $\mathbf{E}_D = \hat{\boldsymbol{\Sigma}}_D - \boldsymbol{\Sigma}_D$. We start by defining $\hat{\rho}(\mathbf{r}) := \frac{\mathbf{r}^\top \hat{\boldsymbol{\Sigma}}_D \mathbf{r}}{\mathbf{r}^\top \hat{\boldsymbol{\Sigma}}_B \mathbf{r}}$ which is the equivalent of $\rho$ using the estimated matrices. Observe that it can be written as $\hat{\rho}(\mathbf{r}) = \frac{\mathbf{r}^\top \boldsymbol{\Sigma}_B \mathbf{r} + \mathbf{r}^\top \mathbf{E}_B \mathbf{r}}{\mathbf{r}^\top \boldsymbol{\Sigma}_D \mathbf{r} + \mathbf{r}^\top \mathbf{E}_D \mathbf{r}}$. Using these we can obtain the following bound on $\hat{\rho}$: for any $\mathbf{r} \in \mathbb{R}^d$, $|\hat{\rho}(\mathbf{r}) - \rho(\mathbf{r})| \leq \theta(\mathbf{p})\varepsilon_1 |\rho(\mathbf{r})|$ w.p. at least $1 - \delta$ (*) as long as $\varepsilon_1 \leq \frac{(\sum_{k=0}^q p_k^2 - \sum_{k=0}^q p_k^2 \kappa_1(k))}{2} \frac{\lambda_{\min}}{\lambda_{\max}}$, which we shall ensure. This is obtained as follows. Define $a(\mathbf{r}) := \mathbf{r}^\top \boldsymbol{\Sigma}_D \mathbf{r}, b(\mathbf{r}) := \mathbf{r}^\top \boldsymbol{\Sigma}_B \mathbf{r}, e(\mathbf{r}) := \mathbf{r}^\top \mathbf{E}_D \mathbf{r}, f(\mathbf{r}) := \mathbf{r}^\top \mathbf{E}_B \mathbf{r}$. Thus, we get that $|e(\mathbf{r})| \leq \varepsilon_1 \lambda_{\max}, |f(\mathbf{r})| \leq \varepsilon_1 \lambda_{\max}, |a(\mathbf{r})| \geq \lambda_{\min} \left(\sum_{k=0}^q p_k^2 - \sum_{k=0}^q p_k^2 \kappa_1(k)\right)$ and $|b(\mathbf{r})| \geq \lambda_{\min} \left(2 \sum_{k=0}^q p_k^2 - \max\left(0, 2\sum_{k=0}^q p_k^2 \kappa_1(k) - \sum_{k=0}^q p_k^2 \kappa_2(k)\right)\right)$. Notice that $\hat{\rho}(\mathbf{r})/\rho(\mathbf{r}) = (1 + e(\mathbf{r})/a(\mathbf{r}))/(1 + f(\mathbf{r})/b(\mathbf{r}))$. Taking $\varepsilon_1 \leq \frac{(\sum_{k=0}^q p_k^2 - \sum_{k=0}^q p_k^2 \kappa_1(k))}{2} \frac{\lambda_{\min}}{\lambda_{\max}}$ allows us to claim that $|\hat{\rho}(\mathbf{r}) - \rho(\mathbf{r})| \leq \varepsilon_1 \theta(\mathbf{p}) \rho(\mathbf{r})$.

For convenience we denote the normalized projection of any vector $\mathbf{r}$ as $\tilde{\mathbf{r}} := \frac{\boldsymbol{\Sigma}^{1/2} \mathbf{r}}{\|\boldsymbol{\Sigma}^{1/2}\mathbf{r}\|_2}$. Now let $\tilde{\mathbf{r}} \in \mathbb{S}^{d-1}$ be a vector such that $\min\{\|\tilde{\mathbf{r}} - \tilde{\mathbf{r}}_*\|_2, \|\tilde{\mathbf{r}} + \tilde{\mathbf{r}}_*\|_2\} \geq \varepsilon_2$. Hence, using the definitions from Lemma H.4, $|\gamma(\mathbf{r})| \leq 1 - \varepsilon_2^2/2$ while $\gamma(\mathbf{r}_*) = 1$ which implies $\rho(\mathbf{r}_*) - \rho(\mathbf{r}) \geq \kappa_3(\mathbf{p})\varepsilon_2^2/2$. Note that $\rho(\mathbf{r}) \leq \rho(\mathbf{r}_*) = 2 + \kappa_3(\mathbf{p})$. Choosing $\varepsilon_1 < \frac{\kappa_3(\mathbf{p})}{4\theta(2+\kappa_3(\mathbf{p}))}\varepsilon_2^2$, we obtain that $\rho(\mathbf{r}_*)(1 - \theta(\mathbf{p})\varepsilon_1) > \rho(\mathbf{r})(1+\theta(\mathbf{p})\varepsilon_1)$. Using this along with the bound (*) we obtain that w.p. at least $1 - \delta$, $\hat{\rho}(\mathbf{r}_*) > \hat{\rho}(\mathbf{r})$ when $\varepsilon_2 > 0$. Since our algorithm returns $\hat{\mathbf{r}}$ as the maximizer of $\hat{\rho}$, w.p. at least $1 - \delta$ we get $\min\{\|\tilde{\mathbf{r}} - \tilde{\mathbf{r}}_*\|_2, \|\tilde{\mathbf{r}} + \tilde{\mathbf{r}}_*\|_2\} \leq \varepsilon_2$. Using Lemma 2.1, $\min\{\|\hat{\mathbf{r}} - \mathbf{r}_*\|_2, \|\hat{\mathbf{r}} + \mathbf{r}_*\|_2\} \leq 4\sqrt{\frac{\lambda_{\max}}{\lambda_{\min}}}\varepsilon_2$. Substituting $\varepsilon_2 = \frac{\varepsilon}{4}\sqrt{\frac{\lambda_{\min}}{\lambda_{\max}}}$, $\|\mathbf{r} - \mathbf{r}_*\|_2 \leq \varepsilon$ w.p. at least $1 - \delta$. The conditions on $\varepsilon_1$ are satisfied by taking it to be $\leq O\left(\frac{\kappa_3(\mathbf{p})\varepsilon^2 \lambda_{\min}}{\theta(\mathbf{p})(2+\kappa_3(\mathbf{p}))\lambda_{\max}}\right)$, and thus we can take $m \geq O\left((d/\varepsilon^4)\log(d/\delta)\left(\frac{\lambda_{\max}}{\lambda_{\min}}\right)^2 \theta(\mathbf{p})^2 \left(\frac{1}{\kappa_3(\mathbf{p})}\right)^2\right)$. Taking $m \geq O\left((d/\varepsilon^4)\log(d/\delta)\left(\frac{\lambda_{\max}}{\lambda_{\min}}\right)^4 q^4 \left(1/\sum_{k=1}^{q-1} p_k^2\right)^2\right)$ satisfies this using bounds in Defn. H.3. This completes the proof. $\qquad\square$

We observe that the sample complexity bound is worse for $\{p_1, \ldots, p_{q-1}\}$ which are not concentrated i.e., the probability of the label sum is supported over many different values. This occurs for e.g. in the noisy setting when the label flip noise is large (see Appendix F).

