# OpenReview forum: "PAC Learning Linear Thresholds from Label Proportions"
_NeurIPS.cc/2023/Conference — NeurIPS 2023 spotlight_

### Official Review · Reviewer_UNzS · 2023-06-24

**Soundness:** 3 good
**Presentation:** 1 poor
**Contribution:** 3 good
**Rating:** 7
**Confidence:** 3

**Summary:**

Learning from label proportions allow training data to aggregate into sets of feature vectors with sum or average of their labels for each set as a label. As supervised learning, the goal is to classify test set of instances and minimize error of the classifier. This work focuses on learnability of LLP over Gaussian distributions of linear threshold functions. Although PAC learnability of LTFs has been studies in the LLP setting, PAC-LLP is intractable to learn LTFs using LTFs. This work formalize the problem into Gaussian distribution, which makes LTFs PAC-learnable. The approach is directly to maximize the instance-level accuracy on the distribution.

**Strengths:**

1. PAC-learnability of LTFs in the LLP setting (which is a very interesting topic) is well-defined as Definition 1.2 and well-addressed by Theorem 1.3, 1.4 and 1.5, especially considering the case for k = q/2.
2. The proof of Theorem 1.4 is sound and solid coming with an algorithm and lemmas (even proofs for lemmas).
3. As a theoretical work, it is nice to have a section for experimental results, although the datasets look relatively simple.

**Weaknesses:**

1. For Theorem 1.3 and Theorem 1.5, it is better to provide one or two sentences for each about proofs instead of putting everything in the appendix. You dont have to provide a separate section like Theorem 1.4. Just few sentences please.
2. The writing looks less readable for some sentences. Please refer to limitations section.

**Questions:**

1. In Theorem 1.5, what is \lambda? What are \lambda_max and \lambda_min? Please clarify notations like what you did on Line 126.
2. What are relationships for r* and r head in Theorem 1.3, 1.4 and 1.5? I know you explained this on Line 145. Please put it ahead.

**Limitations:**

1. If possible, please don’t include citations in the abstract.
2. On Line 59, I guess you typed one more “level”.
3. Please make “Theorem” (Like on Line 247 or 257) and “Thms.” (Like on Line 128) consistent (Use either one).
4. Need a space between “independence” and “i.e.,” on Line 132.
5. Similarly, for Line 148, need a space between “directions” and “We”.
6. You should read over your manuscript carefully and try to rephrase some sentences to make them more readable (I wont correct them here verbatim).
- Some sentences have no subject.
- Some sentences have no periods or spaces like limitations 4 and 5.
- Some pronouns refer ambiguously.

---

> ### Author Rebuttal · Authors · 2023-08-09
>
> Q. *For Theorem 1.3 and Theorem 1.5, it is better to provide one or two sentences for each about proofs instead of putting everything in the appendix. You dont have to provide a separate section like Theorem 1.4. Just few sentences please.*
> *Authors*: While we have included in Section 1.4 an overview of our techniques for Theorems 1.3 and 1.5, we accept the suggestion and will also add a paragraph after Section 4 in the main paper with an informal description of the proofs of Theorems 1.3 and 1.5 along with references to the respective appendices with the formal proofs.
>
> Q. *In Theorem 1.5, what is \lambda? What are \lambda_max and \lambda_min? Please clarify notations like what you did on Line 126.*
> *Authors*: $\\lambda_{\\max}$ and $\\lambda_{\\min}$ are the maximum and minimum eigenvalues respectively of $\\mathbf{\\Sigma}$, and we will explicitly state this before the statements of Theorems 1.4 and 1.5.
>
> Q. *What are relationships for r\* and r head in Theorem 1.3, 1.4 and 1.5? I know you explained this on Line 145. Please put it ahead.*
> *Authors*: The value of $\\hat{\\mathbf{r}}$ output by our algorithms is a close estimate of $\\mathbf{r}^*$ (or possibly $-\\mathbf{r}^*$ in the case of balanced bags) obtained via the mean-estimation (Theorem 1.3) or the PCA based method (Theorems 1.4 and 1.5). We will add this in Section 1.3 as well.
>
> We are grateful to the Reviewer for pointing out inadvertent typos and some editorial feedback. We shall of course correct all the typos as well as improve the phrasing of sentences wherever required.

---

> > ### Comment · Reviewer_UNzS · 2023-08-13
> > **Thanks for rebuttal**
> >
> > Thank authors for their rebuttal and it clears all my concerns. Thanks!

---

### Official Review · Reviewer_FWVX · 2023-06-30

**Soundness:** 4 excellent
**Presentation:** 3 good
**Contribution:** 3 good
**Rating:** 7
**Confidence:** 3

**Summary:**

This theoretical paper investigates the learnability of linear threshold functions (LTFs) in the learning from label proportions (LLP) setting. When the feature-vectors are distributed according to a Gaussian distribution and conditioned on their underlying labels, LTFs can be efficiently properly learnt. For the general Gaussian distribution, this paper develops a principal component algorithm which estimates the means and covariance matrices using subgaussian concentration bounds. Generalization bounds from bag classification error to instance classification error are also provided to resolve the ambiguity between the obtained solutions. The experimental results validate the effectiveness of the proposed techniques.

**Strengths:**

The learnability of LTFs in the LLP setting is an important issue, this work provides some novel and significant theoretical results.

**Weaknesses:**

$\bullet$ From Theorem 1.3 to 1.5, as the distribution becomes more general, the sample complexity required to efficiently properly learn LTFs also increases, and the difference between these sample complexities seems to be obvious. Numerical results on the difference between these sample complexities and associated discussion and analysis are necessary to more intuitively elucidate these theoretical results.

$\bullet$ Intuitively, the theoretical results in this paper will depend on the label proportion $k/q$, but in the end, it is mainly implemented on the bag size $q$, i.e., the order of the sample complexity does not explicitly reflect the label proportion $k/q$. While this is not mathematically problematic, it would be helpful to add some remarks to illustrate this point.

$\bullet$ The readability of the paper needs to be improved. More introductions to the problem setting and related definitions of LLP, as well as further illustrations and explanations about the existing theoretical work will help readers better understand the theoretical results in this paper. In addition, some typos also harm the readability.

Typos:

$\bullet$ Line 27: they're labels The goal -> their labels. The goal

$\bullet$ Line 132: independencei.e., -> independence, i.e.,

$\bullet$ Line 146: $r = q/2$ -> $k = q/2$

$\bullet$ Line 148: directionsWe -> directions. We

$\bullet$ Line 350: bag satisfaction -> bag classification

$\bullet$ Some typos in the Supplementary Material also need to be checked and corrected.

**Questions:**

Please refer to the Weaknesses for details.

**Limitations:**

This work does not seem to have any potential negative societal impact.

---

> ### Author Rebuttal · Authors · 2023-08-09
>
> Q. *From Theorem 1.3 to 1.5, as the distribution becomes more general, the sample complexity required to efficiently properly learn LTFs also increases, and the difference between these sample complexities seems to be obvious. Numerical results on the difference between these sample complexities and associated discussion and analysis are necessary to more intuitively elucidate these theoretical results.*
> *Authors*: We do include experiments in Appendix F on the setting of Theorems 1.3, 1.4 and 1.5, focusing on the performance improvement against existing baselines of [Saket 21, 22] and random LTF. For example, from Tables 3 and 4 we observe that Algorithm 2 (without LTF offset) has better accuracy than Algorithm 4 (with offset) for the same sample complexity, which is consistent with our theoretical results. We will include in Section 5, a discussion of the performance and sample complexity for the various settings.
>
> Q. *Intuitively, the theoretical results in this paper will depend on the label proportion , but in the end, it is mainly implemented on the bag size, i.e., the order of the sample complexity does not explicitly reflect the label proportion. While this is not mathematically problematic, it would be helpful to add some remarks to illustrate this point.*
> *Authors*: The sample complexity depends on both $q$ and $k$, however for ease of notation we have used the upper bound of $q$ for $k$ as well.
> Appendix A gives the proof of Theorem 1.3 where the sample complexity solely depends on $k/q$ (refer to the expression for $\\eta(k, q)$). The algorithm (Algorithm 3) samples a single feature-vector from each bag to compute a mean estimate. Thus, the probability of sampling it from the positive half should only depend on $k/q$ and the sample complexity therefore only depends on $k/q$.
> However, in Algorithm 2, we sample pairs of feature-vectors with and without replacement. When sampling without replacement, the probabilities of the label configurations of the two feature-vectors depend on the size of the bag $q$, and not only on $k/q$. In particular, the probability of sampling a pair of differently labeled feature-vectors without replacement is $2(k/q)(1-k/q)/(1-1/q)$. Keeping $k/q$ the same, this probability decreases with increasing bag size which increases the sample complexity for larger bags. We shall add this explanation in Section 1.4.
>
> Q. *The readability of the paper needs to be improved. More introductions to the problem setting and related definitions of LLP, as well as further illustrations and explanations about the existing theoretical work will help readers better understand the theoretical results in this paper. In addition, some typos also harm the readability.*
> *Authors*: We will add more discussion in Section 1 to describe the LLP setting, and also further explain the existing body of theoretical work in LLP.
>
> We thank the Reviewer for pointing out the inadvertent typos and we will fix them along with any others in the paper as well as the supplementary material.

---

> > ### Comment · Reviewer_FWVX · 2023-08-18
> >
> > Thanks to the authors for their rebuttal, and the authors' responses have addressed my concerns.

---

### Official Review · Reviewer_Xa3y · 2023-07-03

**Soundness:** 4 excellent
**Presentation:** 4 excellent
**Contribution:** 4 excellent
**Rating:** 8
**Confidence:** 4

**Summary:**

This work studies the problem of learning linear threshold functions (LTFs, aka linear classifiers) under the setting of learning from label proportions (LLP), where the training data are "bags" (aka sets) of instances, and the training labels are classifier proportions of the instances in the bag. Under the assumption that the instances are gaussian distributed, bags are instances sampled iid, and the size of the bags are upper bounded, the authors show that it is possible to efficiently (properly) learning LTFs.

**Strengths:**

The problem setting of PAC-LLP deserves more attention.
The authors gives good justification for why LLP matters, e.g., privacy and legal.
In light of the previous theory works on LLP that shows NP-hardness of PAC-LLP learning LTFs,
this current work is surprising, relvant, and interesting to the ML community.
Theorem 1.4 gives an algorithm that highlights the interesting geometry and the clever exploitation of the subtle difference between sampling with and without replacement.

**Weaknesses:**

After Section 1.1, it will be useful for the reader if there is an overview of the paper. Something along the lines of
"Sec 1.3 are the main results. Sec 1.4 gives proof sketch of the main results and high level description of the algorithms. Section 3 state these algorithms precisely..."

For Theorem 1.3, it will be useful to interpret the sample complexity. Is the sample complexity essentially the difficulty of estimating the mean of the bag vectors? If so, the authors should make this explicit.

For Theorem 1.4 and 1.5, are the lambdas eigenvalues of the covariance matrices? Please explicitly say so.

**Questions:**

Why is Definition 1.2 called the bag oracle? Isn't it just the distribution over the training data (of bags)?

Does the authors believe that the results are tight? In other words, is LLP under the considered data assumption essentially equivalent to mean/covariance estimation?

It will be good if there is a discussion regarding the sample-complexities in LLP versus ordinary classification.
From this point of view, what is the "cost" of only having access to bagged data?

**Limitations:**

Yes.

---

> ### Author Rebuttal · Authors · 2023-08-09
>
> Q. *After Section 1.1, it will be useful for the reader if there is an overview of the paper. Something along the lines of "Sec 1.3 are the main results. Sec 1.4 gives proof sketch of the main results and high level description of the algorithms. Section 3 state these algorithms precisely..."*
> *Authors*: We accept this suggestion and will add a paragraph after Section 1.1 providing the organization of the paper.
>
> Q. *For Theorem 1.3, it will be useful to interpret the sample complexity. Is the sample complexity essentially the difficulty of estimating the mean of the bag vectors? If so, the authors should make this explicit.*
> *Authors*: Yes, the sample complexity in Theorem 1.3 is essentially the same as that of mean estimation of the bag-vectors up to a desired error. We note however, that this distribution is not a Gaussian due to having unbalanced bags. We will add this to the overview in Section 1.4 and also to the proof of Theorem 1.3 in Appendix A.
>
> Q. *For Theorem 1.4 and 1.5, are the lambdas eigenvalues of the covariance matrices? Please explicitly say so.*
> *Authors*: Yes, $\\lambda_{\\min}$ and $\\lambda_{\\max}$ denote the minimum and maximum eigenvalues of the covariance matrix $\\mathbf{\\Sigma}$ of the distribution from which the feature-vectors are sampled. We will state this before Theorems 1.4 and 1.5.
>
> Q. *Why is Definition 1.2 called the bag oracle? Isn't it just the distribution over the training data (of bags)?*
> *Authors*: Typically in PAC learning literature, the underlying distribution is defined over the feature-vectors. Given an (unknown) classifier, the corresponding *example oracle* samples a feature-vector according to the distribution and outputs it along with the label assigned by the classifier. We extend this to the case of bags in which the bag oracle – additionally parameterized with a size $q$ and label sum $k$ – samples $k$ iid $1$-labeled feature-vectors and $(q-k)$ iid $0$-labeled feature-vectors and outputs a bag consisting of them.
>
> Q. *Does the authors believe that the results are tight? In other words, is LLP under the considered data assumption essentially equivalent to mean/covariance estimation?*
> *Authors*: The sample complexity lower bound for PAC-LLP learning LTFs is a very relevant problem, however it is out of the scope of this work. While there is a blowup in the sample complexity incurred by the generalization and the stability bounds (Theorem 2.2. , Lemma 2.3), a major component is indeed the mean/covariance estimation (Algorithm 1). Nevertheless, one cannot rule out other algorithmic techniques for this problem which bypass such estimation.
>
> Q. *It will be good if there is a discussion regarding the sample-complexities in LLP versus ordinary classification. From this point of view, what is the "cost" of only having access to bagged data?*
> *Authors*: For ordinary classification in $d$-dimensional space, the sample complexity is essentially $O(d\\log d)$ as one can solve a linear program to obtain an LTF and then use uniform convergence to bound the generalization error. While our algorithms have the same dependence on $d$, we shall add a discussion in Section 1.3 on the blowup incurred in the LLP setting due to the bag size, the condition number of the covariance matrix and the other geometric and error parameters.

---

> > ### Comment · Reviewer_Xa3y · 2023-08-14
> >
> > I thank the authors for their thorough reply. Based on the excellent submission and rebuttal, I have raised my score.

---

### Official Review · Reviewer_VQSz · 2023-07-08

**Soundness:** 4 excellent
**Presentation:** 3 good
**Contribution:** 3 good
**Rating:** 7
**Confidence:** 3

**Summary:**

This paper studies PAC learning when the training data is aggregated into sets or bags of feature vectors. For each bag, we observe the feature vectors of these bags and only the average of the labels in the bag. They focus on the case when the feature vectors are distributed according to a Gaussian distribution, and the hypothesis class is the set of Linear Threshold Functions (LTFs). Their main results include polynomial time algorithms for learning the correct halfspace when the Gausssian is either standard or skewed and the correct LTF is either homogeneous or not. Finally, they compare their Algorithm experimentally with various procedures from prior work.

**Strengths:**

This paper proposes a very interesting direction for learning under label aggregation, which bypasses the lower bounds that existed for this setting in prior work. I think it opens the way for more interesting problems to be considered in this setting, by possibly changing the assumptions on the distribution or the hypothesis class. All the claims are sufficiently explained and the presentation is generally clear.

The authors introduce some interesting technical novelties to extract information about the true vector of the LTF using the aggregated labels. In particular, the idea of comparing the variance of a random pair with and without replacement and finding that it is maximized in the direction of the true vector $\mathbf{r^\star}$ is very elegant and leads to some interesting algorithms, which can be implemented in polynomial time using PCA. Moreover, the techniques for handling non-centered and skewed Gaussians with unknown parameters are non-trivial and interesting.

**Weaknesses:**

This is not a significant weakness, but the technical presentation in some parts of the paper seems quite dense and not necessarily adding to the understanding of this work. For example, in the proof of Theorem 1.4 or Lemma 4.2, too much emphasis is given in the calculation of the optimal setting of parameters and sample complexity. I would prefer a more intuitive explanation at times, although I'm not sure it that's possible.

**Questions:**

How tight are the sample complexities provided in Theorem 1.4 and 1.5 in terms of their dependence on the minimum and maximum eigenvalues of the covariance matrix and the size of the bag? In Theorem 1.5, we also see a term involving $l$, which depends on the parameters of the true LTF. Is such a dependence necessary?



**Limitations:**

No limitations.

---

> ### Author Rebuttal · Authors · 2023-08-09
>
> Q. *For example, in the proof of Theorem 1.4 or Lemma 4.2, too much emphasis is given in the calculation of the optimal setting of parameters and sample complexity. I would prefer a more intuitive explanation at times..*
> *Authors*: We state our results formally in Section 1.3 and therefore include the parametric dependencies explicitly. However, to aid a more intuitive understanding, we will include an informal description of how the parameters in Theorems 1.4 and 1.5 are obtained and what they convey qualitatively in Section 1.4. Also, in the proofs of Lemmas 4.1 and 4.2 we shall add more explanations along the way for ease of understanding.
>
> Q. *How tight are the sample complexities provided in Theorem 1.4 and 1.5 in terms of their dependence on the minimum and maximum eigenvalues of the covariance matrix and the size of the bag?*
> *Authors*: We have optimized the dependencies on the eigenvalues and the bag size in our application of the algorithmic techniques used in our results. However, sample complexity lower bounds for PAC-LLP learning LTFs – while a very relevant problem – is out of the scope of this work.
>
> Q. *In Theorem 1.5, we also see a term involving l, which depends on the parameters of the true LTF. Is such a dependence necessary?*
> *Authors*: The term $l$ tells us the perpendicular distance from the center of the Gaussian to the unknown LTF's hyperplane, normalized by the stretch induced by $\\mathbf{\\Sigma}$ in the direction of $\\mathbf{r}^*$. This is required to estimate the density of the Gaussian distribution near the unknown LTF's hyperplane which directly affects the sample complexity – the less the density, the more the sample complexity. We will add this explanation in the relevant portion of Section 1.4.

---

### Author Rebuttal · Authors · 2023-08-09

We thank the Reviewers for their encouraging and helpful feedback. We have addressed their questions and comments in the respective author rebuttals to the reviews.

---

### Decision · Program_Chairs · 2023-09-21

**Decision:**

Accept (spotlight)

**Comment:**

The  PAC learning of linear threshold functions (LTFs) from label proportions was shown to be intractable on hard bag distributions. This paper proposed a very appealing direction for learning under label aggregation, which shows that there indeed exist polynomial time algorithms for learning the correct halfspace when the feature vectors are Gaussian. The initial concerns are well addressed in the rebuttal. The reviewers unanimously agree that this is a technically solid paper and recommend its acceptance.    Following the strong recommendation from the reviewers, I recommend its acceptance as a spotlight paper.